# Chemical composition of isoprene SOA under acidic and non-acidic conditions: Effect of relative humidity

Klara Nestorowicz[1], Mohammed Jaoui[2], Krzysztof Jan Rudzinski[1], Michael Lewandowski[2], Tadeusz E. Kleindienst[2], Grzegorz Spólnik[3], Witold Danikiewicz[3] and Rafal Szmigielski[1]

5 [1]Environmental Chemistry Group, Institute of Physical Chemistry Polish Academy of Sciences, 01-224 Warsaw, Poland
[2]US Environmental Protection Agency, 109 T.W. Alexander Drive, RTP NC, USA, 27711.
[3]Mass Spectrometry Group, Institute of Organic Chemistry, Polish Academy of Science, 01-224 Warsaw, Poland

*Correspondence to*: Rafal Szmigielski (ralf@ichf.edu.pl); Mohammed Jaoui (jaoui.mohammed@epa.gov)

10 **Abstract.** The effect of acidity and relative humidity on bulk isoprene aerosol parameters has been investigated in several studies, however few measurements have been conducted on individual aerosol compounds. The focus of this study has been the examination of the effect of acidity and relative humidity on secondary organic aerosol (SOA) chemical composition from isoprene photooxidation in the presence of nitrogen oxide (NOx). A detailed characterization of SOA at the molecular level was 15 also investigated. Experiments were conducted in a 14.5 $m^3$ smog chamber operated in flow mode. Based on a detailed analysis of mass spectra obtained from gas chromatography-mass spectrometry of silylated derivatives in electron impact and chemical ionization modes, and ultra-high performance liquid chromatography/electrospray ionization/time-of-flight high resolution mass spectrometry, and collision-induced dissociation in the negative ionization modes, we characterized not only typical 20 isoprene products, but also new oxygenated compounds. A series of nitroxy-organosulfates (OS) were tentatively identified on the basis of high resolution mass spectra. Under acidic conditions, the major identified compounds include 2-methyltetrols (2MT), 2-methylglyceric acid (2MGA) and 2MT-OS. Other products identified include epoxydiols, mono- and dicarboxylic acids, other organic sulfates, and nitroxy- and nitrosoxy-OS. The contribution of SOA products from isoprene oxidation to $PM_{2.5}$ was 25 investigated by analysing ambient aerosol collected at rural sites in Poland. Methyltetrols, 2MGA and several organosulfates and nitroxy-OS were detected in both the field and laboratory samples. The influence of relative humidity on SOA formation was modest in non-acidic seed experiments, and stronger under acidic seed aerosol. Total secondary organic carbon decreased with increasing relative humidity under both acidic and non-acidic conditions. While the yields of some of the specific organic 30 compounds decreased with increasing relative humidity others varied in an indeterminate manner from changes in the relative humidity.

**Keywords:** Isoprene, relative humidity, acidity, SOA, organosulfates

## 1 Introduction

Secondary organic aerosol (SOA) is formed through complex physico-chemical reactions of volatile organic compounds which are emitted into the atmosphere from biogenic and anthropogenic sources and can constitute a substantial portion of the continental aerosol mass (Goldstein and Galbally, 2007; Hallquist et al. 2009). Of the volatile organic compounds, isoprene is the most abundant non-methane hydrocarbon emitted to the atmosphere (Guenther et al., 1995, 2006). Although the SOA yield of isoprene tends to be low, its sizable emissions can contribute to a high organic aerosol loading making it one of the most studied compounds for aerosol formation (Guenther et al., 1995; Henze and Seinfeld, 2006; Fu et al., 2008; Carlton et al., 2009; Hallquist et al., 2009). The primary removal mechanism for isoprene is by gas-phase reactions with hydroxyl radicals (OH), nitrate radicals and, to a lesser extent, ozone. These processes result in the formation of gas and aerosol products including numerous oxidized SOA components. Aerosol species previously reported included 2-methyltetrols, 2-methylglyceric acid, $C_5$-alkene triols and organosulfates) (i.e. Edney et al., 2005; Surratt et al., 2007a, 2010; Riva et al., 2016; Spolnik et al., 2018). While many of these are formed through multiphase chemistry (e.g. IEPOX channel), we cannot exclude their gas phase formation at least for 2-methyltetrols, probably in part through re-evaporation processes (Issacman-VanWertz et al. 2016), and for 2-methylglyceric acid, as these compounds have been linked to gas phase reaction products from the oxidation of isoprene (Kleindienst et al., 2009) and in ambient $PM_{2.5}$ (Xie at al., 2014). Moreover, these compounds were identified in ambient $PM_{2.5}$ in several places around the world, and SOA from isoprene often accounts for 20–50% of the overall SOA budget (Claeys et al., 2004a; Wang et al., 2005; Henze and Seinfeld, 2006; Kroll et al., 2006; Surratt et al., 2006; Hoyle et al., 2007).

An enhancement of isoprene (ISO)-SOA yields is controlled by various factors including $NO_X$ concentration (Kroll et al., 2006; Chan et al., 2010; Surratt et al., 2006, 2010) and the acidity of preexisting aerosol (Jang et al., 2002; Czoschke et al., 2003; Edney et al., 2005; Kleindienst et al., 2006; Surratt et al., 2007a, 2010; Jaoui et al., 2010; Szmigielski et al., 2010). The strength of the acidity depends on the aerosol liquid water content and the relative humidity (Nguyen et al., 2011; Zhang et al., 2011; Lewandowski et al., 2015; Wong et al., 2015) which are coupled. Smog chamber experiments have revealed that the yield of isoprene SOA increases under acidic conditions through an enhanced formation of isoprene-derived oxygenates by acid-catalyzed reactions (Surratt et al., 2007b, 2008, 2010; Gomez-Gonzalez et al., 2008; Offenberg et al., 2009). By one mechanism, isoprene reactions with OH under low- or high-$NO_x$ conditions can form epoxydiols (IEPOX) in high yields followed by their uptake by SOA and subsequent acid-catalyzed particle reactions (Paulot et al., 2009; Surratt et al., 2010; Lin et al., 2013; Budisulistiorini et al., 2015; Rattanavaraha et al., 2016; Gaston et al., 2014a,b; Riedel et al., 2015; Zhang et al., 2018). However, this type of multiphase chemistry following the uptake of IEPOX can be highly dependent on the aerosol phase state and the presence of aerosol coatings from viscous SOA constituents (Zhang et al., 2018). Such coatings can cause a substantial diffusion barrier to the availability to an acidic core.

Atmospheric organosulfates are another class of organic compounds formed from atmospheric reactions of various precursors, including isoprene, and have been identified as components of ambient PM

(Surratt et al., 2008; Froyd et al. 2010; Stone et al., 2012; Tolocka and Turpin, 2012). The most common isoprene organosulfates have been identified both in smog chamber experiments and in field studies (Surratt et al., 2007a; 2008, 2010; Gomez-Gonzalez et al., 2008; Shalamzari et al., 2013; Tao et al., 2014; Hettiyadura et al., 2015; Szmigielski, 2016; Spolnik et al., 2018). For many of these polar oxygenated compounds, chemical structures, MS fragmentation patterns and formation mechanisms have been tentatively proposed (Surratt et al., 2007a,b; 2008, 2010; Gomez-Gonzalez et al., 2008; Zhang et al., 2011; Shalamzari et al., 2013; Schindelka et al., 2013; Nguyen et al., 2014; Tao et al., 2014; Hettiyadura et al., 2015; Riva et al., 2016; Spolnik et al., 2018). The commonly detected components of isoprene SOA attributed to processing of isoprene oxidation products (e.g., IEPOX, methacrolein and methyl vinyl ketone) have the reported molecular weights of 154, 156, 184, 198, 200, 212, 214, 216, 260, and 334 (Surratt et al., 2007b, 2008, 2010; Gomez-Gonzalez et al., 2008; Kristensen et al., 2011; Zhang et al., 2011; Shalamzari et al., 2013; Schindelka et al., 2013; Nguyen et al., 2014; Hettiyadura et al., 2015; Riva et al., 2016). The mechanisms of OS formation were proposed for the conditions of either acidified or non-acidified sulfate aerosol seeds (e.g. 2-methyltetrol organosulfates proposed by Surratt et al. (2007a) and Riva et al. (2016)). Whereas Kleindienst et al. (2006) reported the formation of highly oxygenated products through OH radical oxidation, Riva et al. (2016) proposed an alternative route through acid-catalyzed oxidation by organic peroxides. Isoprene organosulfates were also reported to occur in the aqueous-phase through the photooxidation or dark reactions of isoprene in aqueous solutions containing sulfate and sulfite moieties (Rudzinski et al., 2004, 2009; Noziere et al., 2010). A detailed mechanism of this transformation has been tentatively proposed based on chain reactions propagated by sulfate and sulfite radical anions (Rudzinski et al., 2009) and confirmed by mass spectrometric studies (Szmigielski, 2016). The acid-catalyzed formation of 2-methyltetrols has also been suggested in aqueous phase oxidation of isoprene with $H_2O_2$ (Claeys et al., 2004b).

To date, few smog-chamber studies have examined the effect of relative humidity on ISO-SOA formation (Dommen et al., 2006; Nguyen et al., 2011; Zhang et al., 2011; Lewandowski et al., 2015; Wong et al., 2015; Riva et al., 2016). However, the impact of relative humidity may be an important parameter, in that, it may influence the mechanism of SOA formation and hence the chemical composition, physical properties and yield of isoprene SOA (de P. Vasconcelos et al., 1994; Poulain et al., 2010; Guo et al., 2014). The chamber studies conducted by Dommen et al. (2006) and Nguyen et al. (2011) showed a negligible effect of relative humidity on the SOA yield from the photooxidation of isoprene in the absence of sulfate aerosol. Other studies suggested that ISO-SOA formation yields under high-$NO_x$ conditions with acidified and non-acidified sulfate aerosol decreased with an increase in relative humidity while simultaneously the yield of organosulfates was enhanced (Zhang et al., 2011; Lewandowski et al., 2015). The latter observation can be explained by transformation of isoprene propagated by sulfate/sulfite radical-anions in the aqueous particle phase or on the aqueous surface of aerosol particles (Zhang et al., 2011; Rudzinski et al., 2016; Szmigielski, 2016). The results obtained from the chamber experiments have been in agreement with recent model approaches, when reactive uptake to aqueous aerosol is used rather than a reversible partitioning approach (Pye et al., 2013; Marais et al., 2016). A recent study conducted in our laboratory

focused on the effects of relative humidity on secondary organic carbon (SOC) formation from isoprene photooxidation in the presence of $NO_x$ (Lewandowski et al., 2015). The study indicated that relative humidity can have a profound effect on the acid-derived enhancement of isoprene SOC, while an increasing content of aerosol liquid water suppressed the level of enhancement.

The focus of the present study is to investigate at a molecular level the role of relative humidity on the chemical composition of isoprene SOA obtained under acidic and non-acidic conditions. Organosulfate compounds were analysed using LC/MS measurements (Szmigielski, 2016; Rudzinski et al., 2009; Darer et al., 2011; Surratt et al., 2007a), while non-sulfate oxygenated compounds were examined using derivatization followed by GC-MS analysis (Jaoui et al., 2004). Here we explored the RH effect of a wide range on isoprene

polar oxygenated products, including, 2-methyltetrols, 2-methylglyceric acid, IEPOX, organosulfates), nitroxy-organosulfates (NOS) and other selected oxygenates in the presence of acidified and non-acidified sulfate aerosol. In addition, a chemical analysis of $PM_{2.5}$ field samples has been conducted to assess the possible relationship between the laboratory findings and their role in ambient SOA formation.

**2 Experimental Methods**

**2.1 Smog chamber experiments**

        Chamber experiments were conducted in a stainless-steel, 14.5 $m^3$ fixed volume chamber with interior walls fused with a 40-$\mu$m PTFE Teflon coating. Details of chamber operation, sample collection, derivatization procedure, and gas chromatography–mass spectrometry (GC-MS) analysis method are described in more detail in

Lewandowski et al. (2015), and Jaoui et al. (2004). A combination of UV-fluorescent bulbs was used in the chamber as source of radiation from the 300-400 nm with a distribution photolytically comparable to that of solar radiation (Black et al., 1998). The reaction chamber was operated as a flow reactor with a residence time of 4 h, to produce a steady-state, constant aerosol distribution which could be repeatedly sampled at different seed aerosol acidities.

Isoprene and nitric oxide (NO) were taken from high-pressure cylinders each diluted with $N_2$. Isoprene was obtained from Sigma-Aldrich Chemical Co. (Milwaukee, WI, USA) at the highest purity available and used without further purification. Isoprene and NO were added to the chamber through flow controllers. The temperature in all experiments was ∼ 27 °C (Table 1). Dilute aqueous solutions of ammonium sulfate and sulfuric acid as inorganic seed aerosol were nebulized to the chamber with total sulfate concentration of the combined

solution held constant to maintain stable inorganic concentrations in the chamber (Lewandowski et al., 2015). NO and total oxides of nitrogen ($NO_X$) were measured with a ThermoElectron $NO_X$ analyzer (Model 8840, Thermo Environmental, Inc., Franklin, MA). Ozone formed during the irradiation was measured with a Bendix ozone monitor (Model 8002, Lewisburg, WV). Temperature and relative humidity were measured with an Omega Digital Thermo-Hydrometer (Model RH411, Omega Engineering, Inc., Stamford, CT). Isoprene concentrations were

measured by gas chromatography with flame ionization detection (Hewlett-Packard, Model 5890 GC). Chamber filter samples were collected for 24 h at 16.7 L min$^{-1}$ using 47-mm glass fiber filters (Pall Gelman Laboratory, Ann Arbor, MI, USA).

Two sets of experiments were conducted (Table 1) to explore the effect of humidity and acidity on
isoprene SOA products. The non-acidic experiment (ER667) was conducted at four different humidity levels in the presence of isoprene, $NO_x$ and ammonium sulfate as seed aerosol (1 µg m$^{-3}$). It served as a base case for exploring the changes and nature of SOA products in the absence of significant aerosol acidity. The second experiment ER662 (acidic) was similar but run in the presence of acidic seed aerosol at constant concentration. It included 5 and 4 stages differing in humidity levels for ER667 (9%; 19%; 30%; 39%; and 49%) and ER662 (8%;
18%; 28%; and 44%) respectively. Aerosol concentrations are those from Lewandowski et al. (2015).

**2.2 Ambient aerosol samples.**

Twenty ambient PM$_{2.5}$ samples were collected, onto pre-baked quartz filters using a high-volume aerosol sampler (DH-80, Digitel), from two sites (ten samples each) having strong isoprene emissions: (1) a regional
background monitoring station in Zielonka, in the Kuyavian-Pomeranian Province in the northern Poland (PL; 53°39' N, 17°55' E) during summer 2016 campaign, and (2) a regional background monitoring station in Godow, PL located in the Silesian Province (49°55' N, 18°28' E) in summer 2014 campaign. Sampling times were 12 and 24 hours, respectively. Major tree species at both sites are European oak (*Quercus robur,* L.); European hornbeam (*Carpinus betulus*, L.); Tilia cordata (*Tilia cordata*, Mill); European white birch (*Betula pubescens*, Ehrh); and
European alder (*Alnus glutinosa*, Gaertn). The Zielonka station is in a forested area while the Godow station is located near a coal-fired power station in Detmarovice (Czech Republic). Godow is also close to the major industrial cities of the Silesian region in Poland, and thus aerosol samples collected in Godow were influenced by anthropogenic sources.

Several chemical and physical parameters were measured at the two sites. The temperature during
sampling at both sites ranged from 25-28 °C. The relative humidity during sampling was up to 86% in Zielonka and 94% at Godow. Both locations were influenced by $NO_x$ concentration, modestly in Zielonka at 1.3 µg m$^{-3}$ and at a level of 30 µg m$^{-3}$ in Godow, represented by the nearest monitoring station at Zywiec, PL. The $SO_2$ levels at Zielonka were approximately 0.6 µg m$^{-3}$ and 3.0 µg m$^{-3}$ at Godow. At each site, OC/EC values was determined for each filter using a thermo-optical method (Birch and Cary, 1996). The organic carbon value at Zielonka was
approximately 1.7 µg m$^{-3}$ and 5.4 µg m$^{-3}$ at Godow, although aerosol masses were not determined.

**2.3 Instrumentation and analysis methods.**

Chemicals for extraction and derivatization were obtained from Sigma-Aldrich Chemical Company. *N,O*-bis(trimethylsilyl)-trifluoroacetamide (BSTFA) used as the derivatizing agent included 1% trimethylchlorosilane as a catalyst. For the GC-MS analysis, filters were sonicated for one hour with methanol.

Prior to extraction, 20 µg each of *cis*-ketopinic acid and $d_{50}$-tetracosane were added as internal standards. Following sonication, the methanol extracts were dried and then derivatized with 200 µL BSTFA and 100 µL pyridine. Samples were then heated at 70 °C to complete the reaction (Jaoui et al., 2004). The derivatized extracts were analyzed using a ThermoQuest (Austin, TX, USA) GC coupled to an ion trap mass spectrometer (ITMS).

The injector, heated to 270 °C, was operated in splitless mode. Compounds were separated on a 60-m-long, 0.25-mm-i.d. RTx-5MS column (Restek, Inc., Bellefonte, PA, USA) with a 0.25-µm film thickness. The GC oven temperature program for the analysis started isothermally at 84 °C for 1 min, followed by a temperature ramp of 8 °C $min^{-1}$ to 200 °C, followed by a 2-min hold, then ramped at 10 °C $min^{-1}$ to 300 °C. The ion source, ion trap, and interface temperatures were 200, 200, and 300 °C, respectively. Mass spectra were collected in both the

chemical ionization (CI) and electron ionization (EI) modes (Jaoui et al., 2004). A semi-continuous organic carbon/elemental carbon (OC/EC) analyzer (Sunset Laboratories, Tigard, OR) measured total organic carbon of the aerosol given the absence of elemental carbon in the reaction system. Immediately upstream of the analyzer, a carbon-strip denuder was placed in line to remove gas-phase organic components which could bias the measurements. The analyses for total OC were made on a 15-min duty cycle. Silylations of polar compounds

result in reduced polarity, enhanced volatility and increased thermal stability, and enables the GC-MS analysis of many compounds otherwise involatile or too unstable for these techniques. Therefore, appropriate caution should be taken, for example, with desulfation reactions associated with primary organosulfates (Takano et al., 1992; Kolender et al., 2004; Bedini et al., 2006; Bedini et al., 2017; Cui et al., 2018), and corrections might be warranted when analyzing methyltetrols.

For the LC/MS analysis, from each filter, two 1 $cm^2$ punches were taken and twice extracted for 30 min with 15 mL aliquots of methanol using a Multi-Orbital Shaker (PSU-20i, BioSan). High purity methanol (LC-MS ChromaSolv-Grade; Sigma-Aldrich, PL) was used for the extraction of SOA filters, reconstitution of aerosol extracts, and preparation of the LC mobile phase. The two extracts were combined and concentrated to 1 mL using a rotary evaporator operated at 28 °C and 150 mbar (Rotavapor® R215, Buchi). They were then filtered with 0.2

25 µm PTFE syringe and taken to dryness under a gentle stream of nitrogen. High-purity water (resistivity 18.2 MΩ·$cm^{-1}$) from a Milli-Q Advantage water purification system (Merck, Poland) was used for the reconstitution of aerosol extracts and preparation of the LC mobile phase. The residues were reconstituted with 180 µL of 1:1 high purity methanol/water mixture (v / v), then agitated for 1 min. Recoveries were not taken for compounds analysed in this study, due to lack of authentic standards, however recovery of 94 -101% were measured for

appropriate surrogate compounds.

Extracts were analyzed by ultra-high performance liquid chromatography/electrospray ionization/time-of-flight high resolution mass spectrometry (UHPLC / ESI (-) QTOF) HRMS equipment consisting of a Waters Acquity UPLC I-Class chromatograph coupled to a Waters Synapt G2-S high resolution mass spectrometer. The chromatographic separations were performed using an Acquity HSS T3 column (2.1×100 mm, 1.8 µm particle

size) at room temperature. The mobile phases consisted of 10 mM ammonium acetate (eluent A) and methanol

(eluent B). To obtain appropriate chromatographic separations and responses, a gradient elution program 13 min in length was used. The chromatographic run commenced with 100% eluent A over the first 3 min. Eluent B increased from 0-100% from 3 to 8 min, held constant at 100 % from 8 to 10 min, and then decreased back from 100-0% from 10 to 13 min. The initial and final flow was 0.35 mL min$^{-1}$ while the flow from 3 to 10 min was 0.25 mL min$^{-1}$. An injection volume of 0.5 µL was used. The Synapt G2-S spectrometer equipped with an ESI source was operated in the negative ion mode. Optimal ESI source conditions were 3 kV capillary voltage, 20 V sampling cone at a FWHM mass resolving power of 20,000. High resolution mass spectra were recorded from *m/z* 50-600 in the MS or MS/MS modes. All data were recorded and analyzed with the Waters MassLynx V4.1 software package. During the analyses, the mass spectrometer was continuously calibrated by injecting the reference compound, leucine enkephalin, directly into the ESI source.

## 3 Results and discussion

### 3.1 Chemical characterization

Table 1 shows the input and steady state conditions for all stages of the chamber experiments, including the values determined for carbon yield, secondary organic carbon, and organic mass to carbon mass ratio (OM/OC). The data indicate that with increasing RH, the formation of SOC and carbon yield is reduced, both under acidic and non-acidic conditions. The results obtained are consistent with those of Zhang et al. (2011). Secondary organic aerosol formed under non-acidic conditions was additionally analyzed for OM/OC and SOA yield. The average OM/OC ratio was 1.92 ± 0.13, and the average laboratory SOA yield measured in this experiment was 0.0032 ± 0.0004. For the non-acidic experiment, the carbon yield values range from a low 0.001 (stage 5, Table 1) at the highest relative humidity to a high of 0.004 at the lowest relative humidity (stage 1, Table 1). For the acidified experiment, carbon yield declined from above 0.011 at the lowest relative humidity (8%) to 0.0013 at the highest relative humidity (44%). Although the relative humidity considered for both acidic and non-acidic experiments do not correspond precisely, an increase of SOC was observed under acidic conditions at approximately the same relative humidity. The values of SOA yields agree with previous chamber studies reported in the literature under the same nominal conditions in the presence of NOx (Edney et al., 2005; Dommen et al., 2006; Surratt et al., 2007; Zhang et al., 2011).

**Table 1.** Initial and steady state conditions, yields and OM/OC data for chamber experiments on isoprene photooxidation in the presence of acidic and non-acidic seed aerosol. The initial NOx was entirely nitric oxide. The non-acidic experiment was conducted at a low-concentration ammonium sulfate seed (~1 µg m$^{-3}$). The acidic experiment was conducted with a higher concentration of inorganic seed (~30 µg m$^{-3}$) generated from a nebulized solution for which half the sulfate mass was derived from sulfuric acid and the other half from ammonium sulfate (Lewandowski et al., 2015).

| Experiment ER662: Acidic seed aerosol (½ ammonium sulfate, ½ sulfuric acid by sulfate mass in precursor solution) | | | | |
|---|---|---|---|---|
| | Stage 1 | Stage 2 | Stage 3 | Stage 4 |
| RH (%) | 8 | 28 | 44 | 18 |
| Temperature (C) | 27.0 | 27.3 | 26.9 | 27.5 |
| Initial Isoprene (ppmC) | 6.82 | 6.92 | 7.01 | 7.03 |
| Initial NO (ppm) | 0.296 | 0.296 | 0.296 | 0.296 |
| **Steady state conditions** | | | | |
| O$_3$ (ppm) | 0.303 | 0.292 | 0.245 | 0.339 |
| NO$_x$ (ppm) | 0.220 | 0.213 | 0.205 | 0.234 |
| ΔHC (µg m$^{-3}$) | 3266 | 3318 | 3357 | 3472 |
| Carbon Yield | 0.0112 | 0.0027 | 0.0013 | 0.0051 |
| SOC (µgC m$^{-3}$) | 32.3 | 7.9 | 3.8 | 15.7 |

| Experiment ER667: Non-acidic seed aerosol (ammonium sulfate) | | | | | |
|---|---|---|---|---|---|
| | Stage 1 | Stage 2 | Stage 3 | Stage 4 | Stage 5 |
| RH (%) | 9 | 19 | 30 | 39 | 49 |
| Temperature (C) | 28.2 | 28.5 | 27.9 | 27.8 | 27.6 |
| Initial Isoprene (ppmC) | 8.11 | 8.29 | 8.25 | 8.25 | 8.19 |
| Initial NO (ppm) | 0.347 | 0.347 | 0.347 | 0.347 | 0.347 |
| **Steady state conditions** | | | | | |
| O$_3$ (ppm) | 0.331 | 0.305 | 0.329 | 0.393 | 0.281 |
| NO$_x$ (ppm) | 0.260 | 0.247 | 0.241 | 0.229 | 0.226 |
| ΔHC (µg m$^{-3}$) | 3518 | 3556 | 3558 | 3515 | 3484 |
| SOA yield | 0.007 | 0.004 | 0.002 | 0.002 | 0.001 |
| Carbon Yield | 0.0038 | 0.0022 | 0.0013 | 0.0009 | 0.0010 |
| SOC (µgC m$^{-3}$) | 13.3 | 7.7 | 4.6 | 3.2 | 3.5 |
| OM/OC | 1.96 | 2.00 | 2.02 | 2.03 | 1.59 |

The analysis of isoprene SOA from chamber experiments and field samples is based on the interpretation of mass spectra of the derivatized and underivatized isoprene SOA products by GC-MS (in EI and CI) and by LC-MS (negative ion mode with electrospray ionization), respectively. The characteristic ions for all BSTFA

derivatives are *m/z* 73, 75, 147, and 149. In CI mode, adduct ions from the derivatives included *m/z*: $M^{+\bullet}$ + 73, $M^{+\bullet}$ + 41, $M^{+\bullet}$ + 29, and $M^{+\bullet}$ + 1 while fragment ions included *m/z*: $M^{+\bullet}$ − 15, $M^{+\bullet}$ − 73, $M^{+\bullet}$ − 89, $M^{+\bullet}$ − 117, $M^{+\bullet}$ − 105, $M^{+\bullet}$ − 133, or $M^{+\bullet}$ − 207 (Jaoui et al., 2004). The LC-MS analysis, used to identify organosulfates, nitroxy- and nitrosoxy-organosulfates, are based on the deprotonated ions [M − H]⁻ and the corresponding fragmentation

pathways. Organosulfates were recognized by the loss of characteristic ions of *m/z*: 80 ($SO_3^-$), 96 ($SO_4^-$) and 97 ($HSO_4^-$); (Darer et al., 2011; Szmigielski 2016). The nitroxy-organosulfates and nitrosoxy-organosulfates were identified from additional neutral losses of *m/z* 63 ($HNO_3$) and *m/z* 47 ($HNO_2$), respectively. Table 2 presents the list of compounds tentatively identified in the present study along with proposed structures, molecular weights (MWs) and main fragmentation ions (*m/z*). Additional organic acids were tentatively identified in this study and

further work is being conducted to understand their role in isoprene SOA. At the present time, the organosulfate (MW 230), 2-methyltartaric acid organosulfate (MW 244), and 2-methyltartaric acid nitroxy-organosulfate (MW 275) appear not to have been reported before. An organosulfate with MW 230, but with a distinct structure, was recently reported in the literature from the photooxidation of 2-E-pentanal (Shalamzari et al., 2016).

**Table 2.** Products detected in SOA samples from chamber experiments using GC-MS and LC-MS.

| GC-MS | | | | |
|---|---|---|---|---|
| Chemical Formula | *m/z* BSTFA Derivative (methane-CI) | MW MW$_{BSTFA}$ (g mol$^{-1}$) | Tentative Structure* and Chemical Name | References |
| $C_5H_{10}O_2$ | 247, 231, 157, 147, 73 | 102 246 |  3-methyl-3-butene-1,2-diol ($C_5$-diol-1) | Wang et al. 2005 Surratt et al., 2006 |
| $C_5H_{10}O_3$ | 263, 247, 173, 83, 73, | 118 262 |  2-methyl-2,3-epoxy-but-1,4-diol (IEPOX-1) | Paulot et al., 2009 Surratt et al., 2010 Zhang et al., 2012 |
| $C_5H_{18}O_3$ | 263, 247, 173, 83, 73 | 118 262 |  2-methyl-3,4-epoxy-but-1,2-diol (IEPOX-2) | |

| Chemical Formula | m/z Main Ions | MW (g mol⁻¹) | Tentative Structure and Chemical Name | References |
|---|---|---|---|---|
| $C_4H_8O_4$ | 337, 321, 293, 219, 203 | 120 / 336 |  2-methylglyceric acid (2-MG) | Claeys et al., 2004a; Surratt et al., 2006; Edney et al., 2005; Szmigielski et al. 2007 |
| $C_5H_{12}O_4$ | 409, 319, 293, 219, 203 | 136 / 424 |  2-methylthreitol (2-MT) | Claeys et al., 2004a; Wang et al., 2004; Edney et al., 2005; Surratt et al., 2006; Nozière et al., 2011 |
| $C_5H_{12}O_4$ | 409, 319, 293, 219, 203 | 136 / 424 |  2-methylerythritol (2-MT) | |
| $C_8H_{14}O_7$ | 495, 321, 219, 203, 73 | 222 / 510 |  2-methylglyceric acid dimer (2-MG dimer) | Surratt et al., 2006; Szmigielski et al. 2007 |

**LC-MS**

| Chemical Formula | m/z Main Ions | MW (g mol⁻¹) | Tentative Structure and Chemical Name* | References |
|---|---|---|---|---|
| $C_5H_{10}O_6S$ | 197, 167, 97, 81 | 198 |  IEPOX-derived organosulfate | Tao et al., 2014 |
| $C_4H_8O_7S$ | 199, 119, 97, 73 | 200 |  2-methylglyceric acid organosulfate (2-MG OS) | Surratt et al., 2007a; Gomez-Gonzalez et al., 2008; Shalamzari et al., 2013; Riva et al., 2016 |
| $C_5H_8O_7S$ | 211, 193, 113, 97 | 212 |  2(3H)-furanone, dihydro-3,4-dihydroxy-3-methyl organosulfate | Surratt et al., 2008; Hettiyadura et al., 2015; Spolnik et al., 2018 |

| Formula | Fragment ions | Precursor | Structure | Reference |
|---|---|---|---|---|
| $C_5H_{10}O_7S$ | 213, 183, 153, 97 | 214 |  2,3,4-furantriol, tetrahydro-3-methyl-organosulfate | Hettiyadura et al., 2015 Spolnik et al., 2018 |
| $C_5H_{12}O_7S$ | 215, 97 | 216 |  2-methyltetrol organosulfate (2-MT OS) | Surratt et al., 2007a Gomez-Gonzalez et al., 2008 Surratt et al., 2010 |
| $C_5H_{10}O_8S$ | 229, 149, 97, 75 | 230 |  2-methylthreonic acid organosulfate | This study |
| $C_5H_9O_9S$ | 243, 163, 145, 101 | 244 |  2-methyltartaric acid organosulfate | This study |
| $C_5H_{11}NO_8S$ | 244, 226, 197, 183, 153, 97 | 245 |  2-methyltetrol nitrosoxy-organosulfate | This study |
| $C_5H_{11}NO_9S$ | 260, 197, 183, 153, 97 | 261 |  2-methyltetrol nitroxyorganosulfate | Surratt et al., 2007a Surratt et al., 2008 |
| $C_5H_9NO_{10}S$ | 274, 211, 193, 153, 97 | 275 |  2-methylthreonic acid nitroxy-organosulfate | This study |

* For more stereo-chemically complex molecules a representative isomer is shown.

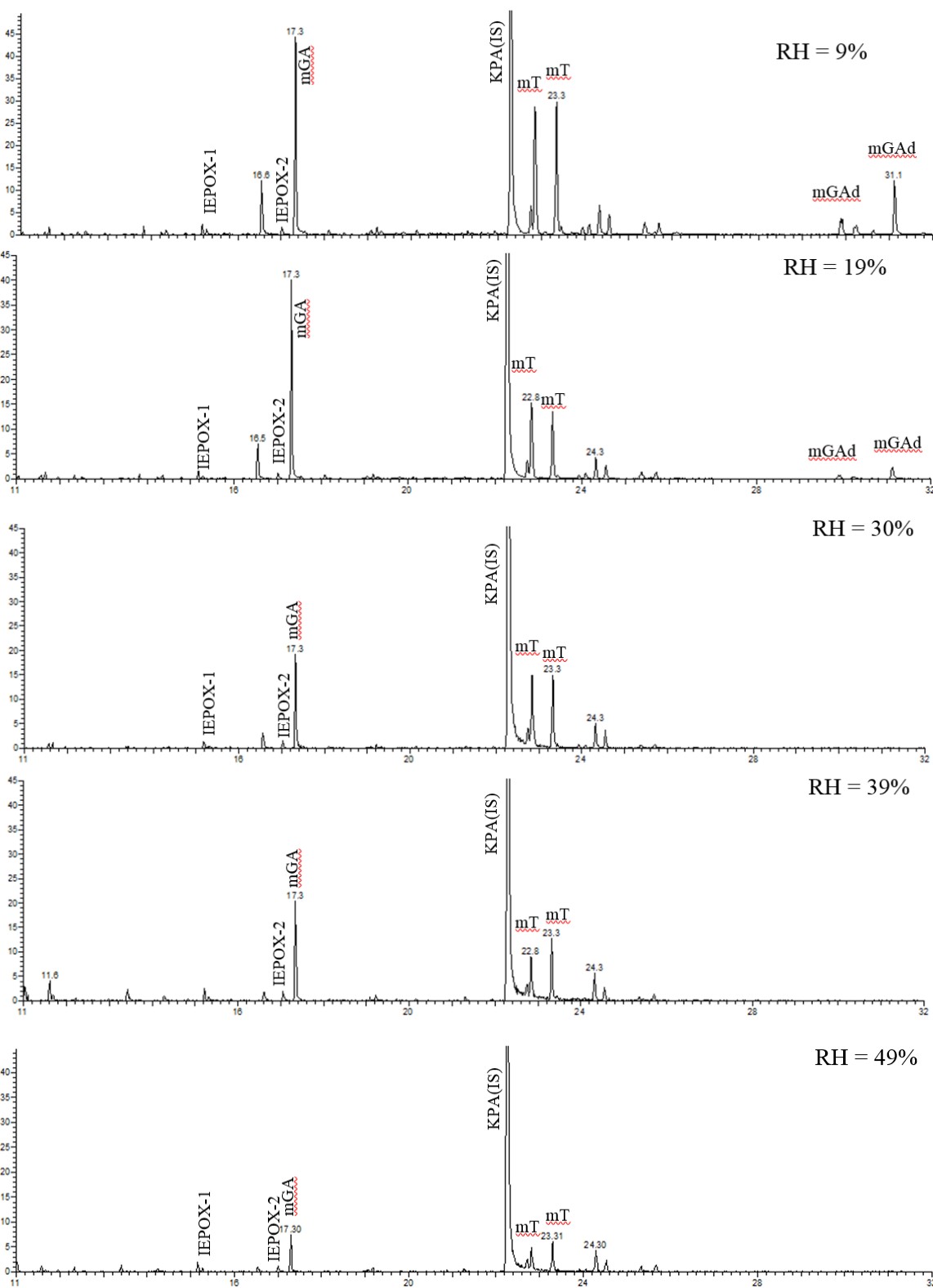

**Figure 1.** Extracted Ion Chromatograms (KPA: *m/z* 165; ketopinic acid (IS)); (IEPOX: *m/z* 173, 2 isomers), (mGA: 321; 2-methylglyceric acid), (mT: *m/z* 409; 2-methyltetrols, 4 isomers), (mGAd: *m/z* 495; 2-methylglyceric acid dimer, 3 isomers) for non-acidic isoprene/NOx photooxidation experiments as a function of RH. Compounds detected as silylated derivatives. For clarity of the figure, not all isomers are shown.

Figure 1 presents GC-MS Extracted Ion Chromatograms (EIC) from the aerosol obtained during the non-acidic experiment (isoprene non-acidic seed irradiation) at a wide range of relative humidities. According to acquired chromatograms shown in Figure 1, several isomers associated with the compounds analyzed can be distinguished, i.e. IEPOX-1 and IEPOX-2, 4 isomers of 2-methyltetrols and their relative contributions to SOA masses at various relative humidity levels.

The formation of isoprene SOA products such as 2-methyltetrols (mT) and 2-methylglyceric acid is well documented in the literature. These compounds are isoprene SOA markers and have been reported in numerous field and chamber studies under low- and high-$NO_x$ conditions (Claeys et al., 2004a; Edney et al., 2005; Kroll et al., 2006; Surratt et al., 2006, 2010). The formation mechanism under low-$NO_x$ conditions has been explained by the reactive uptake of isoprene epoxydiols (IEPOX) onto acidic aerosol seeds (Paulot et al., 2009; Surratt et al., 2010) and under high-$NO_x$ conditions by the further oxidation of methacryloyl peroxynitrate (MPAN) (Chan et al., 2010; Surratt et al., 2010; Nguyen et al., 2015).

The LC-MS analyses focused mainly on the formation of the variety of organosulfates, nitroxy- and nitrosoxy-organosulfates. Mass spectra and proposed fragmentation pathways of newly identified components are presented in section 3.4.

## 3.2 Effect of relative humidity and acidity on products formation

### 3.2.1 Non-acidic aerosol

Table 3 and Figures 2 – 3 present the estimated amounts of polar oxygenated products detected with GC-MS and LC-MS techniques in samples from non-acidic photooxidation experiments with non-acidic aerosol seeds under various RH conditions. Six products were quantified (as sums of respective isomers) based on the response factor of ketopinic acid using GS-MS. Nine other compounds were detected qualitatively using LC-MS, with chromatographic responses representing the amounts of respective analytes. Therefore, the results should be understood as a tendency of product occurrence in the chamber experiments rather than the real amounts formed. Table 3 does not contain data on 2-methyltartaric acid organosulfate (MW 244) because it occurred in the samples merely in trace amounts.

**Table 3.** Estimated concentrations of reaction products (ng m$^{-3}$) from the non-acidic photooxidation experiments (neutral seed [H$^+$] = 54 nmol m$^{-3}$ air: Lewandowski et al., 2015).

| | RH 9 (%) | RH 19 (%) | RH 30 (%) | RH 39 (%) | RH 49 (%) |
|---|---|---|---|---|---|
| **GC-MS data** * | | | | | |
| 2-methylglyceric acid | 379 | 255 | 155 | 171 | 70 |
| 2-methyltetrols | 811 | 384 | 371 | 257 | 157 |
| 2-methylglyceric acid dimer | 308 | 68 | 0 | 0 | 0 |
| IEPOX-1 | 5 | 3 | 2 | 0 | 3 |
| IEPOX-2 | 37 | 21 | 23 | 12 | 19 |
| C$_5$-diol-1 | 9 | 6 | 3 | 0 | 0 |
| **LC-MS data** ** | | | | | |
| *m/z* [M − H]$^-$ | | | | | |
| 197 | 0.28 | 0.22 | 0.19 | 0.37 | 0.33 |
| 199 | 3.22 | 2.46 | 3.60 | 4.66 | 4.01 |
| 211 | 0.44 | 0.20 | 0.06 | 0.09 | 0 |
| 213 | 2.21 | 1.87 | 1.52 | 1.48 | 0.83 |
| 215 | 17.80 | 12.30 | 10.20 | 9.83 | 7.24 |
| 229 | 0.70 | 0.78 | 1.11 | 1.29 | 0.83 |
| 244 | 0.35 | 0.14 | 0 | 0 | 0.08 |
| 260 | 0.49 | 0.35 | 0.32 | 0.28 | 0.18 |
| 274 | 0.08 | 0.10 | 0.08 | 0.08 | 0.07 |

* MW as BSTFA derivative

** chromatographic responses of organosulfates [10$^4$]

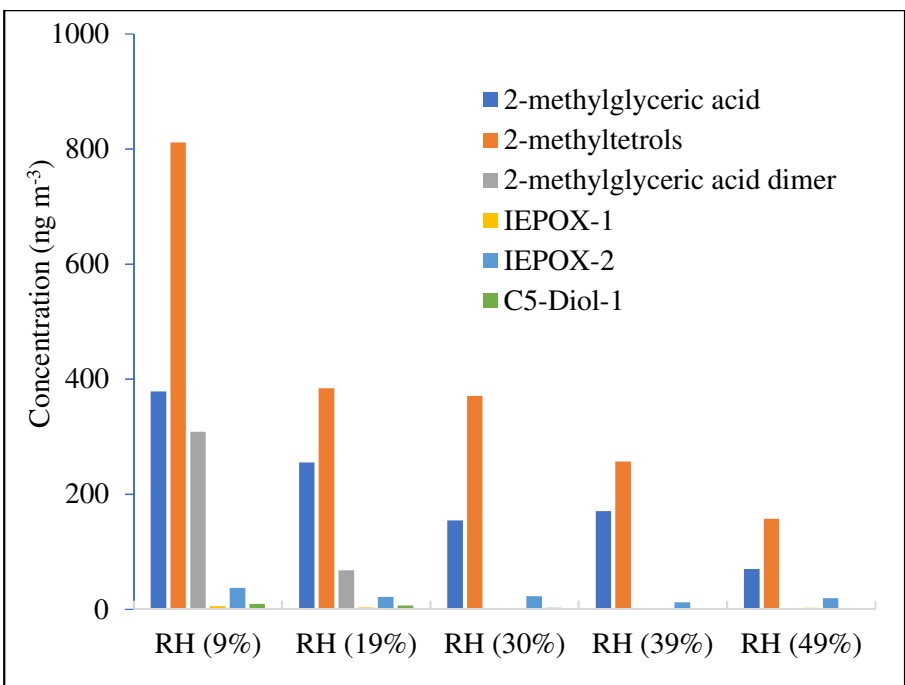

**Figure 2**. Concentrations of particle phase products from the non-acidic seed experiments (non-acidic) estimated with GC-MS.

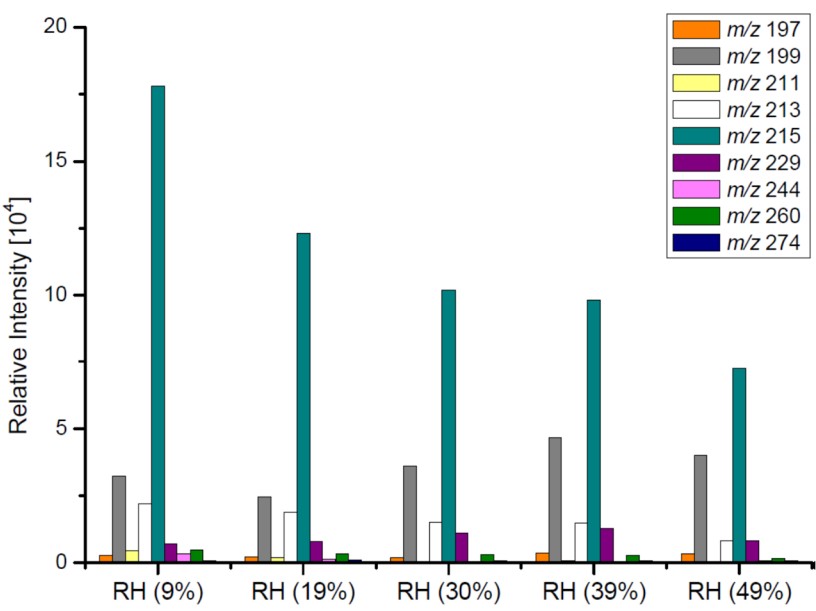

**Figure 3**. LC-MS chromatographic responses of OS and NOSs from the non-acidic seed experiments (non-acidic).

The major SOA components detected were 2-methyltetrols, 2-methylglyceric acid and its dimer, whose maximal estimated concentrations exceeded 800, 350 and 300 ng m$^{-3}$ respectively under low-humidity conditions

of RH 9% (Figure 2. At the two lowest humidities, aerosol liquid water is expected to be very low and the decrease in these compounds may not be controlled by aerosol liquid water but possibly by the SOC levels associated with the particles (Lewandowski et al., 2015), although chamber-related wall effects due to water vapor might also play some role. Among compounds detected with LC-MS (Figure 3) are organosulfates derived from acid-

catalysed multiphase chemistry of IEPOX (MW 216) and MAE/HMML (MW 200) (Surratt et al., 2010; Lin et al., 2012, 2013; Nguyen et al., 2015). Other components were significantly less abundant. In most cases, increasing the humidity resulted in decreased yields of the products detected, although some compounds were observed at higher concentrations at RH 49% compared to RH 9% (i.e. m/z 199: Figure 3). As found in Table 1, total SOC decreased with increased humidity. Generally, the influence of RH on the product yields was modest

consistent with Dommen et al. (2006) and Nguyen et al. (2011), who saw a negligible effect of relative humidity on SOA yield in photooxidation of isoprene in the absence of acidic seed aerosol. By contrast, here the 2-methyltetrols, 2-methylglyceric acid, and 2-methylglyceric acid dimer were found in significantly larger quantities at RH 9% compared to RH 49%. Two recent studies (Lin et al., 2014; Riva et al., 2016) reported an increase in aerosol mass with increasing RH. Riva et al., (2016) also reported an increase in 2-methyltetrols concentrations

with increasing RH. However, the initial conditions for those two studies differed substantially from that in the present study. Here, isoprene is oxidized in the presence of NOx and seed aerosol (acidic and non-acidic) under a wide range of RH. In contrast, in Riva et al. and Lin et al. studies, the reactants were hydroxyhydroperoxide (ISOPOOH) and IEPOX oxidized under NOx-free conditions at two levels of RH. In addition, organosulfates, 2-methyltetrols and SOA yields derived from isoprene photooxidation typically have been enhanced under acidic

conditions (Surratt et al., 2007a,b, 2010; Gomez-Gonzalez et al., 2008; Jaoui et al., 2010; Zhang et al., 2011). Organosulfates were also formed in non-acidic experiments, probably through radical-initiated reactions in wet aerosol particles containing sulfate moieties (Noziere et al., 2010; Perri et al., 2010). The NOS and OS compounds detected here could have been formed via such a mechanism.

**3.2.2 Acidic seed aerosol**

Table 4 and Figures 4 - 5 present the estimated amounts of polar oxygenated products detected using GC-MS and LC-MS techniques in samples from the acidic photooxidation experiments with acidic aerosol seed under various RH conditions. We detected the same compounds as in the non-acidic seed experiments, with the same analytical limitations of the quantitation. The presence of 2-methyltetrols and 2-methylglyceric acid and

their sulfated analogues in isoprene SOA at a wide range of RH conditions, suggests that SOA water content does not significantly affect their formation.

**Table 4.** Estimated concentrations of reaction products (ng m$^{-3}$) from the acidic photooxidation experiments (acidic seed [H$^+$] = 275 nmol m$^{-3}$ air: Lewandowski et al., 2015).

| | RH 8 (%) | RH 18 (%) | RH 28 (%) | RH 44 (%) |
|---|---|---|---|---|

| GC-MS data * | | | | |
|---|---|---|---|---|
| 2-methylglyceric acid | 3070 | 2136 | 982 | 473 |
| 2-methyltetrols | 5357 | 4767 | 1029 | 341 |
| 2-methylglyceric acid dimer | 90 | 144 | 102 | 43 |
| IEPOX-1 | 1 | 13 | 6 | 0 |
| IEPOX-2 | 10 | 3 | 0 | 0 |
| $C_5$-diol-1 | 53 | 0 | 0 | 0 |
| LC-MS data ** | | | | |
| $m/z$ [M – H]$^-$ | | | | |
| 197 | 0.88 | 0.30 | 0.21 | 0.10 |
| 199 | 3.44 | 1.49 | 2.62 | 1.12 |
| 211 | 1.78 | 0.50 | 0.76 | 0.48 |
| 213 | 5.41 | 1.94 | 3.40 | 1.96 |
| 215 | 59.00 | 18.40 | 12.30 | 3.23 |
| 229 | 0.41 | 0.31 | 0.39 | 0.27 |
| 244 | 4.50 | 1.16 | 0.72 | 0.42 |
| 260 | 0.92 | 0.88 | 0.45 | 0.29 |
| 274 | 0.60 | 0.58 | 0.36 | 0.12 |

\* MW as BSTFA derivative

\*\* chromatographic responses of selected main organosulfates [$10^4$]

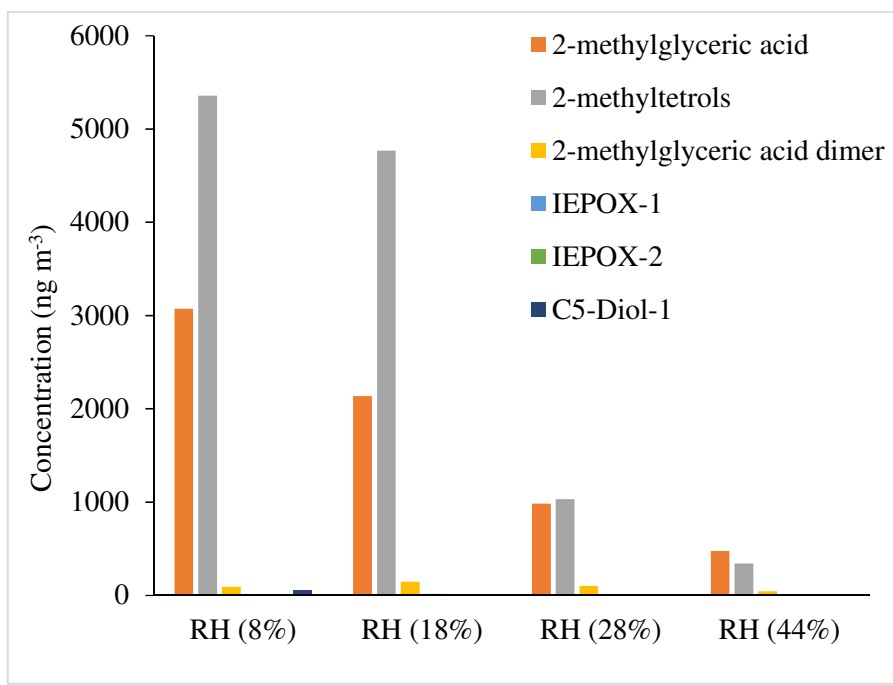

**Figure 4.** Concentrations of particle phase products from the acidic seed experiments estimated with GC-MS.

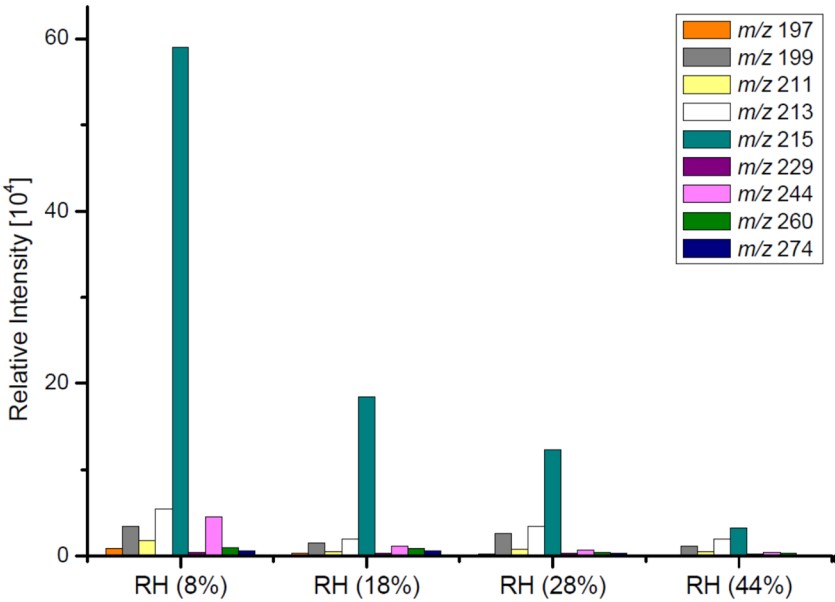

**Figure 5**. LC-MS chromatographic responses of OS and NOS products from the acidic seed experiments.

Early chamber studies on isoprene ozonolysis by Jang et al. (2002) and Czoschke et al. (2003) showed enhanced SOA yields in the presence of acidified aerosol seeds. Recent laboratory results showed that the acidity of aerosol seeds plays a major role in the reactive uptake of isoprene oxidation products by particle phases (Paulot et al., 2009; Surratt et al., 2010; Lin et al., 2012; Gaston et al., 2014a,b; Riedel et al., 2015). In our study, SOC

produced in acidic-seed experiments was always higher than in non-acidic seed ones under the corresponding RH conditions, while the difference diminished with increasing RH to a negligible value of 0.3 µg C m$^{-3}$ at RH 44 – 49% (Table 1 and Figure S1, Supplementary Information; Surratt et al., 2007a). However, the formation of the individual organic compounds did not follow the same pattern. As an example, Figure 6 shows a comparison of the concentrations of 2-methylglyceric acid under acidic and non-acidic condition as a function of relative

humidity. Acidic seed aerosol has a greater effect on 2-methylglyceric acid at lower relative humidity. Some of the compounds produced in higher quantities in the acidic seed experiments included 2-methylglyceric acid, 2-methyltetrols, furanetriol-OS, 2-methyltetrol-NOS, 2-methylthreonic acid NOS, furanone-OS, while some other in the non-acidic seed experiments including IEPOX-2, 2-methylglyceric acid OS, 2-methylthreonic acid OS. Yields of the remaining compounds followed an inconclusive pattern (SI: Figures S1, S2, and S3; Table S1). Thus,

this study shows the effect of relative humidity on the formation of a wide range of isoprene SOA products cannot easily be predicted, although the majority increases with decreasing relative humidity both under acidic and non-acidic conditions.

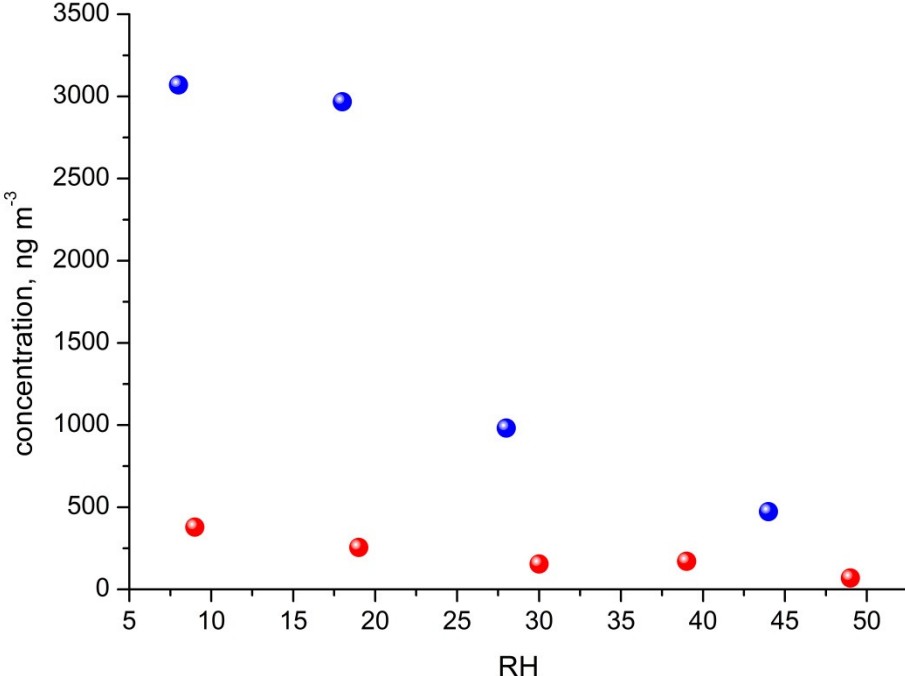

**Figure 6**. Influence of RH and seed acidity on the estimated concentration of 2-methylglyceric acid produced in chamber experiments with non-acidic seeds (red) and with acidic seeds (blue). See Figure S3 for additional compounds.

### 3.3 Chromatographic comparison of chamber experiments and field samples

We compared the results of chamber experiments to samples of $PM_{2.5}$ collected at the two rural sites, Zielonka and Godow. To keep the experimental and ambient conditions as similar as possible, we selected the experiments carried under the highest relative humidities: ER662 at RH 44% (acidic seeds) and ER667 at RH 49% (non-acidic

seeds). Figures 7–10 show the extracted ion chromatograms of selected components detected in the respective filter extracts. Several compounds occurred both in the chamber SOA and in the ambient samples: 2-methylglyceric acid OS (MW 200), furanetriol OS (MW 214), 2-methyltetrol OS (MW 216), 2-methylthreonic acid OS (MW 230), 2-methylthreonic acid NOS (MW 275). The 2-methyltartaric acid OS (MW 244) was also found in ambient samples with only trace amounts in acidic seed aerosol (Figure 9). However, 2-methyltetrol

nitrosoxy-organosulfate (MW 245) was detected in the chamber SOA (Figure 10). The extracted ion chromatograms of 2-methyltetrol nitroxy-organosulfate (MW 261) were insufficient to provide reasonable fragmentation (Figure S4). The comparison shows that isoprene SOA in the presence of acidic seed aerosol and $NO_x$ from the chamber studies provide a reasonable approximation to the ambient processes at both sites even though only Godow is strongly influenced by anthropogenic pollutants, mainly nitrogen oxides due to a nearby

coal-fired power station. It appears that minor amounts of $NO_X$ in the ambient atmosphere are sufficient to produce these compounds. These findings will require further confirmation.

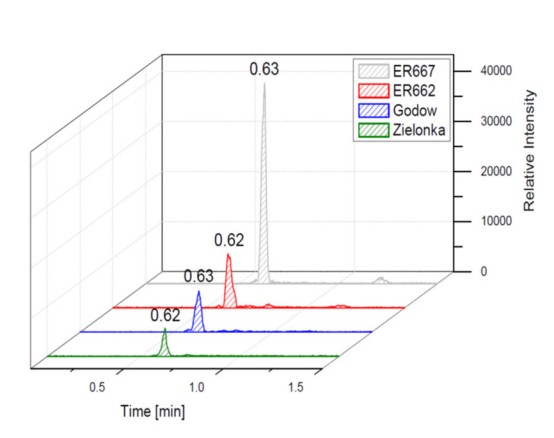

**Figure 7**. Extracted Ion Chromatograms of 2-methylglyceric acid organosulfate with MW 200 from field studies and chamber experiments.

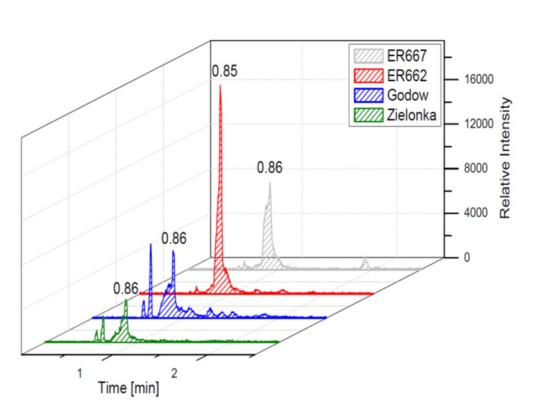

**Figure 8**. Extracted Ion Chromatograms of furanetriol organosulfate with MW 214 from field studies and chamber experiments.

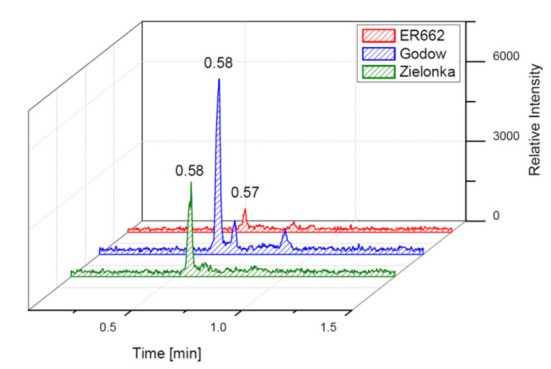

**Figure 9**. Extracted Ion Chromatograms of 2-methyltartaric acid organosulfate with MW 244 from field studies and chamber experiments (not detected in non-acidic sample).

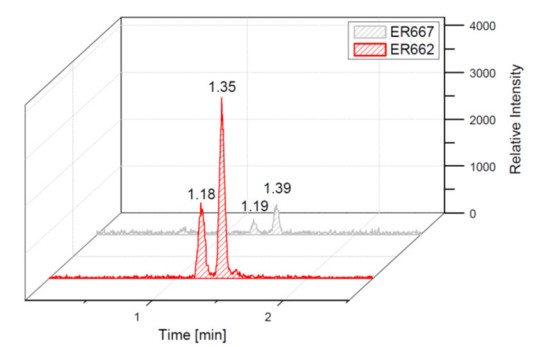

**Figure 10**. Extracted Ion Chromatograms (EIC) of nitrosoxy-organosulfate with MW 245 from chamber experiments (not detected in field samples).

**3. 4 Mass spectra and proposed fragmentation pathways of newly identified organosulfates, nitroxy- and nitrosoxy-organosulfates**

Based on the high-resolution mass data and fragmentation spectra recorded for HPLC-resolved peaks, it is difficult to distinguish between isomers of the same molecular structure. Moreover, some of the peaks for selected *m/z* values in the extracted ion chromatograms may correspond to more than one compound. Therefore, identifications for the structures proposed are tentative. This ambiguity results in the fragmentation spectra having the fragment ions coming from different

precursor ions with the same *m/z*. Our proposed structures for the newly identified organosulfates, nitroxy- and nitrosoxy-organosulfates are based on the accurate mass measurements and the following assumptions:

a) all studied compounds have the same carbon backbone of 2-methylbutane;

b) the presence of the abundant *m/z* 97 peak corresponding to the $HSO_4^-$ ion indicates that the hydrogen atom is present at the carbon atom next to that bearing $HO-SO_2-O-$ moiety (Attygalle et al., 2001). There are, however, exceptions seen in Figures 11 and 12;

c) when the condition given in (b) is not fulfilled, elimination of sulfur trioxide molecule from the precursor ion can be detected (Szmigielski, 2013);

d) elimination of the HONO and $HNO_3$ molecules from the precursor ion is a diagnostic for the presence of the nitrous (-ONO) and nitric ($-ONO_2$) esters, respectively. Similar to assumption (a), a β-hydrogen must be present to enable the β-elimination (Tovstiga et al., 2014).

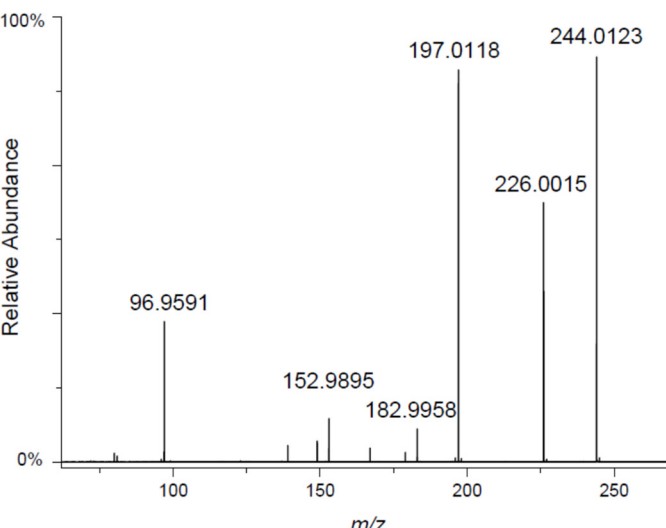

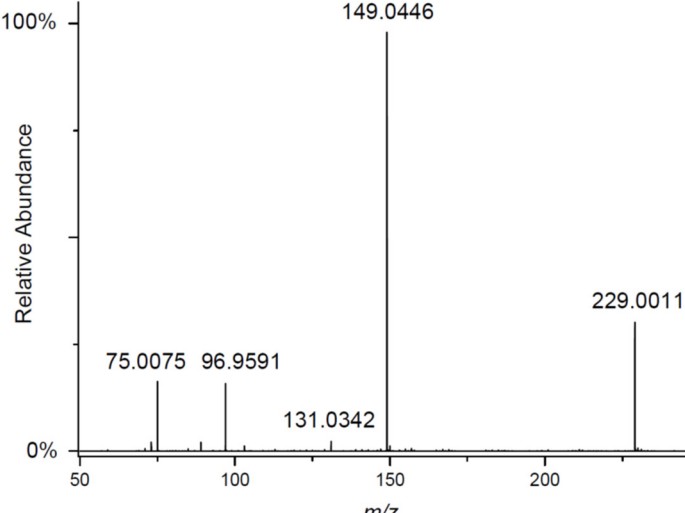

**Figure 11**. (-)Electrospray product ion mass spectrum of 2-methyltetrol nitrosoxy-organosulfate (MW 245) of the RT = 1.35 min peak (Figure 10) acquired for the acidic seed aerosol along with the proposed fragmentation pathway.

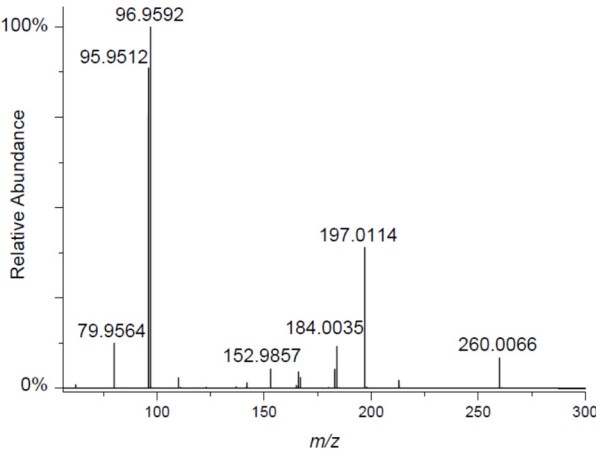

**Figure 12**. (-)Electrospray product ion mass spectrum of 2-methylthreonic acid organosulfate (MW 230) at RT = 0.63 min. (Figure S4) acquired for Zielonka $PM_{2.5}$ aerosol along with the proposed fragmentation pathway.

The 2-methyltetrol nitroxy-organosulfate detected at $m/z$ 260 corresponds to the major early eluting compounds for the chamber and $PM_{2.5}$ as seen in Figure S4. The minor shifts in retention times of eluting compounds are generally due to matrix effects (Spolnik et al., 2018). Two partially resolved peaks with identical MS profiles typically indicate diastereoisomeric forms. This finding is consistent with earlier studies (Gomez-Gonzalez et al., 2008; Surratt et al., 2007a). A detailed

10    interpretation of negative ion electrospray mass spectra led to a proposed structure for 2-methyltetrol nitroxy-organosulfates bearing a nitroxy moiety at the primary hydroxyl group of 2-methyltetrol skeleton and sulfate group at the secondary hydroxyl group seen in Figure 13. The main fragmentation pathways correspond to a neutral loss of 63 u. ($HNO_3$) resulting in $m/z$ 197 as a base peak and to bisulfate ion at $m/z$ 97. Another diagnostic ion at $m/z$ 184 can be attributed to a combined loss of $NO_2$ and $CH_2O$, suggesting the presence of hydroxymethyl group in the molecule. The presence of $m/z$ 213 and 183 ions supports

15    the interpretation given above due to a characteristic neutral loss of a $CH_2O$ fragment. A revised structure for the MW 261 SOA component along with the proposed fragmentation scheme is given in Figure 13, where only the mass spectrum of one diastereoisomer is shown.

**Figure 13**. (-)Electrospray product ion mass spectrum of 2-methyltetrol nitroxy-organosulfate (MW 261) eluting at RT = 2.44 min. (Figure S4) registered for the acidic seed aerosol along with proposed fragmentation pathway.

A second abundant chamber-generated SOA component was detected at $m/z$ 244. In contrast to 2-methyltetrol nitroxy-organosulfate, the MW 245 unknown was not detected in $PM_{2.5}$ which would suggest the compound could play a relevant role as a reactive reaction intermediate in route to particle formation through isoprene-SOA chains. Two base line-resolved peaks of identical electrospray product ion mass spectra could be attributed to diastereoisomers with an isoprene-retained backbone (Figures 10). Surratt and co-workers observed the formation of this compound in the isoprene photooxidation experiment under high-$NO_x$ conditions and proposed the structure to 2-methylglyceric acid nitroxy-organosulfate (Surratt et al., 2007a). However, in the light of our mass spectral data we assign the MW 245 unknown to $C_5$ organosulfate, namely 2-methyltetrol nitrosoxy-organosulfates. The $m/z$ 244 → $m/z$ 226 transition in the product ion mass spectrum (Figure 11) points to the intact secondary hydroxyl moiety of the 2-methyltetrol skeleton. The lack of $HNO_3$ elimination from [M – H]$^-$ ($m/z$ 244) precursor ion clearly excludes the presence of nitroxy group. However, an abundant $m/z$ 197 ion, which forms through the $HNO_2$ loss, could be associated with the existence of the -O-NO residue. The structure assigned to the abundant MW 245 component from ER662 (acidic seed aerosol) along with its proposed fragmentation scheme is presented in Figure 11.

Additional abundant SOA organosulfates were determined at $m/z$ 229 and 243 for the chamber and $PM_{2.5}$ as shown in Figures 12 and 14, respectively, which does not appear to have previously been detected. The accurate mass data was recorded for the Godow sample with the following characteristics: RT = 0.58 min in Figure 9, $C_5H_7O_9S$: 242.9816 Da, error + 0.2 mDa (Figure 14) and RT = 0.63 min in Figure S4, $C_5H_9NO_8S$: 229.0011 Da, error +0.2 mDa (Figure 12) suggested greater oxidation pathways for these unknown organosulfates compared that for the formation the of sulfated-2-methyltetrols. Two partially resolved peaks of identical mass spectrometric signatures can be noted for these organosulfates indicating the presence of two chiral centres in their molecules (Figure 9 and S4). In either case, first eluting diastereoisomers give rise to peaks having

high abundances, while the second peak is of a more minor intensity suggesting the formation of less hindered compounds both in the chamber experiments and PM$_{2.5}$. A detailed interpretation of product ion mass spectra permitted assignment of structures of the MW 244 and MW 230 unknowns to 2-methyltartaric acid organosulfate and 2-methylthreonic acid organosulfate, respectively (Figures 14 and 12 with the mass spectrum of the minor diastereoisomer not shown). Either

5 spectrum displays abundant fragment ions at $m/z$ 163 and 149, respectively, which could be explained by the SO$_3$ elimination from their precursor ions. Further fragmentations of $m/z$ 163 ions, i.e., a neutral loss of water followed by decarboxylation, reveals the simultaneous presence of -O-SO$_3$H and –CO$_2$H residues in the MW 230 diastereoisomeric organosulfates. However, the absence of the bisulfate ion in the spectrum of the MW 244 organosulfate clearly indicates a lack of a proton adjacent to the sulfated group, and thus suggests the sulfation of a secondary hydroxyl group. MW 230 organosulfate and the

10 presence of the bisulfate ion in the MS/MS spectrum does not necessarily reveal unambiguously the sulfation at a primary hydroxyl group in the molecule. The proposed fragmentation schemes for the MW 244 and 230 novel organosulfates are depicted in Figures 14 and 12. Again, the mass spectra of related diastereoisomeric organosulfates are not presented.

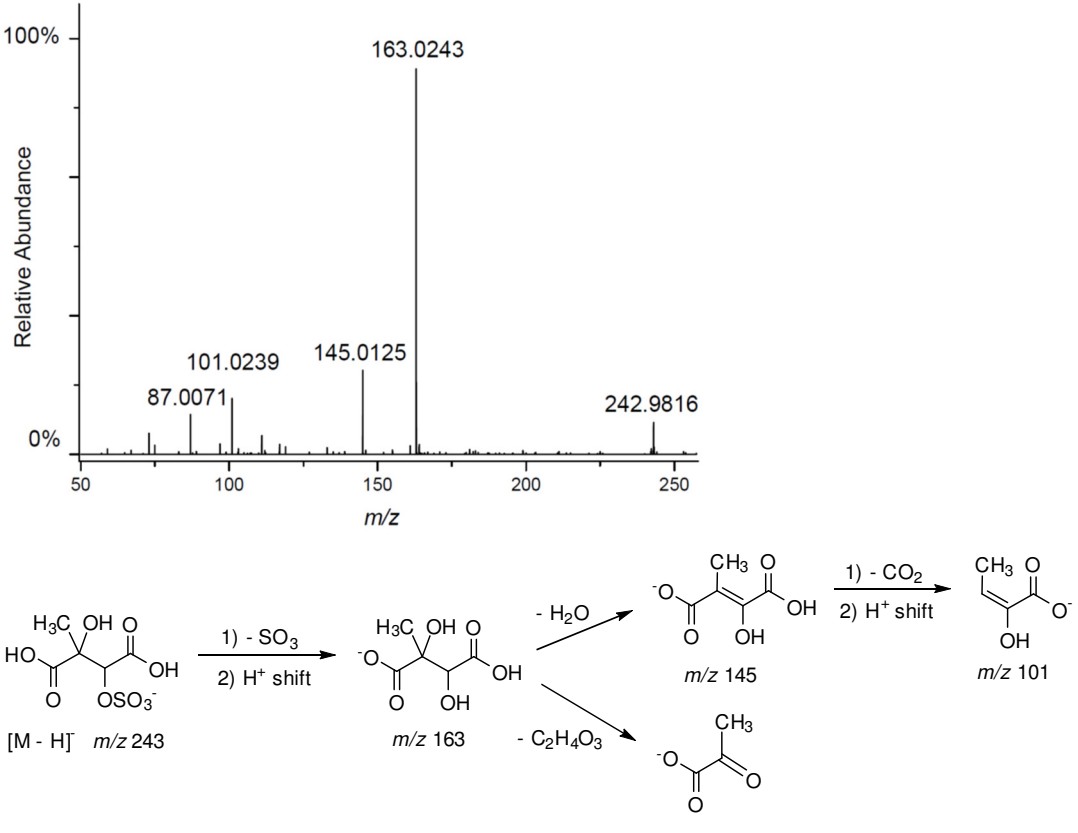

**Figure 14**. (-)Electrospray product ion mass spectrum of 2-methyltartaric acid organosulfate (MW 244) recorded for the RT = 0.58 min peak (Figure 9) from Godow fine aerosol along with the proposed fragmentation pathway.

A final related organosulfate was detected at *m/z* 274 in substantial quantities for isoprene SOA from the chamber and rural PM$_{2.5}$ (Figure S4). To our knowledge this compound has previously not been reported. The compound has transitions of *m/z* 274 → *m/z* 211 (a loss of HNO$_3$) and *m/z* 274 → *m/z* 97 (a loss of C$_5$H$_7$NO$_6$) from the product ion mass spectrum from the Zielonka PM$_{2.5}$ as seen in Figure 15. The high-resolution data for this organosulfate renders the following characteristics, RT = 0.83 min., C$_5$H$_7$NO$_{10}$S: 273.9873 Da, error +0.4 mDa, which clearly points to nitroxy-organosulfate from isoprene. A detailed explanation of other diagnostic ions led to a proposed structure of 2-methylthreonic acid nitroxy-organosulfate (Figure 15). It could be assumed that due to a high oxidation state (C/O = 0.5) the MW 275 organosulfate could serves as an identifying marker of highly processed isoprene aerosol. However, the further study is warranted to rationalize its formation mechanism and reactivity in the atmosphere.

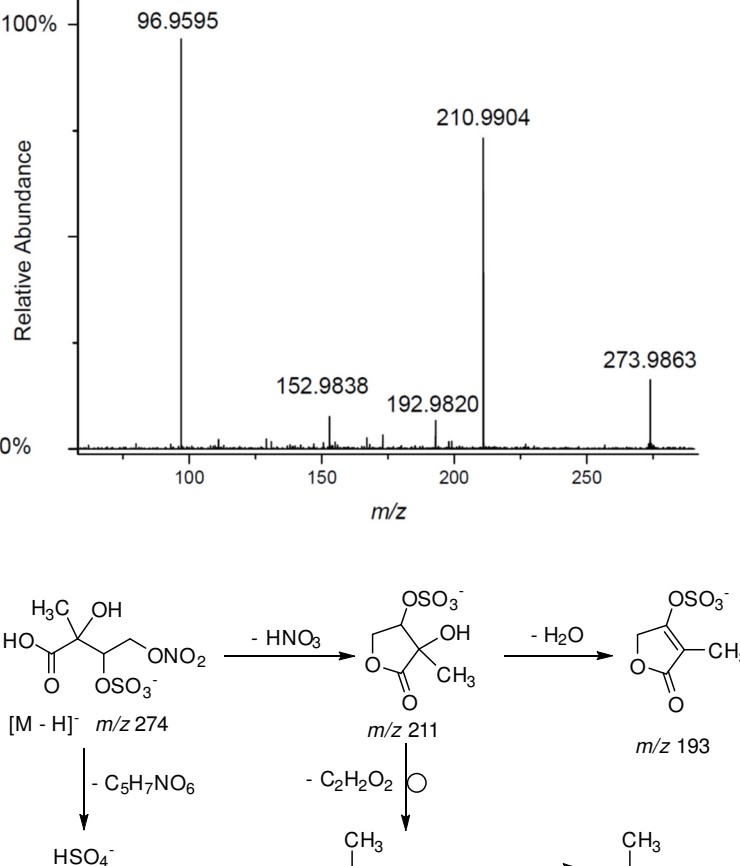

**Figure 15**. (-)Electrospray product ion mass spectrum of 2-methylthreonic acid nitroxy-organosulfate (MW 275) of the RT = 0.83 min peak (Figure S4) recorded for Zielonka PM$_{2.5}$ aerosol along with the proposed fragmentation pathway.

While these experiments provide an analysis of a wide range of isoprene reaction products in the aerosol phase as a function of RH and acidity, they also include a number of shortcomings that need to be addressed in future work. Perhaps the most significant is the use of authentic standards to assess the contribution of these products to SOA mass at different RH. In addition, when the relative humidity is varied, it is important to measure aerosol liquid water content directly or estimated using thermodynamic models, such as ISORROPIA (Fountoukis and Nenes, 2007) or AIM (Wexler and Cregg, 2002), and other gas and particle composition (e.g. inorganic species). Liquid water inorganic species measurements were not available for this study.

The use of these marker compounds for ambient air quality models can follow the approach of Pye et al. (2013). In such an approach, the model is run using a base case chemical mechanism for isoprene, where there is no adjustment for acidity and relative humidity. A comparison can then be made with the same model having such an adjustment incorporated within the isoprene mechanism. The markers can then serve as constraints to the PM observations. For the U.S. the Community Multiscale Air Quality (CMAQ) model is frequently used for ozone and PM ambient concentrations (Pye et al., 2013). For Poland, a similar approach can be used with a European model having the appropriate meteorology and chemical mechanism (Miranda et al., 2015).

## 4. Summary

In this work, we have characterized several organic components from isoprene SOA, some of which have been reported in the literature. Several compounds were identified for the first time, including 2-methylthreonic acid organosulfate (MW 230), 2-methyltartaric acid organosulfate (MW 244) and 2-methyltartaric acid nitroxy-organosulfate (MW 275). The quantitative data showed that the 2-methyltetrols, 2-methylglyceric acid and 2-methyltetrol organosulfates as the most abundant components of isoprene SOA. Other molecular components contributing to SOA mass were epoxydiols, mono- and dicarboxylic acids, organosulfates as well as nitroxy- and nitrosoxy-organosulfates. Several organosulfates and nitroxy-organosulfates identified in chamber samples were also detected in samples of ambient aerosol collected at rural sites in Poland. Such consistency reinforces the relevance of the chamber findings although 2-methyltetrol nitrosoxy-organosulfate (MW 245) was found only in chamber experiments.

The effect of relative humidity on SOA formation was minor in the non-acidic seed experiments, and strong under acidic seed aerosol. Total SOC decreased with increasing relative humidity but the individual components were influenced diversely. The yields of most compounds decreased, but increased levels of IEPOX-OS, 2-methylglyceric acid OS and 2-methylthreonic acid OS were produced at medium to high relative humidity values. The acidic seed experiments enhanced SOC production more than the non-acidic conditions under all RH conditions. However, at high humidity (44–49%), the difference was relatively small. Some of the individual SOA components followed the same pattern as the SOC while others

were more abundant in non-acidic experiments or behaved in inconsistent manner. Further research is warranted to rationalize the mechanisms of their formation in the atmosphere.

**Author contributions.** MJ, ML, KN, and RS designed the study; KN, GS, MJ, and ML conducted experiments and analyzed the samples; KN, GS, WD, KR, MJ, RS, and ML analyzed the data and created figures and tables; all authors interpreted data and provided guidance for writing the paper; KN, RS, MJ, and TK wrote the paper.

**Disclaimer.** The views expressed in this journal article are those of the author(s) and do not necessarily represent the views or policies of the U.S. Environmental Protection Agency. Mention of trade names or commercial products does not constitute endorsement or recommendation for use. The work of Polish researchers was partially supported by funds from National Science Centre, Poland (Grant Nr OPUS8-2014/15/B/ST10/04276). The authors would like to thank Mr. Grzegorz Spolnik for his technical assistance during LC/MS measurements and Mr. Krzysztof Skotak for his assistance in field campaigns.

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
