# Peer review of "Chemical composition of isoprene SOA under acidic and non-acidic conditions: Effect of relative humidity"

_Atmospheric Chemistry and Physics, 2018_

## Referee Comment (RC1) · Anonymous Referee #1 · 2 May 2018

This paper reports the formation of organosulfates and other oxygenated compounds in secondary organic aerosols (SOA) generated in an indoor smog chamber from isoprene oxidation in the presence of seed aerosols. For several OSs, chemical structures are proposed on the basis of high-resolution tandem mass spectra. Isoprene accounts for a large part of VOCs in the global atmosphere, and its photooxidation has been found to contribute to SOA formation. However, relatively few studies have been focused on the oxidation of isoprene under various relative humidities (RH), and there is still uncertainty on its chemical mechanisms and contribution to SOA formation. While this study might provide valuable information for a better understanding of the chemical pathways from the photooxidation of isoprene, the results presented here are not

sufficiently supported by the analytical method and/or do not present a real novelty. In addition, the authors should have a closer look at the literature since some of their results (e.g. 2-methyltetrols) are not consistent with the existing literature. Therefore, additional information/references and major revisions would have to be provided in order to consider this article for publication.

General comments: Page 2, lines 8-10: Too simplified, as written gas-phase oxidation of isoprene leads directly to the formation of isoprene-derived SOA products such as 2-methyltetrols. Please detail.

Page 3, line 5. Why did the authors use Kleindienst et al. 2007 as a reference to explain the formation of organosulfate?

Page 3, lines 13-22. Other groups have investigated the impact of RH on the SOA formation from isoprene oxidation: e.g. Abbatt's (isoprene + OH at different RH) & Surratt's groups (reactive uptake of IEPOX under different RH, acidity,...; isoprene + O3 at different RH and seed).

Page 3, lines 23-24. Couvidat et al. did not incorporate heterogeneous chemistry (i.e. reactive uptake) or the impact of acidity in their model. Pye et al. (2013) and Marais et al. (2016) demonstrate that replacing a reversible partitioning approach with reactive uptake to aqueous aerosol improves agreement with observations. Please revise.

Page 4, lines 18-19. "moderately acidic sulfate aerosol" does not have a real scientific meaning. Please determine the aerosol acidity and liquid water content using thermodynamically model.

Page 4, lines 30-31. Please add additional information. What was the sampling time for the filters? How much mass was collected?

Page 5, lines 17-19: how many filters were analyzed? It is important to know if the tracers were identified in 1, 10 or 100 samples. Please provide some statistical information.

Page 6. Line 3-19: Additional information/explanation are needed in the analytical protocol to validate the results: - Temperature and pressure in the rotavapor? Did the authors evaluate the losses of the most volatile compounds –> tetrols/IEPOX/...? - The authors mentioned that internal standards were used. What is the extraction efficiency/recovery? Why did the author realize this step for the GC-MS analysis only? Extraction/recovery efficiencies have to be provided for both methods.

Page 6, line 27. The authors report a SOA yield of 0.32% and conclude: "The values of SOA yields agree with previous smog chamber studies". It is not exact and some studies have reported SOA yields 10 times higher (Carlton et al., 2009 ACP). The authors should discuss and compare their results in more details. In addition, to really compare apple to apple the authors should discuss the impact of NO/VOC ratio, which can greatly impact isoprene SOA formation (Xu et al., 2014).

Page 7, Table 1. The initial concentration of NO is higher than the steady-state concentration of NOx (= NO + NO2). Why? What was the NO2 concentration? The authors should also discuss the different regimes NO/NO2 (–> impact SOA yields).

Page 8, lines 1-2. Why did the authors look only at the organosulfates? The authors should compare the quantification of the acids from LC/ESI(-)-MS vs GC-MS and polyol LC/ESI(+)-MS vs GC-MS. Indeed, it is now recognized that all thermal analyses (i.e. GC-MS, FIGAERO, SV-TAG) lead to a subsequent fragmentation of the oligomers. Isoprene-derived SOA is assumed to be mainly made out of oligomers especially under acidic conditions. The authors have the information/analytical tools to provide more insights on this topic.

Page 8, lines 3-4. Please add references

Page 8, lines 8-9. Organosulfate at m/z 230 was previously identified from the oxidation of unsaturated aldehydes.

Page 8, Table 2. Please verify the structures/formulae and specify the compounds

already identified in isoprene-derived SOA.

Page 11, lines 13-14. It is confusing. They are particle phase products. As written it can be understood that 2MT and 2MG are formed in the gas phase as secondary products. In addition, 2-MT should be a tertiary product not a secondary. ISO->ISOPOOH-> IEPOX—> 2-MT

Page 12, lines 1-17. Need to use it to compare quantification. While the authors mentioned that the detailed analysis was performed using LC-MS in positive and negative modes, only the negative mode is presented in this study, why?

The authors could have used the benefit of deploying two complementary techniques by comparing the concentrations obtained from both methods. But instead, they are giving qualitative data and do not seem to be eager to tackle the "analytical limitations" (page 15, line 6). The authors need to revise their analytical methods and use the full potential of the methods used in this study: - Comparison LC/ESI(-)-MS vs GC need to be proposed for the acids using similar standards. - Comparison LC/ESI(+)-MS vs GC need to be proposed for the polyols using similar standards. - Additional standards commercially available (e.g. erythritol, organosulfate...) can be used. Indeed the authors used only one acid to quantify a wide range of compounds. What can be the impact? - Which fraction of the SOA can be explained by the compounds identified in the different experiments?

What does the relative abundance mean? Is it normalized by the volume of air collected? The authors mentioned that they did not quantify the organosulfates but in the SI the concentrations are reported...Please explain

Finally, the results presented are not well constrained. Therefore conclusions proposed based on the concentrations appear speculative: "The major SOA components detected were 2-methyltetrols, 2-methylgliceric acid and its dimer, whose maximal concentrations exceeded 800, 350 and 300 ng/m3 respectively under low-humid conditions of RH 5 9% (Fig. 2)."

"Among compounds detected with LC-MS (Fig. 3), the most abundant were organosul-
fates derived from 2-methyltetrols (MW 216) and 2-methylglyceric acid (MW 200)."

Page 14, lines 10-12. The authors found that concentration of 2-MT increase at lower
RH. It is not consistent with previous works (Lin et al., 2014 ES&T; Riva et al., 2016
ES&T). Please explain.

Page 15, Table 4. Concentration of IEPOX-1 is much higher under certain conditions.
Please comment? Variability of the measurements?

Page 17, lines 10-14. What is new here? It has already been reported that acidity
enhances the formation of isoprene-SOA components such as tetrols, organosulfates.
Please add the references and further highlight the novelty.

Page 22, lines 13-18: How would this product be formed? Which kind of chemistry?

---

## Referee Comment (RC2) · Anonymous Referee #2 · 17 May 2018

The manuscript presents interesting new work on elucidation of isoprene SOA formation and the influence of aerosol acidity and relative humidity. The results are generally interesting, but the presentation needs considerable improvement before publication can be considered, in order to provide a less fragmented paper.

The structure of the manuscript could be improved by moving the detailed characterisation (3.3) to an earlier part of results and discussion, and then end with a general discussion of the findings in relation to current literature.

In general the use of English language should be improved. It is not the task of the reviewer to do this, and the authors should carefully read the manuscript to improve

this.

Please define abbreviations the first time they appear, also in the abstract.

Abstract: I suggest adding some concluding remarks at the end of the abstract.

Introduction: The use of references needs significant improvement. References are missing for several statements (e.g. Page 2 line 4 "Isoprene is the most abundant non-methane hydrocarbon..."). There is no need to introduce an abbreviation in the text for isoprene. There are already plenty of abbreviations in the manuscript, and this one only makes the text more difficult to read. Furthermore it is used inconsistently.

Page 2 Line 10: Please add relevant references. Page 2 line 26: It would be relevant to refer to the following studies: Riva et al., Environ. Sci. Technol., 2016, 50 (11), pp 5580–5588 Zhang et al., Environ. Sci. Technol. Lett., 2018, 5 (3), pp 167–174

Page 2 Line 29-31. This sentence is hard to understand and need references.

Page 3 Line 26: Define SOC. I think "e.g." should maybe be "i.e.". P3. Line 30-31: Other research groups were the first to develop analysis of organosulfates using LC/MS. I suggest to remove "developed in our laboratories" from the sentence.

Section 2.2. Please add information on sampling time and tree species in the area. In several instances "emission" should be replaced by "concentration".

P. 6 Line 7: Define the abbreviation.

P. 7 Line 7- page 8 line 5: This should be moved to the experimental section.

Table 2 needs references to studies where these compounds were first identified.

Figure 1: Why are all these chromatograms shown, when they are not discussed in detailed in the text? I suggest to reduce the figure to one or two chromatograms - if they are discussed.

P11.L10: What do you mean by "attained"?

Table 3: Add percentage for RH (RH9 -> RH9%).

Data in Table 4 and Figures 4-5 should be presented and discussed in more detail.

Page 17 Line 9: Please write this as a complete sentence.

Figures 7-14. Some of these should be moved to Supplementary. Instead of experiment number it would be more useful to the reader to list whether an experiment was non-acidic or acidic.

Page 21: please add figure number to the mass spectrum.

Conclusion: How much do the quantified compounds make up of the total SOA mass? It is important to keep this in mind, when discussing the effects of acidity and RH.

---

## Referee Comment (RC3) · Anonymous Referee #3 · 2 Jun 2018

Overall Comment and Recommendation:

This manuscript measures the chemical composition changes (as well as bulk SOA yields) of isoprene SOA produced under acidic and non-acidic conditions as a function of relative humidity. The kind of results presented here could certainly be of value to the atmospheric and aerosol research communities. However, as I will stress in some of my major comments below, I think one thing that is missing is a stronger connection to current models that explicitly predict isoprene SOA formation through acid-catalyzed multiphase chemical processes. Of these processes, the acid-catalyzed multiphase chemistry of IEPOX on acidic (and wet) sulfate aerosol has been shown to be one

of the dominant sources of isoprene SOA in atmospheric PM samples (e.g. Claeys et al., 2004, Science; Lin et al., 2013, ACP; Budisulistiorini et al., 2015, ACP, etc. etc.). The acid-catalyzed multiphase chemistry of high-NOx SOA precursors, such as HMML/MAE, have been shown to yield very little SOA in atmospheric PM samples (e.g., Lin et al., 2013, ACP; Budisulistiorini et al., 2015, ACP; Rattanavaraha et al., 2016, ACP). Since IEPOX has been shown to be so important to forming SOA in atmospheric PM samples, recent work has been really aimed at measuring reactive uptake (or mulitphase chemical) kinetics of IEPOX on differing aerosol types as a function of aerosol acidity and RH (Gaston et al., 2014, ES&T; Riedel et al., 2015, ES&T Letters). Recently, how RH affects both aerosol acidity and aerosol-phase state (morphology) has been examined to determine how the reactive uptake kinetics changes (Zhang et al., 2018, ES&T Letters). These studies have helped to further develop models, such as CMAQ (Pye et al., 2013, ES&T), GAMMA (McNeill 2015, ES&T), and GEOS-Chem (Maraias et al. 2016, ACP), that now explicitly predict 2-methylterols and organosulfates derived from the acid-catalyzed multiphase chemistry of IEPOX as well as predicting 2-methylglyceric acid and organosulfates derived from multiphase chemistry of MAE/HMML. Since there are now models to predict many of the SOA constituents you measure here, I think you need to present your data in a clearer way in how this can improve future modeling efforts. This is a major shortcoming of the present work and why I strongly suggest this manuscript requires revision before full publication in ACP can be considered.

Another major problem with this manuscript is it is poorly written in many sections (including grammar issues and improper citations) and fails to connect their results to recent advances on isoprene SOA chemistry. I've made suggestions below in the major comments section on how some of this can be improved. One of the authors who is a native English speaker should really carefully review the written text for these authors before resubmitting the revised draft. I found the poor writing distracting while reading the manuscript.

Major Revisions:

1.) Page 2, Lines 4-6: Best to cite Guenther et al. 2006, Guenther et al., 1995 for this sentence and remove its citation in the first sentence. Along with the Goldstein and Galbally (2007) in the first sentence (lines 2-4), the authors could cite Hallquist et al. (2009, ACP).

2.) Page 2, Lines 8-10: Again, greater care is needed with this sentence! 2-methyltetrols, 2-methylglyceric acid, and organosulfates all form from multiphase chemistry and NOT gas-phase oxidation chemistry. However, I think the authors mean to say that certain oxidation products from isoprene + OH, isoprene + NO3, or isoprene + O3 undergo subsequent multiphase chemical reactions to yield these important SOA constituents.

3.) Page 2, Line 12: correct the spelling of "Claeys" here.

4.) Page 2, Line 18: Probably worth citing Surratt et al. (2006, JPCA) and Surratt et al. (2010, PNAS) for the NOx concentration having an affect on the isoprene SOA composition.

5.) Page 2, Lines 22-24: Citations to the published literature are needed for this sentence.

6.) Page 2, Lines 24-26: The authors need to also include the fact that the acid-catalyzed multiphase chemistry (or reactive uptake) of IEPOX also highly depends on the aerosol phase state. As recently shown by Zhang et al. (2018, ES&T Letters) the reactive uptake of IEPOX is adversely affected if aqueous sulfate aerosol is coated with viscous SOA. This causes a substantial diffusion barrier that the IEPOX can't react in the aqueous acidic core. This recent work is also supported by initial findings presented in Gaston et al. (2014, ES&T) and Riva et al. (2016, ES&T).

7.) Page 2, Line 26-27: Acid-catalyzed reactions of isoprene ozonlysis products have also been recently reported by Riva et al. (2015, Atmos. Environ.) and Riva et al.

(2017, Atmos. Environ.). These are worth mentioning here.

8.) Page 3, Line 5: Don't the authors mean Surratt et al. (2007, ES&T) and not Kleindienst et al. (2007)? This seems strange to me.

9.) Chemical Artifacts (Potentially Serious Issue):

Since filters were collected and extracted and derivatized for GC/EI-MS, can the authors comment on any potential artifacts? The reason this is so important is that recent work by Lopez-Hilfiker et al. (2016, ES&T) showed that IEPOX-derived SOA had a much lower volatility than expected. It turned out that they provided evidence that 2-methyltetrols and C5-alkene triols are likely thermal degradation products from accretion products (oligomers and organosulfates). Can the authors rule out that these novel GC/MS products are not simply thermal degradation products of accretion products found with the SOA?

10.) Page 5, Lines 23-28: How were OC and SO2 emissions estimated from the Poland sites? This needs to be clarified in the experimental section.

11.) Table 1: It would be easier if you could label on Table 1 what the sulfate mass concentrations were in there. Also, why didn't the authors consider running a thermodynamic model like ISORROPIA to estimate aerosol acidity. That way you can estimate what the aerosol acidity is as a function of RH. Obviously, as RH is increasing it is adding more water to your particles that you atomize the same way at each test condition, and thus, your pH is becoming less acidic.

12.) Page 8, Lines 3-5: Citations are warranted to prior studies that characterized these ions as characteristic ions for organosulfates and nitrooxy organosulfates.

13.) IEPOX-1 and IEPOX-2 is VERY STRANGE:

IEPOX-1 and IEPOX-2 don't make any sense to me. Do the authors mean they are the isomers of 3-MeTHF-3,4-diols? These were first characterized by authentic standards in Lin et al. (2012, ES&T) by the Surratt Group at UNC. 3-MeTHF-3,4-diols have similar

retention times as those shown here in the present study. I'm very confused by this.

14.) Table 2 - LC/MS section:

MW 230 is the wrong structure. I'm surprised by the carelessness here.

15.) Page 11, Lines 19-20: You're specific about the other tracers precursors (i.e., IEPOX and MPAN). Why not be more specific here for these recently reported new SOA tracers?

16.) Page 12, Lines 13-14: What are the uncertainties of using ketopinic acid to quantify all eight isoprene SOA constituents measured by GC/EI-MS?

17.) Page 12, Lines 16-17: The fact that you measure 2-methyltartaric acid organosulfate at levels above baseline in your LC/ESI-MS makes me wonder how important this compound really is to isoprene SOA formation. More specifically, what is the exact precursor to this species that forms from the gas-phase oxidation of isoprene?

18.) Page 14 , Lines 5-7: The terminology "the most abundant were organosulfates derived from 2-methyltetrols (MW 216) and 2-methylglyceric acid (MW 200)" is incorrect. This should really state "Organosulfate monomers derived from acid-catalyzed multiphase chemistry of IEPOX (MW 216) and MAE/HMML (MW 200)" to more accurately reflect their sources (Surratt et al., 2010, PNAS; Lin et al., 2012, ES&T; Lin et al., 2013, PNAS; Nguyen et al., 2015, PCCP). For the IEPOX-derived organosulfates, they are being termed 2-methyltetrol sulfates and 3-methyletrol sulfates to reflect the possible isomers that form from the multiphase chemistry of the IEPOX isomers (i.e., cis- and trans-beta-IEPOX and delta-IEPOX). Recall, Bates et al. (2014, JPCA) showed that the cis- and trans-beta-IEPOX isomers are the predominant isomers that form in the gas phase, with trans-beta-IEPOX being the most abundant. The beta-IEPOX isomers likely lead to the 2-methyltetrol sulfate isomers.

19.) Page 14, Lines 17-18: I'm not in agreement with this statement. Precursors for organosulfates typically form in the gas phase from the oxidation of isoprene. Such

precursors like IEPOX have large Henry's law constants, and thus, can partition into any aerosol water that might be present in the aerosol phase. Thus, the detection of these organosulfates could simply result from the fact that there is enough water on these particles (especially if organics condense and then take up water). I think the authors are unable to rule out this possibility based on their data.

20.) The ER labelling of experiments is really not helpful to readers. Can't you simply just call one set of experiments the acidic experiment at varying RH and the other one the non-acidic experiment at varying RH?

21.) As shown in Table 4 heading, reporting [H+]air concentration isn't really helpful to modeling. Couldn't the authors use one of the thermodynamic models to estimate what the INITIAL pH is of these particles? If the authors recall, McNeill (2015, ES&T), Pye et al. (2013, ES&T) and Maraias et al. (2017, ACP) have developed explicit models to predict IEPOX SOA. These models have been further developed by aerosol flow tube reactors that determine the reactive uptake coefficient of IEPOX as a function of acidity (Gaston et al., 2014, ES&T; Riedel et al., 2015, ES&T Letters), RH (Gaston et al., 2014, ES&T; Zhang et al., 2018, ES&T Letters) and pre-existing SOA coatings (Gaston et al., 2014, ES&T; Zhang et al., 2018, ES&T Letters). It's not clear to me how this data you show in Table 4 and Table 3 can help improve explicit modeling of many of these SOA products. The GAMMA, CMAQ, and GEOS-Chem models all now explicitly predict 2-methyltetrols and the organosulfates derived from the acid-catalyzed chemistry of IEPOX. In addition, some of these models, like CMAQ, now predict 2-methylglyceric acid and the organosulfate derived from MAE/HMML multiphase chemistry. I think much more care is needed by the authors to convince readers and reviewers how this data can be used to further improve these much needed models. I strongly believe these models have to explicitly model the acid-catalyzed multiphase chemistry of isoprene oxidation products that consider the interconnecting effects of aerosol acidity and aerosol phase state, which both depend on the RH condition.

22.) Figure 4 is poorly generated. Too difficult to read. Please regenerate this figure.

Why do some figures use color and others use black and white. I think your figures need to be more consistently generated.

23.) Figure 6: It remains unclear to me how much sulfate was present in all the conditions shown in this figure, the tables of the experimental conditions, and the experimental description. Is sulfate the same concentration in each experiment? 2-MG has been shown to be reduced in concentration if the acidity of the aerosol is high (Nguyen et al., 2015, PCCP). In fact, there is prior evidence that the nucleation of 2-MG and its corresponding oligoesters is enhanced under dry conditions (Nguyen et al., 2011, ACP; Zhang et al., 2011, AcP). I wonder, do you have evidence in your size distribution measurements of nucleation events? I ask this since it appears your sulfate seed aerosol concentrations were quite low at the start of each experimental condition.

24.) Surratt et al. (2007, ES&T) - The authors don't compare there results to that paper. That paper showed 2-MG concentration doesn't change with increasing aerosol acidity, but the 2-methyltetrols do.

---

## Author Comment (AC1) · 14 Jul 2018

Response to reviewer's comments (# 1)

This reviewer said in his general comments:

This paper reports the formation of organosulfates and other oxygenated compounds in secondary organic aerosols (SOA) generated in an indoor smog chamber from isoprene oxidation in the presence of seed aerosols. For several OSs, chemical structures are proposed on the basis of high-resolution tandem mass spectra. Isoprene accounts for a large part of VOCs in the global atmosphere, and its photooxidation has been

found to contribute to SOA formation. However, relatively few studies have been focused on the oxidation of isoprene under various relative humidities (RH), and there is still uncertainty on its chemical mechanisms and contribution to SOA formation. While this study might provide valuable information for a better understanding of the chemical pathways from the photooxidation of isoprene, the results presented here are not sufficiently supported by the analytical method and/or do not present a real novelty. In addition, the authors should have a closer look at the literature since some of their results (e.g. 2-methyltetrols) are not consistent with the existing literature. Therefore, additional information/references and major revisions would have to be provided in order to consider this article for publication.

Response: It is not clear what the reviewer means with "sufficiently supported". We disagree with the statement that the analytical methods used in this study does not support the findings in this paper. The aim of this paper is to investigate changes of the main reaction products observed in isoprene secondary organic aerosol (2-methyltetrols, 2-methylglyceric acid, organosulfates etc.) as a function of relative humidity and acidity. Relatively few studies have been focused on the oxidation of isoprene under various relative humidities (RH) mainly focusing on bulk aerosol properties, there is still uncertainty on its chemical mechanisms and contribution to SOA formation. This is the first study to focus on a wide range of isoprene SOA products evolution with relative humidity under acidic and non-acidic conditions.

The comments raised by the reviewer above are addressed in detail below.

Comment # 1. Page 2, lines 8-10: Too simplified, as written gas-phase oxidation of isoprene leads directly to the formation of isoprene-derived SOA products such as 2-methyltetrols. Please detail.

Response: We changed the following sentences from:

"The primary removal of ISO in the atmosphere is through the gas-phase reactions with hydroxyl radicals (OH), nitrate radicals (NO3) and ozone (O3) which result in

the formation of numerous oxidized SOA components, including 2-methyltetrols, 2-methylglyceric acid, C5-alkene triols and C4/C5 organosulfates (OSs)."

To

"The primary removal mechanism for isoprene is by gas-phase reactions with hydroxyl radicals (OH), nitrate radicals and, to a lesser extent, ozone. These processes result in the formation of gas and aerosol products include numerous oxidized SOA components. Aerosol species reported including 2-methyltetrols, 2 methylglyceric acid, C5-alkene triols and organosulfates (OSs) (i.e. Edney et al., 2005; Surratt et al., 2007a, 2010; Riva et al., 2016; Spolnik et al., 2018). While many of these are formed through multiphase chemistry (e.g. IEPOX channel), we cannot exclude their gas phase formation at least for 2-methyltetrols and 2-methylglyceric acid, as these compounds have been linked to gas phase reaction products from the oxidation of isoprene (Kleindienst et al., 2009) and in ambient PM2.5 (Xie at al., 2014)."

Comment # 2. Page 3, line 5: Why did the authors use Kleindienst et al. 2007 as a reference to explain the formation of organosulfate?

Response: The reviewer is correct. This was corrected as suggested by the reviewer.

Surratt, J. D., Kroll, J. H., Kleindienst, T. E., Claeys, M., Sorooshian, A., Ng, N. L., Offenberg, J. H., Lewandowski, M., Jaoui, M., Flagan, R. C., and Seinfeld, J. H.: Evidence for organosulfates in secondary organic aerosol, Environ. Sci. Technol., 41, 517–527, 2007a.

Comment # 3. Page 3, lines 13-22: Other groups have investigated the impact of RH on the SOA formation from isoprene oxidation: e.g. Abbatt's (isoprene + OH at different RH) &Surratt's groups (reactive uptake of IEPOX under different RH, acidity,: : :; isoprene +O3 at different RH and seed).

Response: More than 20 new references were added to the revised manuscript. The following references were added on page 3.

Wong J. P.S., Lee A.K.Y. and Abbatt J.P.D.: Impacts of Sulfate Seed Acidity and Water Content on Isoprene Secondary Organic Aerosol Formation, Environ. Sci. Technol., 49, 13215−13221, 2015.

Surratt, J. D.; Lewandowski, M.; Offenberg, J. H.; Jaoui, M.; Kleindienst, T. E.; Edney, E. O. and Seinfeld, J. H.: Effect of acidity on secondary organic aerosol formation from isoprene, Environ. Sci. Technol., 41, 5363– 5369, 2007b.

Riva, M.P., Budisulistiorini, S. H.P., Zhang, Z., Gold, A. and Surratt, J. D.: Chemical Characterization of Secondary Organic Aerosol Constituents from Isoprene Ozonolysis in the Presence of Acidic Aerosol, Atmos. Environ., 130, 5-13, 2016.

Spolnik G., Wach P., Rudzinski K.J., Skotak K., Danikiewicz W. and Szmigielski R., Improved UHPLC-MS/MS Methods for Analysis of Isoprene-Derived Organosulfates, Anal. Chem., 90 (5), 3416-3423, 2018.

Also, as suggested by reviewer # 3, we added the following references on page 2, line 24 (original manuscript): (Lin et al., 2013; Budisulistiorini et al., 2016; Rattanavaraha et al., 2016; Gaston et al., 2014; Riedel et al., 2015; Zhang et al., 2018).

Lin Y.H., Knipping E.M., Edgerton E.S., Shaw S.L. and Surratt J.D.: Investigating the influences of SO2 and NH3 levels on isoprene-derived secondary organic aerosol formation using conditional sampling approaches. Atmos. Chem. Phys., 13, 8457–8470, 2013.

Budisulistiorini S.H., Baumann K., Edgerton E.S., Bairai S.T., Mueller S., Shaw S.L., Knipping El. M., Gold A. and Surratt J.D.: Seasonal characterization of submicron aerosol chemical composition and organic aerosol sources in the southeastern United States: Atlanta, Georgia, and Look Rock, Tennessee. Atmos. Chem. Phys., 16, 5171–5189, 2016.

Zhang Y., Chen Y., Lambe A.T., Olson N.E., Lei Z., Craig R.L., Zhang Z., Gold A., Onasch T.B., Jayne J.T., Worsnop D.R., Gaston C.J., Thornton J.A., Vizuete W., Ault

A.P. and Surratt J.D.: Effect of the Aerosol-Phase State on Secondary Organic Aerosol Formation from the Reactive Uptake of Isoprene-Derived Epoxydiols (IEPOX). Environ. Sci. Technol. Lett., 5, 167−174, 2018.

Comment # 4. Page 3, lines 23-24: Couvidat et al. did not incorporate heterogeneous chemistry (i.e. reactive uptake) or the impact of acidity in their model. Pye et al. (2013) and Marais et al. (2016) demonstrate that replacing a reversible partitioning approach with reactive uptake to aqueous aerosol improves agreement with observations. Please revise.

Response: As suggested by the reviewer, we changed the following sentence on page 3 lines 23-24 (original manuscript) from:

"The results obtained in the smog chamber experiments are not compatible with modelling predictions that ISO SOA yield would increase under humid conditions (Couvidat et al., 2011)."

To

"The results obtained from the chamber experiments have been in agreement with recent model approaches, when reactive uptake to aqueous aerosol is used rather than a reversible partitioning approach (Pye et al., 2013; Marais et al., 2016)."

Pye, H. O. T., Pinder, R. W., Piletic, I. R., Xie, Y., Capps, S. L., Lin, Y-H., Surratt, J. D., Zhang, Z., Gold, A., Luecken, D. J., Hutzell, W. T., Jaoui, M., Offenberg, J. H., Kleindienst, T. E., Lewandowski, M., and Edney, E. O.: Epoxide Pathways Improve Model Predictions of Isoprene Markers and Reveal Key Role of Acidity in Aerosol Formation, Environ. Sci. Technol. Lett., 2, 38–42, 2013.

Marais, E. A., Jacob, D. J., Jimenez, J. L., Campuzano-Jost, P., Day, D.A., Hu, W., Krechmer, J., Zhu, L., Kim, P.S., Miller, C. C., Fisher, J. A., Travis, K., Yu, K., Hanisco, T. F., Wolfe, G.M., Arkinson, H.L., Pye, H.O.T., Froyd, K.D., Liao J. and McNeill, V.F.: Aqueous-phase mechanism for secondary organic aerosol formation from isoprene:

application to the southeast United States and co-benefit of SO2 emission controls, Atmos. Chem. Phys., 16, 1603–1618, 2016.

Comment # 5. Page 4, lines 18-19: "moderately acidic sulfate aerosol" does not have a real scientific meaning. Please determine the aerosol acidity and liquid water content using thermodynamically model.

Response: To reflect the reviewer comment, we changed the following sentence (page 4, lines 18-19 original manuscript) from

"The second experiment (ER662) was similar but run in the presence of a moderately acidic sulfate aerosol at constant concentration."

To

"The second experiment ER662 (acidic) was similar but run in the presence of acidic seed aerosol at constant concentration."

We agree with the reviewer that aerosol pH levels or aerosol liquid water concentrations would be of tremendous value to the interpretation of the results. We also generally agree with the reviewer's assessment to use modeling work (i.e., ISORROPIA (Fountoukis and Nenes, 2007); or AIM (Wexler and Clegg, 2002)) of the aerosol acidity and liquid water content, unfortunately, we do not have sufficient composition information to do the modeling with these models (ISORROPIA or AIM) appropriately. While chamber temperature, RH and particle sulfate loading are known for each reaction step, particle phase ammonium and nitrate were not measured in these experiments. And, although not strictly necessary, no gas-phase ammonia or nitric acid concentrations are available (and, as high-NOx experiments, nitric acid concentrations should be non-trivial), further complicating model predictions.

Comment # 6. Page 4, lines 30-31: Please add additional information. What was the sampling time for the filters? How much mass was collected?

Response: The following sentence (page 4, lines 30-31: original manuscript) was

changed from:

"Chamber filter samples were collected for SOA products analysis at 16.7 L min-1 using 47-mm glass fiber filters (Pall Gelman Laboratory, Ann Arbor, MI, USA)."

To

"Chamber filter samples were collected for 24 h at 16.7 L min-1 using 47-mm glass fiber filters (Pall Gelman Laboratory, Ann Arbor, MI, USA)."

Comment # 7. Page 5, lines 17-19: how many filters were analyzed? It is important to know if the tracers were identified in 1, 10 or 100 samples. Please provide some statistical information.

Response: The chemical analyses were performed for 10 ambient PM2.5 samples. The statistical analysis for such a small number of filters gives rise to high uncertainty, and thus was not applied here. Moreover, the statistical correlation for tracers was out of the scope of the paper.

To reflect the reviewer concern we changed section # 3 (page 5, lines 17-28 original manuscript) to:

"Twenty ambient PM2.5 samples were collected, onto pre-baked quartz filters using a high-volume aerosol sampler (DH-80, Digitel), from two sites (ten samples each) having strong isoprene emissions: (1) a regional background monitoring station in Zielonka, in the Kuyavian-Pomeranian Province in the northern Poland (PL; 53°39' N, 17°55' E) during summer 2016 campaign, and (2) a regional background monitoring station in Godow, PL located in the Silesian Province (49°55' N, 18°28' E) in summer 2014 campaign. Sampling times were 12 and 24 hours, respectively. Major tree species at both sites are European oak (Quercus robur L.); European hornbeam (Carpinus betulus L.); Tilia cordata (Tilia cordata Mill); European white birch (Betula pubescens Ehrh); and European alder (Alnus glutinosa Gaertn). The Zielonka station is in a forested area while the Godow station is located near a coal-fired power station

in Detmarovice (Czech Republic). Godow is also close to the major industrial cities of the Silesian region in Poland, and thus aerosol samples collected in Godow were influenced by anthropogenic sources. Several chemical and physical parameters were measured at the two sites. The relative humidity during sampling was up to 86% in Zielonka and 94% at Godow. Both locations were influenced by NOx concentration, modestly in Zielonka at 1.3 $\mu$g m-3 and at a level of 30 $\mu$g m-3 in Godow, represented by the nearest monitoring station at Zywiec, PL. The SO2 levels at Zielonka were approximately 0.6 $\mu$g m-3 and 3.0 $\mu$g m-3 at Godow. At each site, OC/EC values was determined for each filter using a thermo-optical method (Birch and Cary, 1996). The organic carbon value at Zielonka was approximately 1.7 $\mu$g m-3 and 5.4 $\mu$g m-3 at Godow, although aerosol masses were not determined."

Comment # 8. Page 6. Line 3-19: Additional information/explanation are needed in the analytical protocol to validate the results: - Temperature and pressure in the rotavapor? Did the authors evaluate the losses of the most volatile compounds –> tetrols/IEPOX/...? - The authors mentioned that internal standards were used. What is the extraction efficiency/recovery? Why did the author realize this step for the GC-MS analysis only? Extraction/recovery efficiencies have to be provided for both methods.

Response: This section (Page 6, line 3-19) refers to LCMS analysis. The temperature of rotavapor bath was 28 °C at the pressure of 150 mbar. We did not evaluate the recovery of methyltetrols/IEPOX/organosulfates due to lack of authentic standard. However, recoveries were done on other compounds, and the extraction efficiency for the LC/MS analysis of organosulfates was in the range of 94-101%.

The section related to LC-MS (page 6, lines 3-17) was changed to:

"For the LC/MS analysis, from each filter, two 1 cm2 punches were taken and twice extracted for 30 min with 15 mL aliquots of methanol using a Multi-Orbital Shaker (PSU-20i, BioSan). High purity methanol (LC-MS ChromaSolv-Grade; Sigma-Aldrich, PL) was used for the extraction of SOA filters, reconstitution of aerosol extracts, and preparation of the LC mobile phase. The two extracts were combined and concentrated to 1 mL using a rotary evaporator operated at 28 oC and 150 mbar (Rotavapor® R215, Buchi). They were then filtered with 0.2 $\mu$m PTFE syringe and taken to dryness under a gentle stream of nitrogen. High-purity water (resistivity 18.2 MΩÂůcm-1) from a Milli-Q Advantage water purification system (Merck, Poland) was used for the reconstitution of aerosol extracts and preparation of the LC mobile phase. The residues were reconstituted with 180 $\mu$L of 1:1 high purity methanol/water mixture (v / v), then agitated for 1 min. Recoveries were not taken for compounds analysed in this study, due to lack of authentic standards, however recovery of 94 -101% were measured for appropriate surrogate compounds. Extracts were analyzed by ultra-high performance liquid chromatography/electrospray ionization/time- of-flight high resolution mass spectrometry (UHPLC / ESI (-) QTOF) HRMS equipment consisting of a Waters Acquity UPLC I-Class chromatograph coupled to a Waters Synapt G2-S high resolution mass spectrometer. The chromatographic separations were performed using an Acquity HSS T3 column (2.1×100 mm, 1.8 $\mu$m particle size) at room temperature. The mobile phases consisted of 10 mM ammonium acetate (eluent A) and methanol (eluent B). To obtain appropriate chromatographic separations and responses, a gradient elution program 13 min in length was used. The chromatographic run commenced with 100% eluent A over the first 3 min. Eluent B increased from 0-100% from 3 to 8 min, held constant at 100 % from 8 to 10 min, and then decreased back from 100-0% from 10 to 13 min. The initial and final flow was 0.35 mL min-1 while the flow from 3 to 10 min was 0.25 mL min-1. An injection volume of 0.5 $\mu$L was used. The Synapt G2-S spectrometer equipped with an ESI source was operated in the negative ion mode. Optimal ESI source conditions were 3 kV capillary voltage, 20 V sampling cone at a FWHM mass resolving power of 20,000. High resolution mass spectra were recorded from m/z 50-600 in the MS or MS/MS modes. All data were recorded and analyzed with the Waters MassLynx V4.1 software package. During the analyses, the mass spectrometer was continuously calibrated by injecting the reference compound, leucine enkephalin, directly into the ESI source."

Comment # 9. Page 6, line 27: The authors report a SOA yield of 0.32% and conclude: "The values of SOA yields agree with previous smog chamber studies". It is not exact and some studies have reported SOA yields 10 times higher (Carlton et al., 2009 ACP). The authors should discuss and compare their results in more details. In addition, to really compare apple to apple the authors should discuss the impact of NO/VOC ratio, which can greatly impact isoprene SOA formation (Xu et al., 2014).

Response: SOA yield reported in the literature for isoprene photooxidation vary considerably (Carlton et al., 2009). It is very difficult to compare the reported SOA yields as noted by the reviewer, largely because literature data were reported at different conditions, and isoprene SOA yields are highly sensitive to reaction conditions and/or experimental design and conditions (e.g. temperature, relative humidity, NOx level (NO/VOC ratio), concentration and type of seed aerosol, OH concentration...). Our experiments were conducted in the presence of NOx and yields reported under these conditions correspondent to the lowest end compared to those measured under NOx-free condition. Since the objective of this study is not bulk SOA parameter analysis, and to reflect the reviewer concerns, we changed the following sentence (page 6, line 27 original manuscript) from

"The average OM/OC ratio was 1.92 $\pm$ 0.13 and the average laboratory SOA yield measured in this experiment was 0.0032 $\pm$ 0.0004. The values of SOA yields agree with previous smog chamber studies (Edney et al., 2005; Kroll et al., 2006; Dommen et al., 2006; Surratt et al., 2007; Zhang et al., 2011)."

To

"The average OM/OC ratio was 1.92 $\pm$ 0.13, and the average laboratory SOA yield measured in this experiment was 0.0032 $\pm$ 0.0004. For the non-acidic experiment, the carbon yield values range from a low 0.001 (stage 5, Table 1) at the highest relative humidity to a high of 0.004 at the lowest relative humidity (stage 1, Table 1). For the acidified experiment, carbon yield declined from above 0.011 at the lowest relative

humidity (8%) to 0.001 at the highest relative humidity (44%). Although the relative humidity considered for both acidic and non-acidic experiments do not correspond precisely, an increase of SOC was observed under acidic conditions at approximately the same relative humidity. The values of SOA yields agree with previous chamber studies reported in the literature under the same nominal conditions in the presence of NOx (Edney et al., 2005; Dommen et al., 2006; Surratt et al., 2007; Zhang et al., 2011)."

Comment # 9. Page 7, Table 1: The initial concentration of NO is higher than the steady-state concentration of NOx (= NO + NO2). Why? What was the NO2 concentration? The authors should also discuss the different regimes NO/NO2 (–> impact SOA yields).

Response: Our smog chamber experiments were run as a flow reactor and the initial NO is higher than the steady state conditions (Table 1 is correct). The initial NOx (NO + NO2) injected into the chamber was only NO and no NO2 was added to the chamber. To reflect the reviewer concern, we added to Table 1 caption: "Initial NOx was 100% NO". The NO/NO2 effect on SOA yields is important, however, we do believe that is out of the scope of this study.

Comment # 10. Page 8, lines 1-2: Why did the authors look only at the organosulfates? The authors should compare the quantification of the acids from LC/ESI(-)-MS vs GC-MS and polyol LC/ESI(+)-MS vs GC-MS. Indeed, it is now recognized that all thermal analyses (i.e. GC-MS, FIGAERO, SV-TAG) lead to a subsequent fragmentation of the oligomers. Isoprene-derived SOA is assumed to be mainly made out of oligomers especially under acidic conditions. The authors have the information/analytical tools to provide more insights on this topic.

Response: The primary objective of this study is the characterization of organosulfates/nitro-organosulfates, and organic acids/polyols . . . using two complementary methods LC-MS and GC-MS, respectively. We agree with the reviewer about a comparison between the two analytical techniques, unfortunately at the time of these

analysis no comparison was done between these two methods for the organic acids. However, we are working to synthesizing some isoprene reaction products that will be used for such comparison.

Please see our response to reviewer 3 comment # 9 related to thermal decomposition of oligomers.

Comment # 11. Page 8, lines 3-4: Please add references

Response: We added the following three references:

Darer, A.I., Cole-Filipiak N.C., O'Connor A.E. and Elrod M.J.: Formation and stability of atmospherically relevant isoprene-derived organosulfates and organonitrates. Environ. Sci. Technol. 45, 1895-1902, 2011.

Dommen, J., Metzger, A., Duplissy, J., Kalberer, M., Alfarra, M.R., Gascho, A., Weingartner, E., Prevot, A. S. H., Verheggen, B., and Baltensperger, U.: Laboratory observation of oligomers in the aerosol from isoprene/NOx photooxidation, Geophys. Res. Lett., 33(13), L13805, 2006.

Szmigielski R., Evidence for C5 organosulfur secondary organic aerosol components from in-cloud processing of isoprene: Role of reactive SO4 and SO3 radicals, Atmos. Environ., 130, 14-22, 2016.

Comment # 12. Page 8, lines 8-9: Organosulfate at m/z 230 was previously identified from the oxidation of unsaturated aldehydes.

Response: The MW 230 organosulfate was previously detected from photo-oxidation of 2-E-pentenal – one of a key green plant volatile (Shalamzari et al., ACP, 2016), however to the best of our knowledge, has never been detected from photo-oxidation of isoprene. We added the following sentence to the revised manuscript and reads

"An organosulfate with MW 230, but with a distinct structure, was recently reported in the literature from the photooxidation of 2-E-pentanal (Shalamzari et al., 2016)."

Shalamzari M., Vermeylen R., Blockhuys F., Kleindienst T.E., Lewandowski M., Szmigielski R., Rudzinski K.J., Spolnik G., Danikiewicz W., Waenhaut W. and Claeys M.: Characterization of polar organosulfates in secondary organic aerosol from the unsaturated aldehydes 2-E-pentenal, 2-E-hexenal, and 3-Z-hexenal, Atmos. Chem. Phys., 16, 7135–7148, 2016.

Comment # 13. Page 8, Table 2: Please verify the structures/formulae and specify the compounds already identified in isoprene-derived SOA.

Response: The reviewer is correct; the structure was corrected. Table 2 was updated and now contains one additional column describing compounds identified previously as well as the corresponding references.

Comment # 14. Page 11, lines 13-14: It is confusing. They are particle phase products. As written it can be understood that 2MT and 2MG are formed in the gas phase as secondary products. In addition, 2-MT should be a tertiary product not a secondary. ISO->ISOPOOH-> IEPOX—> 2-MT.

Response: For the formation of 2-methyltetrols and 2-methylglyceric acid either in the gas phase or particle phase, please see our respond to reviewer # 3 comment # 2. We do agree with the reviewer that 2-methyltetrols are not secondary oxidation products from isoprene, therefore we changed the following sentence (page 11, lines 13-14) from

"The formation of second generation compounds of ISO SOA such as 2-methyltetrols (mT) and 2-methylglyceric acid (2-mGA) is well documented in the literature."

To

"The formation of isoprene SOA products such as 2-methyltetrols (mT) and 2 methyl-glyceric acid is well documented in the literature."

Comment # 15. Page 12, lines 1-17: Need to use it to compare quantification. While the authors mentioned that the detailed analysis was performed using LC-MS in positive

and negative modes, only the negative mode is presented in this study, why? The authors could have used the benefit of deploying two complementary techniques by comparing the concentrations obtained from both methods. But instead, they are giving qualitative data and do not seem to be eager to tackle the "analytical limitations" (page 15, line 6). The authors need to revise their analytical methods and use the full potential of the methods used in this study: - Comparison LC/ESI(-)-MS vs GC need to be proposed for the acids using similar standards. - Comparison LC/ESI(+)-MS vs GC need to be proposed for the polyols using similar standards. - Additional standards commercially available (e.g. erythritol, organosulfate: : :) can be used. Indeed the authors used only one acid to quantify a wide range of compounds. What can be the impact? - Which fraction of the SOA can be explained by the compounds identified in the different experiments? What does the relative abundance mean? Is it normalized by the volume of air collected? The authors mentioned that they did not quantify the organosulfates but in the SI the concentrations are reported...Please explain Finally, the results presented are not well constrained. Therefore conclusions proposed based on the concentrations appear speculative: "The major SOA components detected we e 2-methyltetrols, 2-methylgliceric acid and its dimer, whose maximal concentrations exceeded 800, 350 and 300 ng/m3 respectively under low-humid conditions of RH 5 9% (Fig. 2)."

Response: We performed LC-MS measurements only in the negative ion mode. It is now corrected in the abstract that only negative mode was done using LC-MS. The aim of this study was not to compare both analytical methods but to detect and follow the evolution of isoprene SOA components formed under various RHs. While GC/MS was applied for quantitation of selected isoprene SOA components, LC/MS was used for more qualitative analysis with a trial to do a semi-quantitation. Since no authentic standards are available for the compound of interest in this study at the time of analysis, LC/MS responses were reduced for comparative analysis based on the contribution of a given peak to the total ion current defined as a normalization level. Due to lack of authentic standards, the quantitative analysis using GC-MS should be regarded as

indicative of the trend of isoprene products as the RH/acidity change. See also our response to comment # 10.

However, we do believe that this comparison should be done, and we are working on 1,3-butadiene oxidation products to do such comparison since the major oxidation products (threitol, erythritol, threonic acid . . .) do exist commercially.

We changed this part (page 12, line 1-17: original manuscript) to

"The LC-MS analyses focused mainly on the formation of the variety of organosulfates, nitroxy- and nitrosoxy-organosulfates. Mass spectra and proposed fragmentation pathways of newly identified components are presented in section 3.4.

3.2 Effect of relative humidity and acidity on products formation 3.2.1 Non-acidic aerosol

Table 3 and Figures 2 – 3 present the estimated amounts of polar oxygenated products detected with GC-MS and LC-MS techniques in samples from ER667 photooxidation experiments with non-acidic aerosol seeds under various RH conditions. Six products were quantified (as sums of respective isomers) based on the response factor of ketopinic acid using GS-MS. Nine other compounds were detected qualitatively using LC-MS, with chromatographic responses representing the amounts of respective analytes. Therefore, the results should be understood as a tendency of product occurrence in the chamber experiments rather than the real amounts formed. Table 3 does not contain data on 2-methyltartaric acid organosulfate (MW 244) because it occurred in the samples merely in trace amounts."

Comment # 16. Page 14, lines 10-12: The authors found that concentration of 2-MT increase at lower RH. It is not consistent with previous works (Lin et al., 2014 ES&T; Riva et al., 2016 ES&T). Please explain.

Response: Riva et al. (2016), and Lin et al. (2014) found that aerosol mass increases as the RH increases, results not consistent with our findings in this paper as well as in

Lewandowski et al. (2015), and Surratt et al. (2007a). In addition, Riva et al. (2016) report an increase in 2-methyltetrols relative concentration as the RH increases from dry condition (RH $\sim$ 5%) to wet condition (RH $\sim$ 55 %), again not consistent with our findings here as well as in Surrat et al. (2007a). The experimental conditions of Riva et al. (2016) and Lin et al. (2014) studies were fundamentally different than those in the present study. In our study, isoprene was oxidized in the presence of NOx and seed aerosol (acidic or non-acidic), however hydroxyhydropeoxide (ISOPOOH) was used as the reactant in the case of Riva et al. study, and IEPOX as the starting reactant for Lin et al. Lin et al. investigated the reactive uptake and multiphase chemistry of isoprene epoxydiols (mainly their focus was on light absorbing compounds: brown carbon) from IEPOX uptake on acidic sulfate aerosol). In addition, both Lin et al. and Riva et al. experiments were conducted under NOx-free conditions. Note Li et al used two types of seed (MgSO4, and (NH4)2(SO4)) and similar behaviour was observed in their studies. We do believe that such comparison is difficult between our results and those of Riva et al. and Lin et al., because different pathways may be responsible for SOA formation.

Under acidic condition, as the RH decreases (water content decreases), the acidity increase, which is consistent with the increase of aerosol mass observed in our study. Therefore, a decrease in RH lead generally in an increase in products abundances (as the aerosol mass increases) at lower RH, consistent with our findings. High concentrations of 2MT and 2-MG at lower RH could be due to enhanced contribution of acid-catalysed multiphase chemistry of IEPOX.

We added the following sentences (page 14, line 14) to the revised manuscript and read:

"Two recent studies (Lin et al., 2014; Riva et al., 2016) reported an increase in aerosol mass with increasing RH. Riva et al., (2016) reported also an increase in 2-methyltetrols concentrations with increasing RH. These two studies were fundamentally different than those in the present study. In our study, isoprene was oxidized in the

presence of NOx and seed aerosol (acidic and non-acidic) under a wide range of humidity, however hydroxyhydropeoxide (ISOPOOH), and IEPOX were used as reactant in Riva et al., and Lin et al. studies under two RH and free-NOx conditions."

Lin Y.H., Budisulistiorini S. H., Chu K., Siejack R.A., Zhang H., Riva M., Zhang Z., Gold A., Kautzman K.E. and Surratt J.D.: Light-Absorbing Oligomer Formation in Secondary Organic Aerosol from Reactive Uptake of Isoprene Epoxydiols., Environ. Sci. Technol., 48, 12012−12021, 2014.

Riva M.,. Budisulistiorini S.H., Chen Y., Zhang Z., D'Ambro E.L., Zhang X., Gold A., Turpin B.J., Thornton J.A., Canagaratna M.R. and Surratt J.D.: Chemical Characterization of Secondary Organic Aerosol from Oxidation of Isoprene Hydroxyhydroperoxides. Environ. Sci. Technol., 50, 9889−9899, 2016.

Comment # 17. Page 15, Table 4: Concentration of IEPOX-1 is much higher under certain conditions. Please comment? Variability of the measurements?

Response: There was a typing error, and this was now corrected. We thank the reviewer.

Comment # 18. Page 17, lines 10-14: What is new here? It has already been reported that acidity enhances the formation of isoprene-SOA components such as tetrols, organosulfates. Please add the references and further highlight the novelty.

Response: The first section on page 17 was changed to reflect the reviewer concerns:

"Early chamber studies on isoprene ozonolysis by Jang et al. (2002) and Czoschke et al. (2003) showed enhanced SOA yields in the presence of acidified aerosol seeds. Recent laboratory results showed that the acidity of aerosol seeds plays a major role in the reactive uptake of isoprene oxidation products by particle phases (Paulot et al., 2009; Surratt et al., 2010; Lin et al., 2012; Gaston et al., 2014; Riedel et al., 2015). In our study, SOC produced in acidic-seed experiments was always higher than in non-acidic seed ones under the corresponding RH conditions, while the difference diminished with increasing RH to a negligible value of 0.3 $\mu$g C m-3 at RH 44 – 49% (Table 1 and Figure S1, Supplementary Information; Surratt et al., 2007a). However, the formation of the individual organic compounds did not follow the same pattern. As an example, Figure 6 shows a comparison of the concentrations of 2-methylglyceric acid under acidic and non-acidic condition as a function of relative humidity. Acidic seed aerosol has a greater effect on 2-methylglyceric acid at lower RH. Some of the compounds produced in higher quantities in the acidic seed experiments included 2-methylglyceric acid, 2-methyltetrols, furanetriol-OS, 2-methyltetrol-NOS, 2 methylthreonic acid NOS, furanone-OS, while some other in the non-acidic seed experiments including IEPOX-2, 2 methylglyceric acid OS, 2-methylthreonic acid OS. Yields of the remaining compounds followed an inconclusive pattern (SI: Figures S1, S2, and S3; Table S1). Thus, this study shows the effect of relative humidity on the formation of a wide range of isoprene SOA products cannot easily be predicted, although the majority increases with decreasing relative humidity both under acidic and non-acidic conditions."

Comment # 19. Page 22, lines 13-18: How would this product be formed? Which kind of chemistry?

Response: The chemistry of the formation of the MW 245 product is not clear to us at present and deserves the further research. We can speculate that it could arise from a complex chain of reactions as indicated below:
* * *
[Figure]

![Chemical reaction scheme showing isoprene conversion pathways with structural formulas and labels "acid-catalyzed multiphase chemistry with IEPOX" and "OH- radical multiphase chemistry on wet/aqueous aerosol followed by recombination with NO2 radicals"]

**Fig. 1.** Authors response to reviewer 1 comments

---

## Author Comment (AC3) · 14 Jul 2018

**Response to reviewer's comments (# 2)**

This reviewer said in the general comments:

"**The manuscript presents interesting new work on elucidation of isoprene SOA formation and the influence of aerosol acidity and relative humidity**. The results are generally interesting, but the presentation needs considerable improvement before publication can be considered, in order to provide a less fragmented paper. The structure of the manuscript could be improved by moving the detailed characterization (3.3) to an earlier part of results and discussion, and then end with a general discussion of the findings in relation to current literature."

**Response**. As noted in our response to the next comment, the revised manuscript was carefully edited for English as well as in terms related to results associated with recent advances on isoprene SOA chemistry (see also our detailed response to the three reviewers). We do believe that the revised manuscript now is clear and comprehensive. Section 3.3 presents a comparison between smog chamber and the field samples, and we do believe that this section should be after the discussion of the chamber and field samples sections.

This reviewer goes on to say

"In general the use of English language should be improved. It is not the task of the reviewer to do this, and the authors should carefully read the manuscript to improve this."

**Response**. The revised manuscript was carefully edited for English as well as terms related to results associated with recent advances on isoprene SOA chemistry.

**Comment # 1.** Please define abbreviations the first time they appear, also in the abstract.

**Response**. This was updated as suggested by the reviewer.

**Comment # 2.** Abstract: I suggest adding some concluding remarks at the end of the abstract.

**Response**. The abstract was edited and additional conclusions were added to the abstract in the revised manuscript:

"**Abstract.** The effect of acidity and relative humidity on bulk isoprene aerosol parameters has been investigated in several studies, however few measurements have been conducted on individual aerosol compounds. While the focus of this study has been the examination of the effect of acidity and relative humidity on secondary organic aerosol (SOA) chemical composition from isoprene photooxidation in the presence of nitrogen oxide (NOx), a detailed characterization of SOA at the molecular level have been also conducted. Experiments were conducted in a 14.5 $m^3$ smog chamber operated in flow mode. Based on a detailed analysis of mass spectra obtained from gas chromatography-mass spectrometry of silylated derivatives in electron impact and chemical ionization modes, and ultrahigh performance liquid chromatography/electrospray ionization/time-of-flight high resolution mass spectrometry, and collision-induced dissociation in the negative ionization modes, we characterized not only typical isoprene products, but also new oxygenated compounds. A series of nitroxy-organosulfates (OS) were tentatively identified on the basic of high resolution mass spectra. Under acidic conditions, the major identified compounds include 2-methyltetrols (2MT), 2-methylglyceric acid (2MGA) and 2MT-OS. Other products identified include epoxydiols, mono- and dicarboxylic acids, other organic sulfates, and nitroxy- and nitrosoxy-OS. The contribution of SOA products from isoprene oxidation to $PM_{2.5}$ was investigated by analysing ambient aerosol collected at rural sites in Poland. Methyltetrols, 2MGA and several organosulfates and nitroxy-OS were detected in both the field and laboratory samples. The influence of relative humidity on SOA formation was modest in non-acidic seed experiments, and robust under acidic seed aerosol. Total secondary organic carbon decreased with increasing relative humidity under both acidic and non-acidic conditions. While the yields of some of the specific organic compounds decreased with increasing relative humidity others varied in an indeterminate manner from changes in the relative humidity."

**Comment # 3.** Introduction: The use of references needs significant improvement. References are missing for several statements (e.g. Page 2 line 4 "Isoprene is the most abundant nonmethane hydrocarbon...").

**Response**. As noted above, the paper was carefully updated for references and almost the majority of requests/suggestions raised by the three reviewers were incorporated in the revised manuscript. The flowing references were added to the revised manuscript:

Xie M., Hannigan M.P., Barsanti K.C., Gas/Particle Partitioning of 2‑Methyltetrols and Levoglucosan at an Urban Site in Denver, Environ. Sci. Technol., 48, 2835−2842, 2014.

Kleindienst T.E., Lewandowski M., Offenberg J.H., Jaoui M., Edney E.O., The formation of secondary organic aerosol from the isoprene + OH reaction in the absence of $NO_x$, Atmos. Chem. Phys., 9, 6541–6558, 2009.

Lin Y.-H., E. M. Knipping, E. S. Edgerton, S. L. Shaw, and J. D. Surratt. Investigating the influences of $SO_2$ and $NH_3$ levels on isoprene-derived secondary organic aerosol formation using conditional sampling approaches, Atmos. Chem. Phys., 13, 8457–8470, 2013.

Budisulistiorini S.H., Baumann K., Edgerton E.S., Bairai S.T., Mueller S., Shaw S.L., Knipping E.M., Gold A., Surratt J.D., Seasonal characterization of submicron aerosol chemical composition and organic aerosol sources in the southeastern United States: Atlanta, Georgia, and Look Rock, Tennessee, Atmos. Chem. Phys., 16, 5171–5189, 2016.

Zhang Y., Chen Y., Lambe A.T., Olson N.E., Lei Z., Craig R.L., Zhang Z., Gold A., Onasch T.B., Jayne J.T., Worsnop D.R., Gaston C.J., Thornton J.A., Vizuete W., Ault A.P., Surratt J.D., Effect of the Aerosol-Phase State on Secondary Organic Aerosol Formation from the Reactive Uptake of Isoprene-Derived Epoxydiols (IEPOX), Environ. Sci. Technol. Lett., 5, 167−174, 2018.

Riva, M., Budisulistiorini, S.H., Zhang, Z., Gold, A., Surratt, J.D., Chemical characterization of secondary organic aerosol constituents from isoprene ozonolysisin the presence of acidic aerosol, Atmos. Environ. 130, 5-13, 2016.

Riva M., Sri Budisulistiorini H., Zhang Z., Gold A., Thornton J.A., Turpin B.J., Surratt J.D., Multiphase reactivity of gaseous hydroperoxide oligomers produced from isoprene ozonolysis in the presence of acidified aerosols. Atmos. Environ., 152, 314-322, 2017.

**Comment # 4.** There is no need to introduce an abbreviation in the text for isoprene. There are already plenty of abbreviations in the manuscript, and this one only makes the text more difficult to read. Furthermore it is used inconsistently.

**Response**. We agree with reviewer, we deleted most of the use of the isoprene abbreviation (ISO) from the revised manuscript. Virtually all abbreviations for use as a noun have been eliminated, except for secondary organic aerosol (SOA) and organicsulfate (OS). Abbreviations are used mostly for adjectives, e.g. SOA formation. The only use of ISO is in combination with SOA when used as an adjective, e.g., ISO-SOA.

**Comment # 5.** Page 2 Line 10: Please add relevant references.

**Response**. See comment # 1 from reviewer #3. We changed the sentences on page 2, lines 8-10 incorporating additional references. They read now:

"The primary removal mechanism for isoprene is by gas-phase reactions with hydroxyl radicals (OH), nitrate radicals and, to a lesser extent, ozone. These processes result in the formation of gas and aerosol products include numerous oxidized SOA components. Aerosol species reported including 2-methyltetrols, 2-methylglyceric acid, $C_5$-alkene triols and organosulfates (OSs) (i.e. Edney et al., 2005; Surratt et al., 2007a, 2010; Riva et al., 2016; Spolnik et al., 2018). While many of these are formed through multiphase chemistry (e.g. IEPOX channel), we cannot exclude their gas phase formation at least for 2-methyltetrols and 2-methylglyceric acid, as these compounds have been linked to gas phase reaction products from the oxidation of isoprene (Kleindienst et al., 2009) and in ambient $PM_{2.5}$ (Xie at al., 2014)."

The new references were added to the reference section.

**Comment # 6.** Page 2 line 26: It would be relevant to refer to the following studies: Riva et al., Environ. Sci. Technol., 2016, 50 (11), pp 5580–5588 Zhang et al., Environ. Sci. Technol. Lett., 2018, 5 (3), pp 167–174.

**Response**. As suggested by the reviewer, we added the following references to the revised manuscript:

Riva, M., Budisulistiorini, S.H., Zhang, Z., Gold, A., and Surratt, J.D.: Chemical characterization of secondary organic aerosol constituents from isoprene ozonolysis in the presence of acidic aerosol. Atmos. Environ. 130, 5-13, 2016.

Riva M., Budisulistiorini S.H., Zhang Z., Gold A., Thornton J.A., Turpin B.J., and Surratt J.D.: Multiphase reactivity of gaseous hydroperoxide oligomers produced from isoprene ozonolysis in the presence of acidified aerosols. Atmos. Environ., 152, 314-322, 2017.

Zhang Y., Chen Y., Lambe A.T., Olson N.E., Lei Z., Craig R.L., Zhang Z., Gold A., Onasch T.B., Jayne J.T., Worsnop D.R., Gaston C.J., Thornton J.A., Vizuete W., Ault A.P., Surratt J.D.: Effect of the Aerosol-Phase State on Secondary Organic Aerosol Formation from the Reactive Uptake of Isoprene-Derived Epoxydiols (IEPOX), Environ. Sci. Technol. Lett., 5, 167–174, 2018.

**Comment # 7.** Page 2 Line 29-31. This sentence is hard to understand and need references.

**Response**. To reflect the reviewer concern, we changed the following sentence from:

"The most common in the atmosphere and investigated were organosulfates derived from the oxidation of ISO that were identified both in smog chamber experiments and in field studies."

To

"The most common isoprene organosulfates have been identified both in smog chamber experiments and in field studies (Surratt et al., 2007a; 2008, 2010; Gomez-Gonzalez et al., 2008; Shalamzari et al., 2013; Tao et al., 2014; Hettiyadura et al., 2015; Spolnik et al., 2018)."

**Comment # 8.** Page 3 Line 26: Define SOC. I think "e.g." should maybe be "i.e.".

**Response**. The reviewer is correct. We deleted from the revised manuscript "e.g.". Note SOC is now defined in the revised manuscript when it appears the first time. SOC: secondary organic carbon.

**Comment # 9.** P3. Line 30- 31: Other research groups were the first to develop analysis of organosulfates using LC/MS. I suggest to remove "developed in our laboratories" from the sentence. Section 2.2.

**Response**. We changed the following sentence from:

"Two techniques developed in our laboratories were used: (1) analysis of organosulfates compounds based on LC/MS (Szmigielski, 2016, Rudzinski et al., 2009) and (2) analysis of non-sulfate compounds based on derivatization techniques followed by GC-MS analysis (Jaoui et al., 2004)."

To

"Organosulfate compounds were analyzed using LC/MS measurements (Szmigielski, 2016; Rudzinski et al., 2009; Darer et al., 2011; Surratt et al., 2007a), while non-sulfate oxygenated compounds were examined using derivatization followed by GC-MS analysis (Jaoui et al., 2004). "

Additional references were added to the revised manuscript.

Darer, A.I., Cole-Filipiak N.C., O'Connor A.E., and Elrod M.J.: Formation and stability of atmospherically relevant isoprene-derived organosulfates and organonitrates, Environ. Sci. Technol. 45, 1895-1902, 2011.

Surratt, J.D., Kroll J.H., Kleindienst T.E., Edney E.O., Claeys M., Sorooshian A., Ng N.L., Offenberg J.H., Lewandowski M., Jaoui M., Flagan R.C., and Seinfeld J.H.: Evidence for organosulfates in secondary organic aerosol, Environ. Sci. Technol. 41, 517-527, 2007.

**Comment # 10.** Please add information on sampling time and tree species in the area. In several instances "emission" should be replaced by "concentration".

**Response**. We changed the following sentence to reflect the reviewer concerns from:

"Ambient fine aerosol samples (PM2.5) were collected onto pre-baked quartz-fiber filters using a high-volume aerosol samplers (DH-80, Digitel) at the regional background monitoring station in Zielonka, located in the Kuyavian-Pomeranian Province in the northern Poland (53°39'N, 17°55'E) during the 2016 summer campaign, and at the regional background monitoring station in Godow, located in the Silesian Province in southern Poland (49°55'N 18°28'E) in summer 2014. At both sites, strong emission of isoprene occurred. The Zielonka station is located in the forested rural area while the Godow station is located near a coal-fired power station in Detmarovice (Czech Republic) and close to big industrial cities of the Silesian agglomeration (Poland). Therefore, SOA collected in Godow can be influenced by anthropogenic aerosol precursors. The relative humidity level during sampling in Zielonka was 86%, SO2 emission was estimated at 0.6 μg/m3 and 25 OC value was 1.68 μg/m3. The relative humidity level during sampling in Godow was 94%, SO2 emission was estimated at 3.0 μg/m3 (approximate value from the nearest sampling station - Zory) and OC value was 5.43 μg/m3. Both locations were influenced by NOx emission – slightly in Zielonka at 1.3 μg/m3 and 30.0 μg/m3 in Godow (approximate value from the nearest sampling station - Zywiec)."

To

"Twenty ambient PM$_{2.5}$ samples were collected, onto pre-baked quartz filters using a high-volume aerosol sampler (DH-80, Digitel), from two sites (ten samples each) having strong isoprene emissions: (1) a regional background monitoring station in Zielonka, in the Kuyavian-Pomeranian

Province in the northern Poland (PL; 53°39' N, 17°55' E) during summer 2016 campaign, and (2) a regional background monitoring station in Godow, PL located in the Silesian Province (49°55' N, 18°28' E) in summer 2014 campaign. Sampling times were 12 and 24 hours, respectively. Major tree species at both sites are European oak (Quercus robur L.); European hornbeam (Carpinus betulus L.); Tilia cordata (Tilia cordata Mill); European white birch (Betula pubescens Ehrh); and European alder (Alnus glutinosa Gaertn). The Zielonka station is in a forested area while the Godow station is located near a coal-fired power station in Detmarovice (Czech Republic). Godow is also close to the major industrial cities of the Silesian region in Poland, and thus aerosol samples collected in Godow were influenced by anthropogenic sources.

Several chemical and physical parameters were measured at the two sites. The temperature range during sampling at both sites range from 27-28 °C. The relative humidity during sampling was up to 86% in Zielonka and 94% at Godow. Both locations were influenced by $NO_x$ concentration, modestly in Zielonka at 1.3 µg m$^{-3}$ and at a level of 30 µg m$^{-3}$ in Godow, represented by the nearest monitoring station at Zywiec, PL. The $SO_2$ levels at Zielonka were approximately 0.6 µg m$^{-3}$ and 3.0 µg m$^{-3}$ at Godow. At each site, OC/EC values was determined for each filter using a thermo-optical method (Birch and Cary, 1996). The organic carbon value at Zielonka was approximately 1.7 µg m$^{-3}$ and 5.4 µg m$^{-3}$ at Godow, although aerosol masses were not determined."

**Comment # 11.**  P. 6 Line 7: Define the abbreviation.

**Response**. This was done as requested by the reviewer.

**Comment # 13.** P. 7 Line 7- page 8 line 5: This should be moved to the experimental section.

**Response**. We do believe that this section should remain here because it provides a discussion on the mass spectra behavior of the products.

**Comment # 14.** Table 2 needs references to studies where these compounds were first identified.

**Response**. We added one column to Table 2, providing references associated with the compounds reported in Table 2.  The revised Table 2 is shown below with the new references:

**Table 2.** Products detected in SOA samples from chamber experiments using GC-MS and LC-MS.

| GC-MS |
| --- |

| Chemical Formula | m/z BSTFA Derivative (methane-CI) | MW MW$_{BSTFA}$ (g mol$^{-1}$) | Tentative Structure* and Chemical Name | References |
|---|---|---|---|---|
| $C_5H_{10}O_2$ | 247, 231, 157, 147, 73 | 102 246 | 3-methyl-3-butene-1,2-diol (C$_5$-diol-1) | Wang et al. 2005 Surratt et al., 2006 |
| $C_5H_{10}O_3$ | 263, 247, 173, 83, 73, | 118 262 | 2-methyl-2,3-epoxy-but-1,4-diol (IEPOX-1) | Paulot et al., 2009 Surratt et al., 2010 Zhang et al., 2012 |
| $C_5H_{10}O_3$ | 263, 247, 173, 83, 73 | 118 262 | 2-methyl-3,4-epoxy-but-1,2-diol (IEPOX-2) | |
| $C_4H_8O_4$ | 337, 321, 293, 219, 203 | 120 336 | 2-methylglyceric acid (2-MG) | Claeys et al., 2004 Surratt et al., 2006 Edney et al., 2005 Szmigielski et al. 2007 |
| $C_5H_{12}O_4$ | 409, 319, 293, 219, 203 | 136 424 | 2-methylthreitol (2-MT) | Claeys et al., 2004 Wang et al., 2004 Edney et al., 2005 Surratt et al., 2006 Nozière et al., 2011 |
| $C_5H_{12}O_4$ | 409, 319, 293, 219, 203 | 136 424 | 2-methylerythritol (2-MT) | |
| $C_8H_{14}O_7$ | 495, 321, 219, 203, 73 | 222 510 | 2-methylglyceric acid dimer | Surratt et al., 2006 Szmigielski et al. 2007 |

| | | | (2-MG dimer) | |

| Chemical Formula | *m/z* Main Ions | MW (g mol$^{-1}$) | Tentative Structure and Chemical Name* | References |
|---|---|---|---|---|
| $C_5H_{10}O_6S$ | 197, 167, 97, 81 | 198 | IEPOX-derived organosulfate | Tao et al., 2014 |
| $C_4H_8O_7S$ | 199, 119, 97, 73 | 200 | 2-methylglyceric acid organosulfate (2-MG OS) | Surratt et al., 2007a  Gomez-Gonzalez et al., 2008  Shalamzari et al., 2013  Riva et al., 2016 |
| $C_5H_8O_7S$ | 211, 193, 113, 97 | 212 | 2(3*H*)-furanone, dihydro-3,4-dihydroxy-3-methyl organosulfate | Surratt et al., 2008  Hettiyadura et al., 2015  Spolnik et al., 2018 |
| $C_5H_{10}O_7S$ | 213, 183, 153, 97 | 214 | 2,3,4-furantriol, tetrahydro-3-methyl-organosulfate | Hettiyadura et al., 2015  Spolnik et al., 2018 |
| $C_5H_{12}O_7S$ | 215, 97 | 216 | 2-methyltetrol organosulfate (2-MT OS) | Surratt et al., 2007a  Gomez-Gonzalez et al., 2008  Surratt et al., 2010 |
| $C_5H_{10}O_8S$ | 229, 149, 97, 75 | 230 | 2-methylthreonic acid organosulfate | This study |

| | | | | | |
|---|---|---|---|---|---|
| $C_5H_9O_9S$ | 243, 163, 145, 101 | 244 |  2-methyltartaric acid organosulfate | This study |
| $C_5H_{11}NO_8S$ | 244, 226, 197, 183, 153, 97 | 245 |  2-methyltetrol nitrosoxy-organosulfate | This study |
| $C_5H_{11}NO_9S$ | 260, 197, 183, 153, 97 | 261 |  2-methyltetrol nitroxyorganosulfate | Surratt et al., 2007a

Surratt et al., 2008 |
| $C_5H_9NO_{10}S$ | 274, 211, 193, 153, 97 | 275 |  2-methylthreonic acid nitroxy-organosulfate | This study |

* For more stereo-chemically complex molecules a representative isomer is shown.

**References**

Claeys M., Graham B., Vas G., Wang W., Vermeylen R., Pashynska V., Cafmeyer J., Guyon P., Andreae M.O., Artaxo P., Maenhaut W., Formation of secondary organic aerosols through photooxidation of isoprene. Science, 303, 1173-1176, 2004.

Edney E.O., Kleindienst T.E., Jaoui M, Lewandowski M, OffenbergJ.H, WangW, ClaeysM., Formation of 2-methyltetrols and 2-methylglyceric acid in secondary organic aerosol from laboratory irradiated isoprene/NOx/SO$_2$/air mixtures and their detection in ambient PM$_{2.5}$ samples collected in the eastern United States., Atmos. Environ., 39, 5281-5289, 2005.

Wang W., Kourtchev I., Graham B., Cafmeyer J., Maenhaut W., Claeys M., .Characterization of oxygenated derivatives of isoprene related to 2-methyltetrols in Amazonian aerosols using trimethylsilylation and gas chromatography/ion trap mass spectrometry, Rapid Commun. Mass Spectrom., 19, 1343-51 2005.

Surratt J.D., Murphy SM, Kroll J.H., Ng N.L., Hildebrandt L, Sorooshian A, Szmigielski R, Vermeylen R, Maenhaut W, Claeys M, Flagan RC, Seinfeld JH. Chemical composition of secondary organic aerosol formed from the photooxidation of isoprene., J. Phys. Chem. A; 110, 9665–9690, 2006.

Szmigielski R, Surratt JD, Vermeylen R, Szmigielska K, Kroll JH, Ng NL, M urphy SM, Sorooshian A, Seinfeld JH, Claeys M., Characterization of 2-methylglyceric acid oligomers in secondary organic aerosol formed from the photooxidation of isoprene using trimethylsilylation and gas chromatography/ion trap mass spectrometry. Journal of Mass Spectrometry, 42, 101, 2007.

Surratt, J. D., Kroll, J. H., Kleindienst, T. E., Claeys, M., Sorooshian, A., Ng, N. L., Offenberg, J. H., Lewandowski, M., Jaoui, M., Flagan, R. C., and Seinfeld, J. H.: Evidence for organosulfates in secondary organic aerosol, Environ. Sci. Tech-nol., 41, 517–527, 2007a.

Gomez-Gonzalez Y.G., Surratt J.D., Cuyckens F., Szmigielski R., Vermeylen R., Jaoui M., Lewandowski M., Offenberg M.,H.,Kleindienst T.E., Edney E.O., Blockhuys F., Van Alsenoy Ch., Maenhaut W., Claeys M., Characterization of organosulfates from the photooxidation of isoprene and unsaturated fatty acids in ambient aerosol using liquid chromatography/(−) electrospray ionization mass spectrometry, J. Mass Spectrom. 43, 371–382, 2008.

Surratt, J. D., Gómez-González, Y., Chan, A. W. H., Vermeylen, R., Shahgholi, M., Kleindienst, T. E., Edney, E. O., Offenberg, J.H., Lewandowski, M., and Jaoui, M.: Organosulfate Formation in Biogenic Secondary Organic Aerosol, J. Phys. Chem. A., 112, 8345–8378, 2008.

Paulot, F., Crounse J. D., Kjaergaard, H.G., Kurten, A., St Clair,J. M., Seinfeld, J. H., and Wennberg, P. O., Unexpected epoxide formation in the gas-phase photooxidation of isoprene, Science, 325, 730–733, 2009.

Surratt, J. D., Chan, A. W. H., Eddingsaas, N. C., Chan, M. N., Loza, C. L., Kwan, A. J., Hersey, S. P., Flagan, R. C., Wennberg, P. O., and Seinfeld, J. H.: Reactive intermediates revealed in secondary organic aerosol formation from isoprene, P. Natl. Acad. Sci. USA, 107, 6640–6645, 2010.

Nozière B., González N.D.J., Borg-Karlson A.K., Pei Y., Redeby J.P., Krejci R., Dommen J., Prevot A.S.H., Anthonsen T., Atmospheric chemistry in stereo: A new look at secondary organic aerosols from isoprene, Geoph. Res. Lett., 38, L11807, doi:10.1029/2011GL047323, 2011.

Jaoui, M., Szmigielski, R., Nestorowicz, K., Kolodziejczyk, A., Rudziński, K.J., Danikiewicz, W., Lewandowski, M., and Kleindienst, T.E.: Organic Acids as Molecular Markers for Highly Oxygenated Molecules (HOMs) in Aged Isoprene Aerosol. In preparation.

Zhang Z., Lin Y.H., Zhang H., Surratt J.D., Ball L.M., Gold A., Synthesis of isoprene atmospheric oxidation products: isomeric epoxydiols and the rearrangement products cis- and trans-3-methyl-3,4-dihydroxytetrahydrofuranAtmos. Chem. Phys., 12, 8529–8535, 2012.

Shalamzari M., Ryabtsova O., Kahnt A., Vermeylen R., Hérent M.F., Quetin-Leclercq J., Van der Veken P., Maenhaut W., Claeys M., Mass spectrometric characterization of organosulfates related to secondary organic aerosol from isoprene, Rapid Commun. Mass Spectrom. 27, 784–794, 2013.

Tao S., Lu X., Levac N., Bateman A.P., Nguyen T.B., Bones D.L., Nizkorodov S.A., Laskin J., Laskin A., Yang X., Molecular Characterization of Organosulfates in Organic Aerosols from Shanghai and Los Angeles Urban Areas by Nanospray-Desorption Electrospray Ionization High-Resolution Mass Spectrometry, Environ. Sci. Technol., 48, 10993–11001, 2014.

Hettiyadura, A. P. S., Stone, E. A., Kundu, S., Baker, Z., Geddes, E., Richards, K., and Humphry, T.: Determination of atmospheric organosulfates using HILIC chromatography with MS detection, Atmos. Meas. Tech., 8, 2347–2358, 2015.

Riva, M., Budisulistiorini, S.H., Zhang, Z. F., Gold, A., Surratt, J.D.: Chemical Characterization of Secondary Organic Aerosol Constituents From Isoprene Ozonolysis in the Presence of Acidic Aerosol, Atmos. Environ., 130, 5–13, 2016.

Spolnik G., Wach P., Rudzinski K.J., Skotak K., Danikiewicz W., Szmigielski R., Improved UHPLC-MS/MS Methods for Analysis of Isoprene-Derived Organosulfates, Anal. Chem., 90, 3416-3423, 2018.

Note, the reference section of the revised manuscript was updated.

**Comment # 15.** Figure 1: Why are all these chromatograms shown, when they are not discussed in detailed in the text? I suggest to reduce the figure to one or two chromatograms - if they are discussed.

**Response**. Figure 1 presents GC-MS Extracted Ion Chromatograms (EIC) obtained for ER667 isoprene photo-oxidation originated under non-acidic seed aerosol at 5 RH. All chromatograms in this figure were acquired using the same analytical method, therefore Figure 1 provides a direct visual comparison of the evolution of reaction products at the 5 RH. The identifications of 2-methylerythronic acid, 2-methylthreonic acid, and 2-methyltartaric acid are tentative and further work is being conducted to understand their role in isoprene SOA, therefore they were deleted from the present manuscript for clarity purpose. An effort is underway to synthesize these compounds in our laboratory.

**Comment # 16.** P11.L10: What do you mean by "attained"? C2 Table 3: Add percentage for RH (RH9 -> RH9%). Data in Table 4 and Figures 4-5 should be presented and discussed in more detail.

**Response**. We changed the following sentence from:

"According to attained chromatographic separations a number of isomers of analyzed compounds were distinguished, i.e. IEPOX-1 and IEPOX-2 or 4 isomers of 2-methyltetrols, however, only some of them are marked on the figure."

To

"According to acquired chromatograms shown in Figure 1, several isomers associated with the compounds analyzed can be distinguished, i.e. IEPOX-1 and IEPOX-2, 4 isomers of 2-methyltetrols and their relative contributions to SOA masses at various relative humidity levels."

Tables 3 and 4 were updated as requested by the reviewer.

We added the following sentence on page 15, line 7 (original manuscript) to the revised manuscript to reflect the reviewer concern:

"The presence of 2-methyltetrols and 2-methylglyceric acid and their sulfated analogues in isoprene SOA at a wide range of RH conditions, suggests that SOA water content does not affect significantly their formation."

**Comment # 17.** Page 17 Line 9: Please write this as a complete sentence.

**Response**. We changed the following sentence on page 17, line 9 from:

"Figure 6 compares the results for 2-methylglyceric acid – combined effect of RH and H2SO4 was stronger than that 10 of RH alone. Some of the compounds were produced in higher quantities in the acidic seed experiments (2-methylglyceric acid, 2-methyltetrols, furanetriol OS, 2-methyltetrol NOS, 2-methylthreonic acid NOS, furanone OS) while some other in the non-acidic seed experiments (IEPOX-2, methylthreonic acid, 2-methylglyceric acid OS, 2-methylthreonic acid OS). Yields of the remaining compounds followed a mixed pattern (supplementary information: see Figs S1, and S2, and Table S1)."

To

"As an example, Figure 6 shows a comparison of the concentrations of 2-methylglyceric acid under acidic and non-acidic condition as a function of relative humidity. Acidic seed aerosol has a greater effect on 2-methylglyceric acid at lower relative humidity.  Some of the compounds produced in higher quantities in the acidic seed experiments included 2-methylglyceric acid, 2-methyltetrols, furanetriol-OS,

2-methyltetrol-NOS, 2-methylthreonic acid NOS, furanone-OS, while some other in the non-acidic seed experiments including IEPOX-2, 2-methylglyceric acid OS, 2-methylthreonic acid OS. Yields of the remaining compounds followed an inconclusive pattern (SI: Figures S1, S2, and S3; Table S1). Thus, this study shows the effect of relative humidity on the formation of a wide range of isoprene SOA products cannot easily be predicted, although the majority increases with decreasing relative humidity both under acidic and non-acidic conditions."

**Comment # 18.** Figures 7-14. Some of these should be moved to Supplementary. Instead of experiment number it would be more useful to the reader to list whether an experiment was non-acidic or acidic.

  **Response**. We moved Figures 9, 10, 13, and 14 to the supplementary information. We changed accordingly the numbering of the remaining figures. We also incorporate when appropriate in the revised manuscript if an experiment is non-acidic on acidic instead of experiment number.

**Comment # 19.** Page 21: please add figure number to the mass spectrum.

**Response**. The mass spectrum is part of Figure 15 as is the case of Figure 16.

**Comment # 20.** Conclusion: How much do the quantified compounds make-up of the total SOA mass? It is important to keep this in mind, when discussing the effects of acidity and RH.

**Response**. Given the relatively limited organic quantification available in these experiments due unfortunately to lack of authentic standards as well using ketopinic acid as surrogate for quantitative analysis, it is difficult to assess their contribution to SOA mass accurately. In this paper, only estimates were provided as noted in the main manuscript.

We have also added the following paragraphs prior to the summary section, which includes an additional review of the limitations of the quantitative analysis presented in the paper, including areas requiring additional investigation in future work, which includes discussion of the need for using authentic standards, and modeling work. The two paragraphs are:

"While these experiments provide an analysis of a wide range of isoprene reaction products in the aerosol phase as a function of RH and acidity, they also include a number of shortcomings that need to be addressed in future work. Perhaps the most significant is the use of authentic standards to assess the contribution of these products to SOA mass at different RH. In addition, when the relative humidity is varied, it is important to measure aerosol liquid water content directly or estimated using thermodynamic models, such as ISOPROPIA (Fountoukis and Nenes, 2007) or AIM (Wexler and Cregg, 2002), and other gas and particle composition (e.g. inorganic species). Liquid water inorganic species measurements were not available for this study.

The use of these marker compounds for ambient air quality models can follow the approach of Pye et al. (2013). In such an approach, the model is run using a base case chemical mechanism for isoprene, where there is no adjustment for acidity and relative humidity. A comparison can then be made with the same model having such an adjustment incorporated within the isoprene mechanism. The markers can then serve as constraints to the PM observations. For the U.S. the Community Multiscale Air Quality (CMAQ) model is frequently used for ozone and PM ambient concentrations (Pye et al., 2013). For Poland, a similar approach can be used with a European model having the appropriate meteorology and chemical mechanism (Miranda et al. 2015)."

The following references were added to the reference section.

Miranda, A., Silveira, C., Ferreira, J., Montheiro, A., Lopes, D., Relvas, H., Borrego, C., and Roebeling, P.: Current air quality plans in Europe designed to support air quality management policies. Atmospheric Pollution Research, 6, 434-443, 2015.

Pye, H. O. T., Pinder, R. W., Piletic, I. R., Xie, Y., Capps, S. L., Lin, Y-H., Surratt, J. D., Zhang, Z., Gold, A., Luecken, D. J., Hutzell, W. T., Jaoui, M., Offenberg, J. H., Kleindienst, T. E., Lewandowski, M., and Edney, E. O.: Epoxide Pathways Improve Model Predictions of Isoprene Markers and Reveal Key Role of Acidity in Aerosol Formation, Environ. Sci. Technol. Lett., 2, 38–42, 2013.

Wexler, A. S. and Clegg, S. L.: Atmospheric aerosol models for systems including the ions $H^+$, $NH_4^+$, $Na^+$, $SO_4^{2-}$, $NO_3^-$, $Cl^-$, $Br^-$, and $H_2O$, J. Geophys. Res., 107, 4207, doi:10.1029/2001JD000451, 2002.

Fountoukis, C., and Nenes, A.: ISORROPIA II: a computationally efficient thermodynamic equilibrium model for $K^+ - Ca^{2+} - Mg^{2+} - NH_4^+ - Na^+ - SO_4^{2-} - NO_3^- - Cl^- - H_2O$ aerosols. Atmos. Chem. Phys., 7, 4639–4659, 2007.

---

## Author Comment (AC4) · 14 Jul 2018

**Response to reviewer's comments (# 3)**

This reviewer said in the general comments:

"This manuscript measures the chemical composition changes (as well as bulk SOA yields) of isoprene SOA produced under acidic and non-acidic conditions as a function of relative humidity. The kind of **results presented here could certainly be of value to the atmospheric and aerosol research communities**. *However, as I will stress in some of my major comments below, I think one thing that is missing is a stronger connection to current models that explicitly predict isoprene SOA formation through acid-catalyzed multiphase chemical processes.* Of these processes, the acid-catalyzed multiphase chemistry of IEPOX on acidic (and wet) sulfate aerosol has been shown to be one of the dominant sources of isoprene SOA in atmospheric PM samples (e.g. Claeys et al., 2004, Science; Lin et al., 2013, ACP; Budisulistiorini et al., 2015, ACP, etc. etc.). The acid-catalyzed multiphase chemistry of high-NOx SOA precursors, such as HMML/MAE, have been shown to yield very little SOA in atmospheric PM samples (e.g., Lin et al., 2013, ACP; Budisulistiorini et al., 2015, ACP; Rattanavaraha et al., 2016, ACP). Since IEPOX has been shown to be so important to forming SOA in atmospheric PM samples, recent work has been really aimed at measuring reactive uptake (or mulitphase chemical) kinetics of IEPOX on differing aerosol types as a function of aerosol acidity and RH (Gaston et al., 2014, ES&T; Riedel et al., 2015, ES&T Letters). Recently, how RH affects both aerosol acidity and aerosol-phase state (morphology) has been examined to determine how the reactive uptake kinetics changes (Zhang et al., 2018, ES&T Letters). These studies have helped to further develop models, such as CMAQ (Pye et al., 2013, ES&T), GAMMA (McNeill 2015, ES&T), and GEOS-Chem (Maraias et al. 2016, ACP), that now explicitly predict 2-methylterols and organosulfates derived from the acid-catalyzed multiphase chemistry of IEPOX as well as predicting 2-methylglyceric acid and organosulfates derived from multiphase chemistry of MAE/HMML. **Since there are now models to predict many of the SOA constituents you measure here, I think you need to present your data in a clearer way in how this can improve future modeling efforts. This is a major shortcoming of the present work and why I strongly suggest this manuscript requires revision before full publication in ACP can be considered."**

**Response.** First, we would like to thank this reviewer for the time and effort spent in reading and evaluating this manuscript. While the results presented in this paper could certainly benefit from additional information regarding modeling work, we do not believe that this information is vital to the comparisons presented. Although, we do agree with the reviewer's assessment of the importance of modeling.

Multiphase chemistry (e.g. relative uptake of IEPOX on aerosol particles) has been shown to be important in forming SOA in atmospheric PM samples (e.g. Gaston et al., 2014; Riedel et al., 2015), and as noted in our response to reviewer # 2, comment # 20, it is important to measure directly or have an accurate estimate of aerosol liquid water content in order to accurately "model" the formation of isoprene reaction products formed under a wide range of RH. Liquid water inorganic species measurements were not available for this study. Air quality modeling to improve organic PM predictions using the organic markers as constraints is both outside the scope of this paper and, in any case, is not at the stage to help improve model results. How such modeling might function is given in the last paragraph of the Results and Discussion.

We have added now the following paragraph to address this issue just before the Summary section in the revised manuscript:

"While these experiments provide an analysis of a wide range of isoprene reaction products in the aerosol phase as a function of RH and acidity, they also include a number of shortcomings that need to be addressed in future work. Perhaps the most significant is the use of authentic standards to assess the contribution of these products to SOA mass at different RH. In addition, when the relative humidity is varied, it is important to measure aerosol liquid water content directly or estimated using thermodynamic models, such as ISOPROPIA (Fountoukis and Nenes, 2007) or AIM (Wexler and Cregg, 2002), and other gas and particle composition (e.g. inorganic species). Liquid water inorganic species measurements were not available for this study.

"The use of these marker compounds for ambient air quality models can follow the approach of Pye et al. (2013). In such an approach, the model is run using a base case chemical mechanism for isoprene, where there is no adjustment for acidity and relative humidity. A comparison can then be made with the same model having such an adjustment incorporated within the isoprene mechanism provided that absolute concentrations can be assigned. The markers can then serve as constraints to the PM observations. For the U.S. the Community Multiscale Air Quality (CMAQ) model is frequently used for ozone and PM ambient concentrations (Pye et al., 2013). For Poland, a similar approach can be used with a European model having the appropriate meteorology and chemical mechanism (Miranda et al. 2015)."

The reviewer goes on to say"

"Another major problem with this manuscript is it is poorly written in many sections (including grammar issues and improper citations) and fails to connect their results to recent advances on isoprene SOA chemistry. I've made suggestions below in the major comments section on how some of this can be improved. One of the authors who is a native English speaker should really carefully review the written text for these authors before resubmitting the revised draft. I found the poor writing distracting while reading the manuscript."

**Response.** The revised manuscript was carefully edited for appropriate scientific-English terminology as well as to explicate certain sections or paragraphs that may have lacked clarity or been perplexing. The results have been augmented to reflect recent advances in isoprene SOA chemistry.

**Major revisions.**

**Comment #1.** "Page 2, Lines 4-6: Best to cite Guenther et al. 2006, Guenther et al., 1995 for this sentence and remove its citation in the first sentence. Along with the Goldstein and Galbally (2007) in the first sentence (lines 2-4), the authors could cite Hallquist et al. (2009, ACP)."

**Response.** This was corrected as suggested by the reviewer. Guenther et al., 2006 was added to the reference section.

**Comment #2.** "Page 2, Lines 8-10: Again, greater care is needed with this sentence. 2- methyltetrols, 2-methylglyceric acid, and organosulfates all form from multiphase chemistry and NOT gas-phase oxidation chemistry. However, I think the authors mean to say that certain oxidation products from isoprene + OH, isoprene + NO3, or isoprene + O3 undergo subsequent multiphase chemical reactions to yield these important SOA constituents."

**Response.** We did not understand the reviewer concerns here. We refer to isoprene removal through gas phase chemistry and not to its oxidation products. We agree somewhat with the reviewer that 2- methyltetrols, 2-methylglyceric acid, and organosulfates are formed only through multiphase chemistry but we cannot exclude their gas phase formation also, at least for the 2-methyltetrols and 2-methylglyceric acid. For examples, 2-methyltetrols and 2-methylglyceric acid were detected in our laboratory in the gas phase from the chamber

oxidation of isoprene and as indicated in the associated mechanism (Kleindienst et al., 2009). Our chamber data are consistent with isoprene oxidation products 2-methyltetrols and 2-methylglyceric acid presence either in the gas phase, in the particle phase, or both. 2-Methyltetrols were also reported in ambient samples in the gas phase (Xie et al., 2014). During a recent inter-comparison study during 2013 SOAS field study, using four "real-time" instruments (FIGAERO-HRToF-CIMS with acetate ionization source; FIGAERO-HRToF-CIMS with iodide ionization source; semi-volatile thermal desorption aerosol GC-MS (SV-TAG); high-resolution thermal desorption proton-transfer reaction mass spectrometer (HR-TD-PTRMS)) for gas/particle partitioning of organic species, Thomson et al. (2016) show the difficulties of these instruments in the interpretation of complex ambient samples. Note that the formation of 2-methyltetrols and 2-methylglyceric acid and other isoprene/momoterpene products in the gas or particle phase through multiphase chemistry remain poorly constrained.

However, to reflect the reviewer comment, we changed the following sentences from:

"The primary removal of ISO in the atmosphere is through the gas-phase reactions with hydroxyl radicals (OH), nitrate radicals (NO3) and ozone (O3) which result in the formation of numerous oxidized SOA components, including 2-methyltetrols, 2-methylglyceric acid, C5-alkene triols and C4/C5 organosulfates (OSs). These compounds were identified in ambient PM2.5 (particulate matter with diameter < 2.5 μm) in several places around the world while SOA generated from isoprene was reported to account for up to 20 – 50% of the overall SOA budget (Clayes et al., 2004a; Wang et al., 2005; Henze and Seinfeld, 2006; Kroll et al., 2006; Surratt et al., 2006; Hoyle et al., 2007)." which is changed to:

"The primary removal mechanism for isoprene is by gas-phase reactions with hydroxyl radicals (OH), nitrate radicals and, to a lesser extent, ozone. These processes result in the formation of gas and aerosol products include numerous oxidized SOA components. Aerosol species reported including 2-methyltetrols, 2-methylglyceric acid, $C_5$-alkene triols and organosulfates (OSs) (i.e. Edney et al., 2005; Surratt et al., 2007a, 2010; Riva et al., 2016; Spolnik et al., 2018). While many of these are formed through multiphase chemistry (e.g. IEPOX channel), we cannot exclude their gas phase formation at least for 2-methyltetrols and 2-methylglyceric acid, as these compounds have been linked to gas phase reaction products from the oxidation of isoprene (Kleindienst et al., 2009) and in ambient $PM_{2.5}$ (Xie at al., 2014). Moreover, these compounds were identified in ambient $PM_{2.5}$ in several places around the world, and SOA from isoprene often accounts for 20–50% of the overall SOA budget (Claeys et al., 2004a; Wang et al., 2005; Henze and Seinfeld, 2006; Kroll et al., 2006; Surratt et al., 2006; Hoyle et al., 2007)."

The following references were added to the reference section.

Mingjie X., Hannigan M.P. and Barsanti K.C.: Gas/Particle Partitioning of 2-Methyltetrols and Levoglucosan at an Urban Site in Denver, Environ. Sci. Technol., 48, 2835−2842, 2014.

Kleindienst T.E., Lewandowski M., Offenberg J.H., Jaoui M. and Edney E.O.: The formation of secondary organic aerosol from the isoprene + OH reaction in the absence of $NO_x$., Atmos. Chem. Phys., 9, 6541–6558, 2009.

Thompson S.L., Reddy L.N. Yatavelli, Harald Stark, Joel R. Kimmel, Jordan E. Krechmer, Douglas A. Day, Weiwei Hu, Gabriel Isaacman-VanWertz, Lindsay Yee, Allen H. Goldstein, M. Anwar H. Khan, Rupert Holzinger, Nathan Kreisberg, Felipe D. Lopez-Hilfiker, Claudia Mohr, Joel A. Thornton, John T. Jayne, Manjula Canagaratna, Worsnop D.R. and Jimenez J.L: Field intercomparison of the gas/particle partitioning of oxygenated organics during the Southern Oxidant and Aerosol Study (SOAS) in 2013, Aerosol Sci. Technol., 51, 30-56, 2017.

**Comment #3.** *"Page 2, Line 12: correct the spelling of "Claeys" here."*
**Response.** We thank the reviewer for this. This has been corrected.

**Comment #4. "**Page 2, Line 18: Probably worth citing Surratt et al. (2006, JPCA) and Surratt et al. (2010, PNAS) for the NOx concentration having an affect on the isoprene SOA composition"
**Response.** This was updated as suggested by the reviewer.

**Comment #5. "**Page 2, Lines 22-24: Citations to the published literature are needed for this sentence."
**Response.** To reflect the reviewer comment, we added the following references on page 2, line 24 (original manuscript): (Lin et al., 2013; Budisulistiorini et al., 2016; Rattanavaraha et al., 2016; Gaston et al., 2014; Riedel et al., 2015; Zhang et al., 2018).

These new references were added to the revised manuscript.

Lin Y.-H., Knipping E. M., Edgerton E. S., Shaw S. L., and Surratt J. D.: Investigating the influences of $SO_2$ and $NH_3$ levels on isoprene-derived secondary organic aerosol formation using conditional sampling approaches. Atmos. Chem. Phys., 13, 8457–8470, 2013.

Budisulistiorini S.H., Baumann K., Edgerton E.S., Bairai S.T., Mueller S., Shaw S.L., Eladio M. Gold K.A. and Surratt J.D.: Seasonal characterization of submicron aerosol chemical composition and organic aerosol sources in the southeastern United States: Atlanta, Georgia, and Look Rock, Tennessee, Atmos. Chem. Phys., 16, 5171–5189, 2016.

Zhang Y., Chen Y., Lambe A.T., Olson N.E., Lei Z., Craig R.L., Zhang Z., Gold A., Onasch T.B., Jayne J.T., Worsnop D.R., Gaston C.J., Thornton J.A., Vizuete W., Ault A.P. and Surratt J.D.: Effect of the Aerosol-Phase State on Secondary Organic Aerosol Formation from the Reactive Uptake of Isoprene-Derived Epoxydiols (IEPOX), Environ. Sci. Technol. Lett., 5, 167–174, 2018.

Rattanavaraha, W., Chu, K., Budisulistiorini, S. H., Riva, M., Lin, Y.-H., Edgerton, E. S., Baumann, K., Shaw, S. L., Guo, H., King, L., Weber, R. J., Neff, M. E., Stone, E. A., Offenberg, J. H., Zhang, Z., Gold, A. and Surratt, J. D.: Assessing the impact of anthropogenic pollution on isoprene-derived secondary organic aerosol formation in $PM_{2.5}$ collected from the Birmingham, Alabama, ground site during the 2013 Southern Oxidant and Aerosol Study, Atmos. Chem. Phys., 16, 4897–4914, 2016.

Gaston C. J., Thornton J. A. and Ng N. L.: Reactive uptake of $N_2O_5$ to internally mixed inorganic and organic particles: the role of organic carbon oxidation state and inferred organic phase separations, Atmos. Chem. Phys., 14, 5693-5707, 2014.

Riedel T. P., Lin Y-H., Budisulistiorini S. H., Gaston C. J., Thornton J. A., Zhang Z., Vizuete W., Gold A., and Surratt J. D. : Heterogeneous Reactions of Isoprene-Derived Epoxides: Reaction Probabilities and Molar Secondary Organic Aerosol Yield Estimates, Environ. Sci. Technol. Lett., 2, 38–42, 2015.
Riedel et al., 2015.

**Comment #6.** "Page 2, Lines 24-26: The authors need to also include the fact that the acid catalyzed multiphase chemistry (or reactive uptake) of IEPOX also highly depends on the aerosol phase state. As recently shown by Zhang et al. (2018, ES&T Letters) the reactive uptake of IEPOX is adversely affected if aqueous sulfate aerosol is coated with viscous SOA. This causes a substantial diffusion barrier that the IEPOX can't react in the aqueous acidic core. This recent work is also supported by initial findings presented in Gaston et al. (2014, ES&T) and Riva et al. (2016, ES&T)."

**Response.** We added the following sentences to the revised manuscript on page 2, line 25 (original manuscript) and read:

"However, this type of multiphase chemistry following the uptake of IEPOX can be highly

dependent on the aerosol phase state and the presence of aerosol coatings from viscous SOA constituents (Zhang et al., 2018). Such coatings can cause a substantial diffusion barrier to the availability to an acidic core."

**Comment #7.** "Page 2, Line 26-27: Acid-catalyzed reactions of isoprene ozonlysis products have also been recently reported by Riva et al. (2015, Atmos. Environ.) and Riva et al. (2017, Atmos. Environ.). These are worth mentioning here."

**Response.** We have added the Riva et al., 2016 (we believe the reviewer is referring to 2016 Riva et al. paper and not to a 2015 paper) and Riva et al., 2017 references to the revised manuscript.

Riva, M., Budisulistiorini, S.H., Zhang, Z., Gold, A. and Surratt, J.D.: Chemical characterization of secondary organic aerosol constituents from isoprene ozonolysis in the presence of acidic aerosol. Atmos. Environ. 130, 5-13, 2016.

Riva M., Budisulistiorini S.H., Zhang Z., Gold A., Thornton J.A., Turpin B.J. and Surratt J.D.: Multiphase reactivity of gaseous hydroperoxide oligomers produced from isoprene ozonolysis in the presence of acidified aerosols. Atmos. Environ. 152, 314-322, 2017.

**Comment #8.** "Page 3, Line 5: Don't the authors mean Surratt et al. (2007, ES&T) and not Kleindienst et al. (2007)? This seems strange to me."

**Response.** This is corrected as suggested by the reviewer.

**Comment #9.** "Chemical Artifacts (Potentially Serious Issue): Since filters were collected and extracted and derivatized for GC/EI-MS, can the authors comment on any potential artifacts? The reason this is so important is that recent work by Lopez-Hilfiker et al. (2016, ES&T) showed that IEPOX-derived SOA had a much lower volatility than expected. It turned out that they provided evidence that 2-methyltetrols and C5-alkene triols are likely thermal degradation products from accretion products (oligomers and organosulfates). Can the authors rule out that these novel GC/MS products are not simply thermal degradation products of accretion products found with the SOA?"

**Response.** Artifact peaks associated with silylation (BSTFA) derivatization elute early in the chromatogram and are clearly recognizable from isoprene oxidation products (e.g. by the interpretation of their mass spectra and their occurrence in blank and background samples), which elute after artifact peaks. In addition, BSTFA

artifacts were identified in our work by acquiring blank and/or background chamber, as well as from analyzing about a hundred of standards derivatized in our laboratory, including methyltetrols, and IEPOX.

The point brought by the reviewer associated with accretion products is very important. Lopez-Hilfiker et al. (2016) analyzing samples collected during the 2013 SOAS field study, hypothesize that 2-methyltetrols and C5-alkene triols *are likely thermal decomposition* products from accretion products. These authors analyzed thermograms originated from filters collected on FIGAERO-CIMS system (PM1), and measured the bulk aerosol composition. The possibility of artifacts in our methods or in the FIGAERO-CIMS must always be recognized, and they are typically checked but not necessarily reported. In prior work, we have addressed possible sample handling problems and do not believe that further work is required on our behalf. We feel that the potential issues using the CIMS-FIGAERO instrument might need further exploration to address that instrument-specific formation artifacts and analyte losses (e.g. thermal decomposition, using vacuum and its effect on more volatile compounds losses).

Our data provide strong evidence that 2-methylglyceric acid, 2-methyltetrols and C5-triols exist in our systems as monomers as well as dimers. (See compounds observed as dimer in the figures below as well as Jaoui et al., 2008). The derivatization technique used in this study in not associated with thermal decomposition as speculated by Lopez-Hilfiker et al. (2016). As an example, the figure below shows the presence of monomers: 2-methylglyceric acid and 2-methyltetrols as monomers as well as and dimers (2-methyltetrols_2-methylglyceric acid dimers) in the same or similar system (Jaoui et al. 2008; also SOA from isoprene + ozone). This is consistent that silylation reactions does not leading to isoprene dimers through thermo-decomposition by the extraction and derivatization procedure given that we detect dimers and monomers as TMS-derivative. In addition, we conducted two derivatizations on isoprene SOA extract using the same amount of the extract for each derivatization. Both derivatizations underwent the same steps except one was heated to 70 °C for 1 hour (same technique used in this study), and the second was left for 24 hours at room temperature (no heat). The results show that chromatograms acquired from both derivatization were essentially identical, confirming that the BSTFA derivatization does not lead to thermal decomposition of the TMS-dimer isoprene products. In fact, the silylation derivatization can be a useful tool for identifying the presence of oligomers in SOA as shown in the figure.

[Figure]

**Figure 1.** Total (top) and extracted (bottom) ion chromatograms of an isoprene ozone experiment showing the presence of monomeric and dimers oxidation products using silylation reaction (BSTFA) using the same protocol used in the paper.

[Figure]

**Figure 2.** Mass spectra associated with isoprene monomer 2-methylglyceric acid and dimers observed simultaneously in isoprene SOA.

**Comment #10. "**Page 5, Lines 23-28: How were OC and SO2 emissions estimated from the Poland sites? This needs to be clarified in the experimental section."

**Response.** We changed "section 2.2." to reflect the reviewer comments and read:

"Twenty ambient $PM_{2.5}$ samples were collected, onto pre-baked quartz filters using a high-volume aerosol sampler (DH-80, Digitel), from two sites (ten samples each) having strong isoprene emissions: (1) a regional background monitoring station in Zielonka, in the Kuyavian-Pomeranian Province in the northern Poland (PL; 53°39' N, 17°55' E) during summer 2016 campaign, and (2) a regional background monitoring station in Godow, PL located in the Silesian Province (49°55' N, 18°28' E) in summer 2014 campaign. Sampling times were 12 and 24 hours, respectively. Major tree species at both sites are European oak (*Quercus robur* L.); European hornbeam (*Carpinus betulus* L.); Tilia cordata (Tilia cordata Mill); European white birch (Betula pubescens Ehrh); and European alder (Alnus glutinosa Gaertn). The Zielonka station is in a forested area while the Godow station is located near a coal-fired power station in Detmarovice (Czech Republic). Godow is also close to the major industrial cities of the Silesian region in Poland, and thus aerosol samples collected in Godow were influenced by anthropogenic sources.

Several chemical and physical parameters were measured at the two sites. The relative humidity during sampling was up to 86% in Zielonka and 94% at Godow. Both locations were influenced by $NO_x$ concentration, modestly in Zielonka at 1.3 µg m$^{-3}$ and at a level of 30 µg m$^{-3}$ in Godow, represented by the nearest monitoring station at Zywiec, PL. The $SO_2$ levels at Zielonka were approximately 0.6 µg m$^{-3}$ and 3.0 µg m$^{-3}$ at Godow. At each site, OC/EC values was determined for each filter using a thermo-optical method (Birch and Cary, 1996). The organic carbon value at Zielonka was approximately 1.7 µg m$^{-3}$ and 5.4 µg m$^{-3}$ at Godow, although aerosol masses were not determined."

**Comment #11. "**Table 1: It would be easier if you could label on Table 1 what the sulfate mass concentrations were in there. Also, why didn't the authors consider running a thermodynamic model like ISORROPIA to estimate aerosol acidity. That way you can estimate what the aerosol acidity is as a function of RH. Obviously, as RH is increasing it is adding more water to your particles that you atomize the same way at each test condition, and thus, your pH is becoming less acidic."

**Response.** To address the reviewer comment, we updated the caption of Table 1, and now reads:

"**Table 1.** Initial and steady state conditions, yields and OM/OC data for chamber experiments on isoprene

photooxidation in the presence of acidic and non-acidic seed aerosol. The initial NOx was entirely nitric oxide. Experiment ER667 was conducted at a low-concentration ammonium sulfate seed (~1 µg m$^{-3}$). Experiment ER662 was conducted with a higher concentration of inorganic seed (~30 µg m$^{-3}$) generated from a nebulized solution for which half the sulfate mass was derived from sulfuric acid and the other half from ammonium sulfate (lewandowski et al., 2015). As for using thermodynamic models for aerosol acidity, see the responses to reviewers from Lewandowski et al., 2014, and our response to reviewer # 2, comment 20. A copy of Lewandowski et al. 2015 response to similar comment is provided below and reads:"

"In the varied RH experiments, where [H$^+$]$_{air}$ measurements are of limited value (since the maximum dissociated H$^+$ in the extracts remains unchanged, but actual aerosol pH is expected to change with liquid water content), we agree with the reviewer that aerosol pH levels or aerosol liquid water concentrations would be of tremendous value to the interpretation of the results. Unfortunately, we do not have sufficient composition information to do the modeling with ISORROPIA or AIM appropriately. While chamber temperature, RH, and particle sulfate loading are known for each reaction step, particle phase ammonium and nitrate were not measured in these experiments. And, although not strictly necessary, no gas-phase ammonia or nitric acid concentrations are available (and, as high-NOx experiments, nitric acid concentrations should be non-trivial), further complicating model predictions. Additionally, as with the SOAS field measurements reported in Guo et al., 2014, we have a significant aerosol fraction composed of isoprene-related organic aerosol to contend with, which can further contribute to aerosol phase water content, but is not accounted for in the models. Given these limitations, we believe that any modeled pH levels or aerosol liquid water concentrations that we could generate from our existing data would be suspect and potentially counterproductive to the analysis (due more to our under-analysis of the aerosol composition than to any limitations inherent in the models). While we readily admit that humidity provides, at best, an indirect measure of the physically important aerosol parameters of interest in these comparisons, it is nevertheless the most reliable measurement surrogate that we have to work with in this data set."

See our response to reviewer 2, comment # 20 for addition text added to the revised manuscript.

**Comment #12.** "Page 8, Lines 3-5: Citations are warranted to prior studies that characterized these ions as characteristic ions for organosulfates and nitrooxy organosulfates."
**Response.** We added the following references.

Darer, A.I., Cole-Filipiak N.C., O'Connor A.E. and Elrod M.J.: Formation and stability of atmospherically relevant isoprene-derived organosulfates and organonitrates. Environ. Sci. Technol. 45, 1895-1902, 2011.

Szmigielski, R.,. Evidence for $C_5$ organosulfur secondary organic aerosol components from in-cloud processing of isoprene: Role of reactive $SO_4$ and $SO_3$ radicals, Atmos. Environ. 130, 14-22, 2016.

**Comment #13. "**IEPOX-1 and IEPOX-2 is VERY STRANGE: IEPOX-1 and IEPOX-2 don't make any sense to me. Do the authors mean they are the isomers of 3-MeTHF-3,4-diols? These were first characterized by authentic standards in Lin et al. (2012, ES&T) by the Surratt Group at UNC. 3-MeTHF-3,4-diols."

**Response.** We are not sure what the reviewer is referring here. IEPOX-1 and IEPOX-2 are defined in Table 2 with their structure and nomenclature. These compounds are reported in the literature during the last six years and are isomers of 3-MeTHF-3,4-diols. To reflect the reviewer comment, we added Lin et al., 2012 reference to the revised manuscript.

Lin Y.-H., Zhang Z., Docherty K.S., Zhang H., Budisulistiorini S.H., Rubitschun C.L., Shaw S.L., Knipping E.M., Edgerton E.S., Kleindienst T.E., Gold A., Surratt J.D.: Isoprene Epoxydiols as Precursors to Secondary Organic Aerosol Formation: Acid-Catalyzed Reactive Uptake Studies with Authentic Compounds, Environ Sci Technol., 3, 46, 250–258, 2012.

**Comment #14. "**Table 2 - LC/MS section: MW 230 is the wrong structure. I'm surprised by the carelessness here."

**Response.** We thank the reviewer. This was corrected.

**Comment #15.** *"Page 11, Lines 19-20: You're specific about the other tracers precursors (i.e., IEPOX and MPAN). Why not be more specific here for these recently reported new SOA tracers?"*

**Response.** See response to comment 15, reviewer # 2.

**Comment #16.** *"Page 12, Lines 13-14: What are the uncertainties of using ketopinic acid to quantify all eight isoprene SOA constituents measured by GC/EI-MS?"*

**Response.** Due to lack of authentic standards, we used ketopinic acid for all samples, as we have in past studies, to estimate the changes associated with each compound reported in this study as the RH or acidity changes. As noted in Figures 2, 4, Tables 3, 4 captions, these are necessarily estimates. Without authentic standards, it is difficult to estimates the uncertainties of using ketopinic acid. (See Jaoui et al., (2005) for a further discussion of this point.

**Comment #17.** *"Page 12, Lines 16-17: The fact that you measure 2-methyltartaric acid organosulfate at levels above baseline in your LC/ESI-MS makes me wonder how important this compound really is to isoprene SOA formation. More specifically, what is the exact precursor to this species that forms from the gas-phase oxidation of isoprene?"*

**Response.** See our response to comment #15, reviewer #2.

**Comment #18.** *"Page 14 , Lines 5-7: The terminology "the most abundant were organosulfates derived from 2-methyltetrols (MW 216) and 2-methylglyceric acid (MW 200)" is incorrect. This should really state "Organosulfate monomers derived from acid-catalyzed multiphase chemistry of IEPOX (MW 216) and MAE/HMML (MW 200)" to more accurately reflect their sources (Surratt et al., 2010, PNAS; Lin et al., 2012, ES&T; Lin et al., 2013, PNAS; Nguyen et al., 2015, PCCP). For the IEPOX-derived organosulfates, they are being termed 2-methyltetrol sulfates and 3-methyletrol sulfates to reflect the possible isomers that form from the multiphase chemistry of the IEPOX isomers (i.e., cis- and trans-beta-IEPOX and delta-IEPOX). Recall, Bates et al. (2014, JPCA) showed that the cis- and trans-beta-IEPOX isomers are the predominant isomers that form in the gas phase, with trans-beta-IEPOX being the most abundant. The beta-IEPOX isomers likely lead to the 2-methyltetrol sulfate isomers."*

**Response.** The whole paragraph (page 14, line 4-14: original manuscript was updated and now reads:

**"** The major SOA components detected were 2-methyltetrols, 2-methylglyceric acid and its dimer, whose maximal estimated concentrations exceeded 800, 350 and 300 ng m$^{-3}$ respectively under low-humid conditions of RH 9% (Figure 2). Among compounds detected with LC-MS (Figure 3) are organosulfates derived from acid-catalysed multiphase chemistry of IEPOX (MW 216) and MAE/HMML (MW 200) (Surratt et al., 2010; Lin et al., 2012, 2013; Nguyen et al., 2015). Other components were significantly less abundant. In most cases, increasing the RH resulted in decreased yields of the products detected, although some compounds were observed at higher concentrations at RH 49% compared to RH

9% (i.e. m/z 199: Figure 3). Total SOC decreased with increased RH (Table 1). Generally, the influence of RH on the product yields was mild, with the exception of 2-methyltetrols, 2-methylglyceric acid, and 2-methylglyceric acid dimer which were produced in significantly larger amounts at RH 9% compared to RH 49%. This is generally consistent with Dommen et al. (2006) and Nguyen et al. (2011), who saw a negligible effect of relative humidity on SOA formation in photooxidation of isoprene in the absence of acidic seed aerosol. Two recent studies (Lin et al., 2014; Riva et al., 2016) reported an increase in aerosol mass with increasing RH. Riva et al., (2016) reported also an increase in 2-methyltetrols concentrations with increasing RH. These two studies were fundamentally different than those in the present study. In our study, isoprene was oxidized in the presence of NOx and seed aerosol (acidic and non-acidic) under a wide range of humidity, however hydroxyhydroperoxide (ISOPOOH), and IEPOX were used as reactant in Riva et al., and Lin et al. studies under two RH and free-NOx conditions."

**Comment #19.** *"Page 14, Lines 17-18: I'm not in agreement with this statement. Precursors for organosulfates typically form in the gas phase from the oxidation of isoprene. Such precursors like IEPOX have large Henry's law constants, and thus, can partition into any aerosol water that might be present in the aerosol phase. Thus, the detection of these organosulfates could simply result from the fact that there is enough water on these particles (especially if organics condense and then take up water). I think the authors are unable to rule out this possibility based on their data."*

**Response.** In our sentence on page 14, lines 17-19 that reads "However, organosulfates were also formed in non-acidic experiments, probably through radical-initiated reactions in wet aerosol particles containing sulfate moieties (Noziere et al., 2010; Perri et al., 2010). The NOS and OS compounds we detected could also occur *via* this mechanism."

We did not understand the reviewer concerns here. We refer in this sentence to the formation of organosulfates in the non-acidic seed aerosol and how they may be formed, that is, from where the sulfate group originates. We do agree with the reviewer that IEPOX may be a precursor to these organosulfates in the non-acidic aerosol.

**Comment #20.** *"The ER labelling of experiments is really not helpful to readers. Can't you simply just call one set of experiments the acidic experiment at varying RH and the other one the non-acidic experiment at varying RH?"*

**Response.** The ER label is associated with the chamber used to generate these data. Therefore, for quality assurance purposes and clarity these labels were kept in the revised manuscript, when appropriate. We do not feel this is too distracting for the reader.

**Comment #21.** *"As shown in Table 4 heading, reporting [H+] air concentration isn't really helpful to modeling. Couldn't the authors use one of the thermodynamic models to estimate what the INITIAL pH is of these particles? If the authors recall, McNeill (2015, ES&T), Pye et al. (2013, ES&T) and Maraias et al. (2017, ACP) have developed explicit models to predict IEPOX SOA. These models have been further developed by aerosol flow tube reactors that determine the reactive uptake coefficient of IEPOX as a function of acidity (Gaston et al., 2014, ES&T; Riedel et al., 2015, ES&T Letters), RH (Gaston et al., 2014, ES&T; Zhang et al., 2018, ES&T Letters) and pre-existing SOA coatings (Gaston et al., 2014, ES&T; Zhang et al., 2018, ES&T Letters). It's not clear to me how this data you show in Table 4 and Table 3 can help improve explicit modeling of many of these SOA products. The GAMMA, CMAQ, and GEOS-Chem models all now explicitly predict 2-methyltetrols and the organosulfates derived from the acid-catalyzed chemistry of IEPOX. In addition, some of these models, like CMAQ, now predict 2-methylglyceric acid and the organosulfate derived from MAE/HMML multiphase chemistry. I think much more care is needed by the authors to convince readers and reviewers how this data can be used to further improve these much needed models. I strongly believe these models have to explicitly model the acid-catalyzed multiphase chemistry of isoprene oxidation products that consider the interconnecting effects of aerosol acidity and aerosol phase state, which both depend on the RH condition."*

**Response.** We generally agree with the reviewer's assessment of the limitations of the [H+] air measurement. See our response to comment # 11 above for running a thermodynamic model to estimate the initial pH. Also see our comment to reviewer # 2, comment # 20.

**Comment #22.** *"Figure 4 is poorly generated. Too difficult to read. Please regenerate this figure. Why do some figures use color and others use black and white. I think your figures need to be more consistently generated."*

**Response.** This figure is regenerated as shown in the revised manuscript. All figures in the revised manuscript are now consistent.

**Comment #23.** *"Figure 6: It remains unclear to me how much sulfate was present in all the conditions shown in this figure, the tables of the experimental conditions, and the experimental description. Is sulfate the same concentration in each experiment? 2-MG has been shown to be reduced in concentration if the acidity of the aerosol is high (Nguyen et al., 2015, PCCP). In fact, there is prior evidence that the nucleation of 2-MG and its corresponding oligoesters is enhanced under dry conditions (Nguyen et al., 2011, ACP; Zhang et al., 2011, AcP). I wonder, do you have evidence in your size distribution measurements of nucleation events? I ask this since it appears your sulfate seed aerosol concentrations were quite low at the start of each experimental condition."*

**Response.** See our response to Comment #11 above for sulfate concentrations associated with both experiments. Sulfate concentrations are internally consistent within each experiment, but the two runs use both different inorganic concentrations and different inorganic compositions, as noted in our response to comment #11. See the experimental section of Lewandowski et al. (2015) for additional text describing the experimental conditions. As all experiments start with non-trivial concentrations of inorganic material (at least one-third of the total particle mass in ER662, and at least 5% in ER667), would we expect any organic products formed to condense on existing inorganic particles rather than undergoing nucleation. However, as all sampling was conducted only after the chamber system reached steady-state conditions, transient nucleation events, should they somehow occur despite the presence of preexisting inorganic particles, would not be detected.

**Comment #24.** Surratt et al. (2007, ES&T) - The authors don't compare their results to that paper. That paper showed 2-MG concentration doesn't change with increasing aerosol acidity, but the 2-methyltetrols do.

**Response.** See our response to reviewer # 2, comment # 17.

---

## Author Comment (AC6) · 14 Jul 2018

**Chemical composition of isoprene SOA under acidic and non-acidic conditions: Effect of relative humidity**

**Supplementary Information**

K. Nestorowicz[1], M. Jaoui[2], K. J. Rudzinski[1], M. Lewandowski[2], T. Kleindienst[2], W. Danikiewicz[3] and R. Szmigielski[1]

[1]Environmental Chemistry Group, Institute of Physical Chemistry Polish Academy of Sciences, 01-224 Warsaw, Poland

[2]US Environmental Protection Agency, 109 T.W. Alexander Drive, RTP NC, USA, 27711.

[3]Mass Spectrometry Group, Institute of Organic Chemistry, Polish Academy of Science, 01-224 Warsaw, Poland

*Correspondence to*: Rafal Szmigielski (ralf@ichf.edu.pl); Mohammed Jaoui (jaoui.mohammed@epa.gov)

[Figure]

**Figure S1.** Relative amounts of aerosol components detected with GC-MS acidic seed (pink) and non-acidic seed (yellow) experiments (the areas of the circles are proportional to the estimated mass concentrations of compounds).

[Figure]

**Figure S2.** Relative abundances of aerosol components detected with LC-MS in acidic seed (pink) and non-acidic seed (yellow) experiments (the areas of the circles are proportional to relative abundances of compounds detected).

**Table S1.** Comparison of product yields in acidic seed experiments vs. non-acidic seed experiments at various RH levels ( > higher, = equal and lower < )

| Product | MW | *m/z* | RH = 8 − 9 | RH = 18 − 20 | RH = 28 − 30 | RH = 39 − 49 |
|---|---|---|---|---|---|---|
| 2-methylglyceric acid | 120 | | > | | | |
| 2-methyltetrol OS | | 244 | > | | | |
| 2-methylthreonic acid NOS | | 274 | > | | | |
| furanone OS | | 211 | > | | | |
| 2-methyltetrols | 136 | | > | | | = |
| 2-methyltetrol NOS | | 260 | > | | = | |
| furanetriol OS | | 213 | > | = | | |
| 2-methyltetrol OS | | 215 | > | = | | |
| IEPOX-1 | 118 | | = | > | = | |
| dimer of 2-methylglyceric acid | 222 | | < | > | | |
| C5-diol | 102 | | > | < | = | |
| IEPOX OS | | 197 | > | = | | < |
| 2-methylglyceric acid OS | | 199 | = | < | | |
| IEPOX-2 | 118 | | < | | | |
| 2-methylthreonic acid OS | | 229 | < | | | |

**Figure S3**. Concentrations or relative abundances of some compounds in acidic seed experiments (blue) and non-acidic seed experiments (red) – influence of Relative Humidity

[Figure]

[Figure]

[Figure]

[Figure]

2-methylthreitol (left) and 2-methylerythritol (right)

[Figure]

[Figure]

[Figure]

[Figure]

[Figure]

[Figure]

[Figure]

[Figure]

[Figure]

[Figure]

[Figure]

[Figure]

[Figure]

**Figure S4.** Extracted Ion Chromatograms (EIC) of selected components detected in the respective filter extracts from smog chamber ISO SOA (ER667 – non-acidic seed aerosol; ER662 – acidic seed aerosol) and PM$_{2.5}$ ambient summer aerosol from Godow and Zielonka sites.

[Figure]

2-methyltetrol organosulfate (MW 216)

2-methylthreonic acid organosulfate (MW 230)

2-methyltetrol nitroxy-organosulfate
(MW 261)

2-methylthreonic acid nitroxy-organosulfate
(MW 275)

---

## Author Response (AR2)

**Response to reviewer's comments (# 1)**

The reviewer said in general comment: The authors greatly improved the quality of the paper.
**Author Comment:** We feel the comments of Reviewer 1 have aided in improving the quality.

**Comment # 1.** If the authors want to use acronym they should be consistent (e.g. OS vs OSs) and use them. […] Please go through the article and make it consistent.
**Response:** This was updated as suggested by the reviewer. We replaced "OSs" with "OS" throughout the manuscript.

**Comment # 2.** Line 14, page 2. It needs to be rephrased. […] Instead, they should clearly mention that it is probably due to the re-evaporation of the 2-MT as recently suggested. (Isaacman-VanWertz et al., 2016 ES&T).
**Response:** We agree with the reviewer's comment. We added the appropriate reference and changed the sentence as suggested:

*"While many of these are formed through multiphase chemistry (e.g. IEPOX channel), we cannot exclude their gas phase formation at least for 2-methyltetrols – probably in part through the re-evaporation processes (Isaacman-VanWertz et al., 2016) – and for 2-methylglyceric acid, as these compounds have been linked to gas phase reaction products from the oxidation of isoprene (Kleindienst et al., 2009) and in ambient $PM_{2.5}$ (Xie at al., 2014)."*

*Isaacman-VanWertz, G, Yee, L. D., Kreisberg. N. M, Wernis, R., Moss, J. A, Hering, S. V, de Sá, S. S., Martin, S. T., Alexander, M. L., Palm, B. B., Hu, W., Campuzano-Jost, P., Day, D. A., Jimenez, J. L., Riva, M., Surratt, J. D., Viegas, J., Manzi, A., Edgerton, E., Baumann, K., Souza, R., Artaxo, P., Goldstein, A. H.: Ambient gas-particle partitioning of tracers for biogenic oxidation, Environ. Sci. Technol., 50, 9952–9962, 2016.*

**Comment # 3.** The authors claimed that they cannot estimate the aerosol acidity because they didn't have any particle phase measurement providing the concentrations of inorganic species. […] In this previous study the same authors estimated the aerosol acidity, so why don't they use their previous study to determine the aerosol acidity if the parameter (besides the RH) were identical?
**Response:** The aerosol acidity in Lewandowski et al., 2015 was measured (not estimated) as nmol $H^+$ per $m^3$ air sample volume, an acidity measure that gives air concentration (nmol $m^{-3}$) rather than an aerosol pH. We did report the aerosol acidity in the main manuscript as nmol $H^+$ per $m^3$ in tables 3 and 4 caption for ER667 and ER662 experiments. The average aerosol acidity level estimated for acidic seed and non-acidic seed experiments was 275 nmol $m^{-3}$ and 54 nmol $m^{-3}$, respectively.

As reported in Lewandowski et al., 2015:

*"The [H+] air was calculated by dividing the measured aqueous concentration of hydrogen ions by the volume of air collected, as described by Surratt et al. (2007). While this method provides a simple, easily repeatable measure of bulk acidity, it does not fully capture the actual acidity of individual aerosol particles, which is more likely to be of physical significance in these chemical systems. Nevertheless, in the absence of a true aerosol pH measurement, the [H+] air approach appears to provide a useful surrogate measure under sufficiently constrained experimental conditions."*

See also our previous response to the same reviewer (comment # 5) that reads:

*"We agree with the reviewer that aerosol pH levels or aerosol liquid water concentrations would be of tremendous value to the interpretation of the results. We also generally agree with the reviewer's assessment to use modeling work (i.e., ISORROPIA (Fountoukis and Nenes, 2007); or AIM (Wexler and Clegg, 2002)) of the aerosol acidity and liquid water content, unfortunately, we do not have sufficient composition information to do the modeling with these models (ISORROPIA or AIM) appropriately. While chamber temperature, RH and particle sulfate loading are known for each reaction step, particle phase ammonium and nitrate were not measured in these experiments. And, although not strictly necessary, no gas-phase ammonia or nitric acid concentrations are available (and, as high-NOx experiments, nitric acid concentrations should be non-trivial), further complicating model predictions."*

**Comment # 4.** The experiments performed using AS as seed aerosols are a bit strange. At RH < 40% the seed aerosols should be effloresced and be metastatic (i.e., a crystal). However, the results presented in this study clearly show that the RH matter, which is a bit surprising (e.g., Fig. MS at 9% vs 19%). How do the authors explain such results? […] One explanation might be the larger wall losses due to humid walls but that would imply that 2mGAd are not formed in particle phase.

**Response:** We agree with the reviewer that the aerosol liquid water was probably negligible in the neutral seed case. We attribute this decrease in the 2-methyltetrol, 2-MGA, and 2 MGA dimer to a decrease in the organic aerosol level as observed by Lewandowski et al., 2015. We now include a sentence to state this specifically and mention the possibility that chamber wall-effects may play a role in this decrease as suggested by the reviewer.

We changed the relevant paragraph to improve the readability as given on page 15-16 to:

*"The major SOA components detected were 2-methyltetrols, 2-methylglyceric acid and its dimer, whose maximal estimated concentrations exceeded 800, 350 and 300 ng m$^{-3}$ respectively under low-humidity conditions of RH 9% (Figure 2. At the two lowest humidities, aerosol liquid water is expected to be very low and the decrease in these compounds may not be controlled by aerosol liquid water but possibly by the SOC levels associated with the particles (Lewandowski et al., 2015), although chamber-related wall effects due to water vapor might also play some role. Among compounds detected with LC-MS (Figure 3) are organosulfates derived from acid-catalysed multiphase chemistry of IEPOX (MW 216) and MAE/HMML (MW 200) (Surratt et al., 2010; Lin et al., 2012, 2013; Nguyen et al., 2015). Other components were significantly less abundant. In most cases, increasing the humidity resulted in decreased yields of the products detected, although some compounds were observed at higher concentrations at RH*

*49% compared to RH 9% (i.e. m/z 199: Figure 3). As found in Table 1, total SOC decreased with increased humidity. Generally, the influence of RH on the product yields was modest consistent with Dommen et al. (2006) and Nguyen et al. (2011), who saw a negligible effect of relative humidity on SOA yield in photooxidation of isoprene in the absence of acidic seed aerosol. By contrast, here the 2-methyltetrols, 2-methylglyceric acid, and 2-methylglyceric acid dimer were found in significantly larger quantities at RH 9% compared to RH 49%. Two recent studies (Lin et al., 2014; Riva et al., 2016) reported an increase in aerosol mass with increasing RH. Riva et al., (2016) also reported an increase in 2-methyltetrols concentrations with increasing RH. However, the initial conditions for those two studies differed substantially from that in the present study. Here, isoprene is oxidized in the presence of NOx and seed aerosol (acidic and non-acidic) under a wide range of RH. In contrast, in Riva et al. and Lin et al. studies, the reactants were hydroxyhydroperoxide (ISOPOOH) and IEPOX oxidized under NOx-free conditions at two levels of RH. In addition, organosulfates, 2-methyltetrols and SOA yields derived from isoprene photooxidation typically have been enhanced under acidic conditions (Surratt et al., 2007a,b, 2010; Gomez-Gonzalez et al., 2008; Jaoui et al., 2010; Zhang et al., 2011). Organosulfates were also formed in non-acidic experiments, probably through radical-initiated reactions in wet aerosol particles containing sulfate moieties (Noziere et al., 2010; Perri et al., 2010). The NOS and OS compounds detected here could have been formed via such a mechanism."*

=================================================================================

**Response to reviewer's comments (# 2)**

**Comment # 1.** Q16: Data in Table 4 and Figures 4-5 should be presented and discussed in more detail. Provided response "The presence of 2-methyltetrols and 2-methylglyceric acid and their sulfated analogues in isoprene SOA at a wide range of RH conditions, suggests that SOA water content does not affect significantly their formation" only partly answers the request, but at least please correct to "does not significantly".
**Response:** This was updated as suggested by the reviewer.

**Comment # 2.** Q20: ISOPROPIA should be ISORROPIA
**Response:** The name has been corrected in accordance with the reviewer's

=================================================================================

**Reviewer's comments (# 3)**
This reviewer said in the general comments:
"I thank the authors for revising their manuscript and considering some of my initial comments. Unfortunately, there appears to be some major deficiencies remaining that must be addressed before full publication in ACP can be considered. My largest concern relates to the fact that extreme care must be given when applying GC/Ei-MS (or other thermal analytical methods) to the chemical characterization of isoprene SOA (and likely to other types of SOA in general)"
**Response.** In analytical chemistry, derivatization mainly silylation has been used since the late 1950's in gas chromatography and mass spectrometry for the derivatization of a wide variety of compounds with a wide range of functional groups. Silylation of a polar compound results in reduced polarity, enhanced volatility and increased thermal stability, and enables the GC-MS analysis of many compounds otherwise involatile or too unstable for these techniques. GC/EI-MS analysis of derivatized compounds is not a thermal analytical method as suggested by the reviewer. We showed clearly in our previous revision (see response to comment # 9 from the same reviewer: revision 1)

that thermal degradation of accretion isoprene products does not happen in our system using GC-MS analysis of silylated isoprene reaction products. It is possible that the reviewer may be confusing desulfation with thermal decomposition associated with silylation of organosulfate compounds as shown in Reaction 1 below:

**Reaction 1:** Desulfation reaction of organosulfates upon silylation

[Figure]

Although, desulfation reaction is used regularly mainly in carbohydrates chemistry on sulfated polysaccharides (e.g. glycosaminoglycans) using a variety of desulfation agents (Takano et al., 1992; Kolender et al., 2004; Bedini et al., 206; Bedini et al. 2017 (review)), we are not aware of any study related to desulfation reaction occurring with small organosulfate molecules found in ambient particulate matter. Takano et al. showed the occurrence of desulfation when a silylation reagent was added to carbohydrate sulfates and observed that only primary sulfated alcohols were desulfated (formation of compound 3 from compound 1: Reaction 1). However, sulfated secondary alcohols and other organosulfates were not desulfated (formation of Compound 2 from Compound 1 by Reaction 1). Although some speculation has been reported in the literature of artifacts associated when PM organosulfates are subjected to derivatization, no literature data could be found because of the absence of organosulfate authentic standards. Therefore, more analytical work on desulfation reaction of organosulfates associated with ambient PM is necessary. (References below.)

The reviewer general comment, and Points 1, 2, and 6 make essentially the same argument. The reviewer appears to use this review to promote the LC-MS technique described by Cui et. 2018 over the method used in this manuscript. It is not clear if the main reviewer argument is with the derivatization only when organosulfate are present, although ambient PM can contain hundreds of non-sulfated organic compounds. In our response below by addressing each of the comments separately, we will hope to highlight the main shortcoming of the reviewer arguments, mainly associated with Cui et al. 2018: purity of the standards used, comparison between GC-MS and LC-MS, CIMS-FIGAERO techniques. That said, to reflect the reviewer comments and concerns, and considering the paper by Cui et al. (2018), we have added the following sentences and references to the manuscript on page 6, line 15:

*"Silylations of polar compounds result in reduced polarity, enhanced volatility and increased thermal stability, and enables the GC-MS analysis of many compounds otherwise involatile or too unstable for these techniques. Therefore, appropriate caution should be taken, for example, with desulfation reactions associated with primary organosulfates (Takano et al., 1992; Kolender et al., 2004; Bedini et al., 2006; Bedini et al., 2017; Cui et al., 2018), and corrections might be warranted when analyzing methyltetrols."*

Takano, R., Matsuo, M., Kamei-Hayashi, K., Hara, S., and Hirase, S. A.: Novel regioselective desulfation method specific to carbohydrate 6-sulfate using silylation reagents, Biosci. Biotech. Biochem., 56 (10), 1577-1580, 1992.

Kolender, A. A., Matulewicz, M. C.: Desulfation of sulfated galactans with chlorotrimethylsilane. Characterization of b-carrageenan by [1]H NMR spectroscopy, Carbohydr. Res., 339, 1619–1629, 2004.

Bedini, E., Laezza, A., Ladonisi, A.: Chemical derivatization of sulfated glycosaminoglycans, EurJOC., https://doi.org/10.1002/ejoc.201600108, 2016.

Bedini, E., Laezza, A., Parrilli, M.: A review of chemical methods for the selective sulfation and desulfation of polysaccharides, Carbohydr. Polym., 174 (15), 1224-1239, 2017.

Cui, T., Zeng, Z., dos Santos, E. O., Zhang, Z., Chen, Y., Zhang, Y., Rose, C. A., Budisulistiorini, S. H., Collins, L. B., Bodnar, W. M., de Souza, R. A. F., Martin, S. T., Machado, C. M. D., Turpin, B. T., Gold, A., Ault, A. P., and Surratt, J. D.: Development of a hydrophilic interaction liquid chromatography (HILIC) method for the chemical characterization of water-soluble isoprene epoxydiol (IEPOX)-derived secondary organic aerosol, Environ. Sci.: Processes Impacts, DOI: 10.1039/c8em00308d, 2018.

**Comment #1.** "I would argue we (as a research community) need to get away from using GC/EI-MS with prior derivatization for chemically characterizing isoprene SOA, and possibly for other SOA systems. Importantly, a new HILIC/ESI-HR-QTFOMS recently published by Cui et al. (2018, ESPI) from the Surratt group can measure both 2-methyletrols and organosulfates with the SAME non-thermal analytical method without the need of prior derivatization. Further, this is all done in negative ESI mode. Yes, the 2-methyltetrols can be measured by HILIC/ESI-HR-QTOFMS in the negative ion mode and can be resolved from the organosulfates! This is exciting. Furthermore, they showed that 2-methyltetrols measured by GC/MS with prior derivatization was so much higher than HILIC/ESI-HR_QTOFMS, and further showed with authentic standards that the IEPOX-derived organosulfates (i.e., 2-methyltetrol sulfates and 3-methyltetrol sulfates) decomposed into 2-methyltetrols and C5-alkene triols!!! This has to be considered here in this study! Thus, artifacts of GC/EI-MS must be acknowledged and this could affect the interpretation of the current results."

**Response.** We disagree with the reviewer regarding the statement "get away from using GC-MS with prior derivatization". The author appears to be using this review as a medium to promote advantages of the LC technique described by Cui et al, 2018 and in the process, recommend the abandonment of a technique used by many researchers over nearly three decades. We believe that the reviewer should show caution in promoting a method published a month ago which has not been tested independently by the scientific community for a broader range of SOA derived organosulfates. We do see the method referenced by the reviewer (Cui et al., 2018) a step forward to analyze a set of ambient aerosol compounds from isoprene methyltetrols and their corresponding organosulfates. However, as we highlight below, several analytical inconsistencies can be associated with this method, and we feel that some of the data presented in Cui et al. paper does not support many of the arguments of the reviewer's comments. The paper of Cui et al. 2018 is dependent on two important compounds synthesized: (1) Methyltetrols (MT) are used as the starting materials to synthesize methyltetrols organosulfates (MT-OS); (2) MT-OS used for comparing GC-MS and LC-MS methods. The data provided in Cui et al. 2018 appears not to support the purity or the standard procedure of organic chemistry synthesis of MT-OS.

**Synthesis**
**MT:** There is lack of comprehensive experimental data ([1]H NMR, [13]C NMR) therefore the purity could not be verified in Cui et al. paper of MT. The procedure used should be described, mainly the

purification method since up to four stereoisomers can be formed, in the SI and/or in the main manuscript and $^1$H NMR data should be provided for the intermediates as well the MT synthesized (see for example Lessmeier et al. 2018 for methyltetrols synthesis).

Lessmeier J., Dette H. P., Godt A., and Koop T. Physical state of 2-methylbutane-1,2,3,4-tetraol in pure and internally mixed aerosols. Atmos. Chem. Phys., 18, 15841–15857, 2018.

**MT-OS:** The synthesis of 2-MT-sulfate (from synthesized MT) described by Cui et al. 2018 is a three-step approach. The authors did not provide experimental data (NMR, MS, …) necessary to confirm (1) the selectivity reported in the acetylation reaction (step 1); (2) for the structure of the intermediates resulted from the sulfation reaction (step 2); and (3) the reaction of tri-acetylated 2-MT sulfate with ammonia (step 3). It appears that the action of ammonia with tri-acetylated 2-MT sulfate gives rise to deacetylation and desulfation of the sulfated group (removal of all groups therefore data should be provided), which would likely explain the origin of additional signals in the $^1$H NMR spectrum (two singlets at 1.4 and 1.5 ppm) (Cui et al., 2018). It appears also that 3-MeTHF-3,4-diols formation might occur during any synthetic steps, the most likely during acetylation. This is consistent with Figure 1 reported in Cui et al. 2018 as well as our simulated $^1$H NMR for 2MT-OS, 3-MT OS, MT, and methyl-THF-3,4diol (see Figure S1 below).

**Purity**
**MT:** Little if any data is presented in Cui et al. 2018 for MT purity.
**MT-OS:** The analysis of the $^1$H NMR spectrum in Figure S1 of Cui et al., 2018, and our simulated $^1$H NMR for 2-MT-OS, 3-MT-OS, MT, and 3-MeTHF-3,4diol (see Figure S1 below), it appears impurities are present in the synthesized Cui et al. 2-MT-OS. The $^1$H NMR spectrum (Figure S1, Cui et al. 2018) shows a multiplets (singlet) arising at 2.04 ppm, corresponding to the methyl group, which exist in a vicinity of strongly electronegative substituent/group. It is consistent with the presence of $OSO_3H$ moiety in the 2-MT sulfate. However, the existence of the two preceding multiplets at 1.54 and 1.47 ppm (Figure S1, Cui et al. 2018) indicate the presence of impurities bearing methyl group(s). Our simulated $^1$H NMR spectra for MT, 2-MT-OS, 2-MT triacetylated, and 3-MeTHF-3,4-diol derivative (see Figure S1), clearly shows that the source of these multiplets are methyl groups which originate from either the 2-MT and/or 3-MeTHF-3,4-diol molecule. It is evident that the vicinity of the methyl group in 2-MT is less electronegative than in 2-MT sulfate giving rise to a characteristic shift towards the lesser values. Moreover, the peak integration in the $^1$H NMR of the synthesized ammonium 1,3,4-trihydroxy-2-methylbutan-2-yl sulfate (i.e., 2-MT-OS, Figure S1; Cui et al., 2018) is not consistent with the structure of this compound. It is our contention, that Cui et al., 2018 did not provide sufficient analytical data proving the purity and structural assignment (95.5% reported) of the other organosulfate (3-MT-OS) synthesized.

[Figure]

The reviewer main argument is based on Cui et al., 2018 paper. The authors of that paper did not provide consistent analytical data associated with the purity of the methyltetrols as well as the methyltetrols organosulfates. They based their purity on only one [1]H NMR spectrum provided in the SI. They should provide for example [13]C NMR spectra as well as GCMS data for the methyltetrols (starting materials used to the synthesize of MT-OS) as well as the [13]C NMR and [1]H NMR spectra for the MT-OS. In addition, the [1]H NMR spectrum provided in the SI show that impurities are present in their synthesized 2-MT-OS and is not consistent with the purity reported in their paper of 99% for 2MT-OS. The size of the two peaks between 1.4 and 1.6 ppm could not be associated with the organosulfates synthesized, therefore, we believe caution should be taken when referring to Cui et al. 2018 and conclusions associated to the discrepancies between GC-MS and LCMS. For example, Cui et al. 2018 conclude that "*We also demonstrate that conventional GC/EI-MS analyses overestimate 2-methyltetrols by up to 188%, resulting (in part) from the thermal degradation of methyltetrol sulfates. Lastly, C₅-alkene triols and 3-methyltetrahydrofuran-3,4-diols are found to be largely GC/EI-MS artifacts formed from thermal degradation of 2-methyltetrol sulfates and 3-methyletrol sulfates, respectively, and are not detected with HILIC/ESI-HR-QTOFMS.*" We find this statement as speculative since C5-alkene triols and 3-methyltetrahydrofuran-3,4-diols are also formed when no acid seed is present therefore other pathways leading to these compounds should not be ignored. We should not rule out that the overestimation presented by Cui et al. 2018 could also result from the impurities introduced with the starting materials MT.

We disagree with the reviewer comment related to the "GC/EI-MS for the chemical characterization of isoprene SOA". We are not sure if the reviewer issue is with the GC/EI-MS or with the derivatization itself. We believe that "thermal decomposition" does not occur in our case since silylated isoprene species are sufficiently volatile and unlikely to decompose in the injector/column of the GC-MS. This method is used for over 30 years by scientists and researchers not only for small molecules but also for high molecular weight species. The derivatization main

purpose is avoiding thermal decomposition of labile species by making them more volatile. We believe that the reviewer refers to the derivatization itself and not the GC/EI-MS, although the Surratt group (since the reviewer refer to this group) used this technique in their published work. We do not come to the same conclusion for the work of Cui et al 2018 that GC-MS thermal decomposition is the main factor contributing to discrepancies between LC-MS and GC-MS. Cui et al. 2018 does not report how the comparison between GC-MS and LC-MS was done. The silylation reaction need to be done under water free condition, since organosulfates were synthesized in aqueous solution and no recoveries were reported on the extraction from the chamber, and C data were also not reported. For example, these compounds when water is evaporated can lead to lactone formation.

Possible shortcomings should not be ignored when using HILIC/LC-MS including: (1) HILIC analysis is often not reproductible and can be time consuming due to long times needed between analyses for column re-equilibration. (2) Lower separation power of LC-MS (ESI/APCI) compared to GC-MS method when complex systems are analyzed. (3) The software often used in the HRMS can assign a number of possibly ambiguous formulae to a given peak.

**Comment #2.** "Table 2: The C5H10O3 compounds in Table 2 are NOT IEPOX isomers. They are in fact 3-MeTHF-3,4-diols, which are now known to be thermal degradation products from organosuflates (as shown recently in Cui et al., 2018, ESPI). Further, Lin et al. (2012, ES&T) and Zhang et al. (2012, ACP) from the Surratt group showed that GC/MS measures these ions at m/z 262 and 118 as 3-MeTHF-3,4-diols. This was proven with the use of authentic standards. These are not the correct assignments shown here. Furthermore, these compounds are decomposition products of low-volatility products, such as the IEPOX-derived organosulfates (Cui et al., 2018, ESPI)."

**Response.** These compounds were observed also in isoprene/NOx system (see Figure 1 in the main paper) without acidic seed aerosol used, therefore there is no organosulfates in the system. Therefore, these compounds could not be degradation products of organosulfates as shown above.

Figure S1 suggests possible issues with the purity reported by Cui et al. 2018, therefore other compounds bearing methyl groups are present in the synthesized organosulfates; see the peaks between 1.4 and 1.6 ppm in Figure S1 (Cui et al. 2018). These two peaks are mostly associated with methyltetrols and/or 3-MeTHF-3,4-diols as impurities (note methyltetrols were used in the synthesis of organosulfates). Therefore, 3-MeTHF-3,4-diols are not necessarily degradation products or organosulfates as claimed by the reviewer and Cui et al. 2018 when derivatization/GCMS technique is used. The authors did not detect 3-MeTHF-3,4-diols in the LC-MS method, most probably because ESI in negative ion mode is not the appropriate analysis method for these compounds. (See following response as well.)

We do not dismiss the possible presence of 3-MeTHF-3,4-diols in our samples. In fact, we do detect them in both under acidic and non-acidic seed aerosol based on authentic standards (see mass spectrum below: Figures S2 and S3). We do detect IEPOX peaks in our samples (see Figure 1), which elutes later in our chromatogram while having similar mass spectra to the 3-MeTHF-3,4-diols. However, we felt that this was outside the scope of the paper and we have not reported all isoprene compounds we identified.

[Figure]

**Figure S2.** CI-Mass spectrum (methane) of cis-3-MeTHF-3,4-diols (standard).

[Figure]

**Figure S3.** CI-Mass spectrum (methane) of IEPOX in isoprene SOA.

It is our contention that based on our analysis of data reported in Cui et al. 2018, that the 3-MeTHF-3,4-diols are not degradation products of organosulfates, but products from isoprene oxidation.

In addition, Lin et al. 2012, reports that "reactive uptake on the acidified sulfate aerosols through catalyzed intramolecular rearrangement of IEPOX leads to cis- and trans-3-methyltetrahydrofuran-3,4-diols (3-MeTHF-3,4-diols) in the particle phase." Cui et al. 2018, presumptively from the same group, reports that 3-MeTHF-3,4-diols are reaction artifacts from IEPOX derived organosulfates. Although the paper of Cui et al. speculates that 3-MeTHF-3,4-diols are artifacts of GC/EI-MS

analysis. It is not clear which statement is correct. Furthermore, the paper of Hu et al. 2015, shows the presence of $C_5$-alkene triols in the Southeast U.S.A. during SOAS field study without invoking decomposition arguments.

For the $C_5H_{10}O_3$ compounds, several structures can be associated with this formula including IEPOX (and isomers), and 3-MeTHF-3,4-diols (and isomers). It is not clear what arguments the reviewer makes in stating that these are 3-MeTHF-3,4-diols. Our results show the presence of both IEPOX and 3-MeTHF-3,4-diols in isoprene SOA as well as in gas phase. Due to the number of products we identify in this system, we report only the main products consistent with the objective of this study. Therefore, we did report IEPOX in Table 2 and not 3-MeTHF-3,4-diols. 3-MeTHF-3,4-diols eluting earlier in the chromatogram than IEPOX. Figures S2 and S3 shows mass spectra associated with 3-MeTHF-3,4-diols (standard) and IEPOX (isoprene SOA). Both IEPOX and 3-MeTHF-3,4-diols do indeed have similar fragmentation patterns.

Lin YH, Zhang Z, Docherty KS, Zhang H, Budisulistiorini SH, Rubitschun CL, Shaw SL, Knipping EM, Edgerton ES, Kleindienst TE, Gold A, Surratt JD. Isoprene epoxydiols as precursors to secondary organic aerosol formation: acid-catalyzed reactive uptake studies with authentic compounds. Environ Sci Technol., 2011.

W. W. Hu, P. Campuzano-Jost, B. B. Palm, D. A. Day, A. M. Ortega, P. L. Hayes, J. E. Krechmer, Q. Chen, M. Kuwata, Y. J. Liu, S. S. de Sá, K. McKinney, S. T. Martin, M. Hu, S. H. Budisulistiorini, M. Riva, J. D. Surratt, J. M. St. Clair, G. Isaacman-Van Wertz, L. D. Yee, A. H. Goldstein, S. Carbone, J. Brito, P. Artaxo, J. A. de Gouw, A. Koss, A. Wisthaler, T. Mikoviny, T. Karl, L. Kaser, W. Jud, A. Hansel,K. S. Docherty, M. L. Alexander, N. H. Robinson, H. Coe, J. D. Allan, M. R. Canagaratna, F. Paulot,and J. L. Jimenez. Characterization of a real-time tracer for isoprene epoxydiols-derived secondary organic aerosol (IEPOX-SOA) from aerosol mass spectrometer measurements. Atmos. Chem. Phys., 15, 11807-11833, 2015.

**Comment #3.** *"Page 2, Line 6: Again, this statement "although the SOA yield of isoprene tends to be low," is misleading. This is true if you look back at the prior literature from Kroll et al. (2005, GRL), Kroll et al. (2006, ES&T), Edney et al. (2005, AE), but if you consider the multiphase chemistry of its oxidation products (especially IEPOX), then using the SOA yield approach in determining amount of SOA possilbe from isoprene is not a good way to model it. Specifically, the EPA CMAQ model no longer uses the Odum 2-product model approach to constrain the products from isoprene oxidation. Specifically, it models SOA from isorpene as a multiphase chemical processes by modeling the reactive uptake of IEPOX and other important products."*

**Response.** Isoprene yields (even when heterogenous chemistry is considered) reported in these references are low compared to other important biogenic compounds (i.e. alpha-pinene, d-limonene…), and we do not see this as misleading. We do recognize the possibility of heterogenous chemistry in isoprene SOA formation (our group is still involved in heterogenous isoprene chemistry), although many unresolved difficulties are remaining (i.e. mechanistic, analytical, rate constants…) in the role of heterogenous chemistry in isoprene aerosol formation and presently outside of the scope of this work. We emphasize that isoprene SOA yield (ratio of aerosol mass formed to the isoprene reacted), is a "bulk" property, measured and reports in the literature account for many processes involved in the chemistry leading to the aerosol formation including heterogenous chemistry. Therefore, the role of one compound or other (i.e. IEPOX) tends to be immaterial to the aerosol bulk property (i.e. yield).

**Comment #4.** "Page 2, Line 11: change "products include" to "products, including...."
**Response.** This was updated as suggested by the reviewer.

**Comment #5.** Page 2, Line 11: change "reported including" to "previously reported include"
**Response.** This was updated as suggested by the reviewer.

**Comment #6.** "Page 2, Lines 14-15: Previous analytical work suggests as you heat isoprene SOA you see the off gassing of 2-MTs and C5-alkene triols, especially in thermodunders and FIGAERO-CIMS (Lopez-Hilfiker et al., 2016, ES&T). As shown in the FIGAERO-CIMS, the 2-methyltetrols and C5-alkene triol peaks didn't make sense due to their location in the low-volatility section of the thermograms. Thus, as this study showed, 2-methyltetrols and C5-alkene triols were likely thermal degradation products of lower volatility compounds in isoprene SOA. Importantly, the Cui et al. (2018, ESPI) from the Surratt Group recently demonstrated with authentic standrads that the 2-methyletrol sulfates (2MT-OS) and 3-MT-OS degrade in GC/MS with prior derivatization into C5-alkene triols and 2-methyltetrols! This is a huge deal, as it appears that most of the isorpene SOA is in the organosulfate forms of IEPOX (including sulfated oligomers). This has important consequences for the results presented here. Previous statements about artifacts from GC/MS can NO LONGER be neglected and these authors must recognize this now in their analyses."
**Response.** Here the reviewer is interpreting the Lopez-Hilfiker et al. (2016) data and trying to apply it to our data having different conditions and analysis method. The relationship of the FIGAERO CIMS instrument and our method is at best tenuous. The FIGAERO inlet is specifically designed to heat a laboratory or ambient sample collected on a specialized filter (Lopez-Hilfiker et al, 2014). The active heating program volatilizes the components of the collected aerosol to produce gas-phase constituents which are then measured by chemical ionization mass spectrometry. The method has been tested against laboratory sample and ambient field samples. The approach has been described in detail by Lopez-Hilfiker et al. (2014).

The purported relationship of the real-time FIGAERO CIMS approach for studying ambient aerosol and off-line approach for studying laboratory aerosol is unclear to us. First, our collection process is conducted entirely at ambient temperature unlike the FIGAERO CIMS, which uses an active temperature ramp designed to decompose the constituents volatilized from the aerosol sample. As remarked by Lopez-Hilfiger et al. (2014) regarding the instrument operation, "lower volatility components are likely larger molecular weight dimer, trimers, or other oligomeric or extremely low volatility compounds which thermally decompose during desorption." As best we can tell from the figure of Lopez-Hilfiker et al. (2016), the decomposition temperature for the isoprene-derived organic sulfates in an ambient sample is in the range 100-150 C. The suggestion that there is a temperature decomposition in our collection is unfounded given the collection is conducted at ambient temperature. The derivatization reaction, which occurs in a condensed phase, is extremely rapidly in forming the TMS derivatives leaving little, if any time, for decomposition to occur with the modest heating employed to ensure quantitative conversion. The same derivative conversions have been found when the BSTFA reaction takes place at room temperature over a more extended period (e.g., overnight). As noted by Lopez-Hilfiker et al., (2014), the FIGAERO system is "designed" to enhance decomposition of the analytes. By contrast, our method is designed to retain the analytes in their original (undecomposed) form. Thus, we feel that the reviewer's argument is

unsupported. Again, this is a single paper from which the reviewer wishes to make overly broad statements applied to our paper.

The reviewer's comment that methylketols are degradation products of isoprene organosulfates is not consistent with our data. First, we detect methyltetrols in systems that do not have organosulfates and therefore it is unreasonable to think they are degradation products of organosulfates. Second, we responded to the same reviewer (first revision) that when isoprene SOA was silylated, both methyltetrols and dimers were detected as the silylated derivatives, therefore thermal degradation is unlikely to be occurring.

**Comment # 7.** "Page 3, Lines 31-33: The authors should note that "Subsquent studies..." is not correct for the fact they cite Zhang et al. 2011 (ACP). Zhang et al. (2011, ACP) was published before Nguyen et al. (2011), right? They were published very close together though."
**Response.** To reflect the reviewer concern, "Subsequent" was replaced by "other"

**Comment #8.** "Page 5, Lines 28-30: What was the average RH? Especially during the day ?"
**Response.** We added the RH as suggested by the reviewer.

**Comment #9.** "Page 5, Line 32: As shown by the many previous studies (e.g., Gaston et al., 2014, ES&T; Riedel et al., 2015, ES&TL; etc.) sulfate aerosol is the most important field parameter to report over SO2 concentrations. Its the sulfate particles that provide the surface for the multiphase chemical reactions to occur on."
**Response.** Only SO2 were measured in this study.

[revised manuscript text omitted]

15   5, 167−174, 2018.